# Vagal sensory neurons mediate the Bezold–Jarisch reflex and induce syncope

Jonathan W. Lovelace[1,2,6], Jingrui Ma[1,2,6], Saurabh Yadav[1,2], Karishma Chhabria[3], Hanbing Shen[2], Zhengyuan Pang[2], Tianbo Qi[2], Ruchi Sehgal[2], Yunxiao Zhang[2], Tushar Bali[2], Thomas Vaissiere[4], Shawn Tan[2], Yuejia Liu[2], Gavin Rumbaugh[4], Li Ye[2], David Kleinfeld[1,3], Carsen Stringer[5] & Vineet Augustine[1,2✉]

Visceral sensory pathways mediate homeostatic reflexes, the dysfunction of which leads to many neurological disorders[1]. The Bezold–Jarisch reflex (BJR), first described[2,3] in 1867, is a cardioinhibitory reflex that is speculated to be mediated by vagal sensory neurons (VSNs) that also triggers syncope. However, the molecular identity, anatomical organization, physiological characteristics and behavioural influence of cardiac VSNs remain mostly unknown. Here we leveraged single-cell RNA-sequencing data and HYBRiD tissue clearing[4] to show that VSNs that express neuropeptide Y receptor Y2 (NPY2R) predominately connect the heart ventricular wall to the area postrema. Optogenetic activation of NPY2R VSNs elicits the classic triad of BJR responses— hypotension, bradycardia and suppressed respiration—and causes an animal to faint. Photostimulation during high-resolution echocardiography and laser Doppler flowmetry with behavioural observation revealed a range of phenotypes reflected in clinical syncope, including reduced cardiac output, cerebral hypoperfusion, pupil dilation and eye-roll. Large-scale Neuropixels brain recordings and machine-learning-based modelling showed that this manipulation causes the suppression of activity across a large distributed neuronal population that is not explained by changes in spontaneous behavioural movements. Additionally, bidirectional manipulation of the periventricular zone had a push–pull effect, with inhibition leading to longer syncope periods and activation inducing arousal. Finally, ablating NPY2R VSNs specifically abolished the BJR. Combined, these results demonstrate a genetically defined cardiac reflex that recapitulates characteristics of human syncope at physiological, behavioural and neural network levels.

Homeostatic reflexes through sensing and integrating internal states are crucial for survival, motivation and emotional expression[1]. Peripheral sensory neurons send continuous signals to the brain about the visceral state for interpretation and processing. Dysfunction of these interoceptive signals has not only been implicated in physiological diseases but also psychiatric and neurological disorders. The heart is a vital organ that lies at the crossroads of autonomic physiology and mental functions such as emotion and cognition[5,6]. However, little is known about how the brain integrates and responds to cardiac signals. There is a major understudied cardiac sensory network that transmits beat-to-beat information to the central nervous system (CNS) through the vagus nerve (associated nodose ganglia), dorsal root ganglia and other peripheral ganglia[7]. In fact, PIEZO ion channels that mediate the baroreflex are the only well-defined genetic components of this cardiac afferent network[8]. PIEZO2 VSNs form claw-like structures and surround the aortic arch to regulate the baroreflex[9]. Beyond this reflex, there are other excitatory and inhibitory homeostatic reflex arcs that are associated with different anatomical cardiac locations, such as the atrial Bainbridge reflex (causes tachycardia)[10] or the ventricular BJR (first reported in 1867 and causes bradycardia)[2,3,11]. However, because of the closed loop nature of the cardiovascular system, it has been difficult to disentangle these various reflex arcs. Thus, it is imperative to genetically dissect cardiac sensory pathways to better understand heart physiology and its influence on brain states and behaviour.

The transcriptomic identities, anatomical organization and functional role of cardiac VSNs remain mostly unknown. In particular, the cardiac ventricles are innervated by VSNs with mainly unmyelinated c-fibres[12,13]. Medical textbooks postulate that activation of these VSNs gives rise to the cardioinhibitory BJR that causes bradycardia and systemic hypotension, which in turn leads to syncope[14,15]. Syncope, or the 'little faint', is associated with a transient loss of consciousness and postural tone followed by rapid recovery[16]. Although syncope is a frequent cause for visits to the emergency department and has a 40%

[1]Department of Neurobiology, University of California, San Diego, CA, USA. [2]Department of Neuroscience, Scripps Research, La Jolla, CA, USA. [3]Department of Physics, University of California, San Diego, CA, USA. [4]University of Florida–Scripps Biomedical Research, Jupiter, FL, USA. [5]HHMI Janelia Research Campus, Ashburn, VA, USA. [6]These authors contributed equally: Jonathan W. Lovelace, Jingrui Ma. ✉e-mail: vaugustine@ucsd.edu

lifetime prevalence[17,18], it has not been mechanistically investigated because of the lack of genetically tractable animal models[19]. To address these issues, we set out to genetically identify and characterize the VSNs that underlie the BJR and to investigate the role of these VSNs in syncope induction and chart their influence on CNS networks.

## Genetic and anatomical heart–brain links

We reasoned that because the ventricular wall is anatomically separate from the aortic arch, there could also be genetic segregation. Thus, to gain a specific genetic handle on VSNs that innervate the ventricular wall, we reanalysed single-cell RNA sequencing (scRNA-seq) data[20] of the nodose ganglia and searched for groups that were separate from PIEZO2. NPY2R VSNs formed a distinct genetic cluster from PIEZO2 (Extended Data Fig. 1a–d) and were also previously reported to modulate autonomic function[21]. Next, to assess the existence of cardiac innervation, we bilaterally injected AAV.PHP.S-DIO-gCOMET into the nodose ganglia of NPY2R-Cre mice to specifically label sensory but not motor neurons (Fig. 1a,b). VSNs project to the brainstem[22], and, as expected, we observed dense terminals in the nucleus of the solitary tract (NTS) and the area postrema (AP; Fig. 1c). There are limited studies of cardiac innervation by VSNs because the fibres are thin and spread out[23,24]. Moreover, the heart is dense, opaque and filled with blood, which causes autofluorescence and impedes imaging. To overcome these obstacles, we used a new whole-organ tissue-clearing approach, HYBRiD, followed by high-resolution light sheet microscopy[4]. We observed larger numbers of NPY2R VSN fibres in the ventricular wall compared with the atria or arch (Fig. 1d, Extended Data Fig. 1e and Supplementary Video 1). Specifically, we observed two types of putative sensory endings: end nets (73.4%) and flower sprays[12] (26.6%; Fig. 1d). Innervation of other visceral organs was also observed (Extended Data Fig. 1f), such as the known NPY2R vagal afferents from the lung[21]. To investigate whether a single VSN projects to multiple organs, we performed paired injections of retro-AAVs with distinct fluorophores into the heart–lung and heart–gut of NPY2R-Cre mice (Fig. 1e and Extended Data Fig. 2a). There was negligible double labelling in the nodose, which indicated that distinct subsets of NPY2R VSNs project differentially to the heart, lungs and gut (Fig. 1f,g and Extended Data Fig. 2b,c). Within the brainstem, there was spatial segregation of heart, lung and gut NPY2R VSN terminals. Notably, the AP predominantly received innervation from heart VSNs, whereas the NTS was labelled by heart, lung and gut VSNs (Fig. 1h,i and Extended Data Figs. 2d,e and 3a). Nerve fibres were only observed in the organs that received retro-AAV injection, which further confirmed the lack of collateralization (Fig. 1j and Extended Data Fig. 2f,g). Parallel results were obtained from organ-specific retrograde tracing in VGLUT2-Cre mice (Extended Data Fig. 2h–l), heart–trachea pairs (Extended Data Fig. 2m–o) and retrograde tracing from the AP in NPY2R-Cre mice (Extended Data Fig. 3b–f). Taken together, these findings support the idea that there is a one-to-one map for organ innervation by VSNs, and heart-projecting NPY2R VSNs mainly target the ventricular wall and dominate AP innervation.

## Heart-innervating NPY2R VSNs in syncope

The impact of cardiac VSN manipulation on behaviour has been understudied. As there is spatial organization of heart ventricle, lung and gut vagal terminals in the brainstem[25,26] (Fig. 1h and Extended Data Figs. 2d and 3a), we reasoned that placing an optic fibre over the AP in NPY2R–channelrhodopsin (ChR2) mice would be an effective strategy to stimulate predominantly ventricular terminals (Fig. 1k). This method enabled us to not only probe the influence of ventricular VSNs on cardiovascular physiology but also on behaviour in awake freely moving animals (vagal NPY2R to AP stimulation (vNAS)). Within a few seconds of vNAS, mice spontaneously fell over and

became immobile (Fig. 1l and Supplementary Video 2). This response was not observed in mice with improper fibre position or in control mice (Extended Data Fig. 4a). Analgesics had no effect, which makes it unlikely that pain responses are causes of this effect (Extended Data Fig. 4b).

To gain deeper insight into this phenotype, we sought to record brain activity in freely moving mice during vNAS and to develop an unbiased electrophysiological biomarker by recording electroencephalograms (EEGs) alongside vNAS (Fig. 1k). Sudden, large drops (>50%) in EEG power within the 8–100 Hz band, predominantly in the gamma range (Fig. 1m and Extended Data Fig. 4c,e), were observed. Notably, two independent metrics—time to 50% power drop and time to immobility—were similar (unpaired $t$-test, $P = 0.85$, comparing Fig. 1n). This power drop was not observed in mice with improper fibre position (Extended Data Fig. 4d). EEGs in human patients with syncope also show reduced brain oscillations over the broad frequency bands that we observed[27,28]. Therefore, our data suggest that vNAS may induce syncope and that latency to a 50% EEG power drop in the 8–100 Hz range is a reasonable and unbiased estimate for syncope onset.

## Physiological phenotypes of vNAS

Next, we examined the physiological effects of vNAS on cardiovascular function. Photostimulation induced a time-locked slowing of the electrocardiogram (ECG; Fig. 2a). This effect was scalable, with higher frequencies causing a larger dip in heart rate (Fig. 2b). Next, we developed a new strategy to combine high-resolution ultrasonography alongside optogenetics (opto-ultrasound; Fig. 2c). This method enabled us to visualize different cardiac views in real time (Extended Data Fig. 5a) and to quantify cardiovascular parameters with genetically defined manipulation of VSNs. vNAS stalled heart beats (Supplementary Video 3) alongside significant decreases in cardiac output, ejection fraction, ascending aorta diameter and aortic valve peak blood-flow velocity. Increases in left ventricular volume, area and aortic acceleration time (the time to reach peak aortic blood flow) were observed. Stroke volume, fractional shortening and other cardiovascular parameters were unaltered, which highlights the specificity of the manipulation (Fig. 2d and Extended Data Fig. 5b,c). Additionally, rapid time-locked changes in blood pressure (decrease; Fig. 2e) and respiration (decrease followed by increase; Fig. 2f) occurred with photostimulation. Lower frequency stimulation did not cause large effects (Extended Data Fig. 5d–h). Furthermore, body temperature was unchanged, which shows that primarily cardiovascular parameters were affected (Fig. 2g). Decreased heart rates were also seen in freely behaving mice (Fig. 2h and Extended Data Fig. 5i). In summary, vNAS induces bradycardia, hypotension and a reduction in respiration rate, all of which are consistent with the physiological changes reported during syncope in humans[16,29,30]. Furthermore, this triad of responses is a hallmark of the BJR, which has long been conjectured to be caused by ventricular VSNs[2,12,13]. As vNAS predominantly activates ventricular sensory fibres, we speculate that NPY2R VSNs may be a substrate for the BJR, which has also been argued to be a trigger for syncope[2,11,31]. Conceptually, these findings, in conjunction with the fact that NPY2R VSNs are distinct from baroreflex-mediating PIEZO2 VSNs, may indicate genetically segregated pathways for various reflex arcs in the framework of the cardiac system.

## Brain-wide activity during syncope

Brain activity and pupil dynamics vary distinctly across a broad spectrum of arousal states, from comatose and sleep to high attentional and fear states[32–34]. Brain state data in the context of syncope are sparse, but some clinical studies show 'flattening' of EEG signals, which indicates drops in broadband frequency oscillations in human patients[27,28,35]. To quantify neural and behavioural states during vNAS-induced

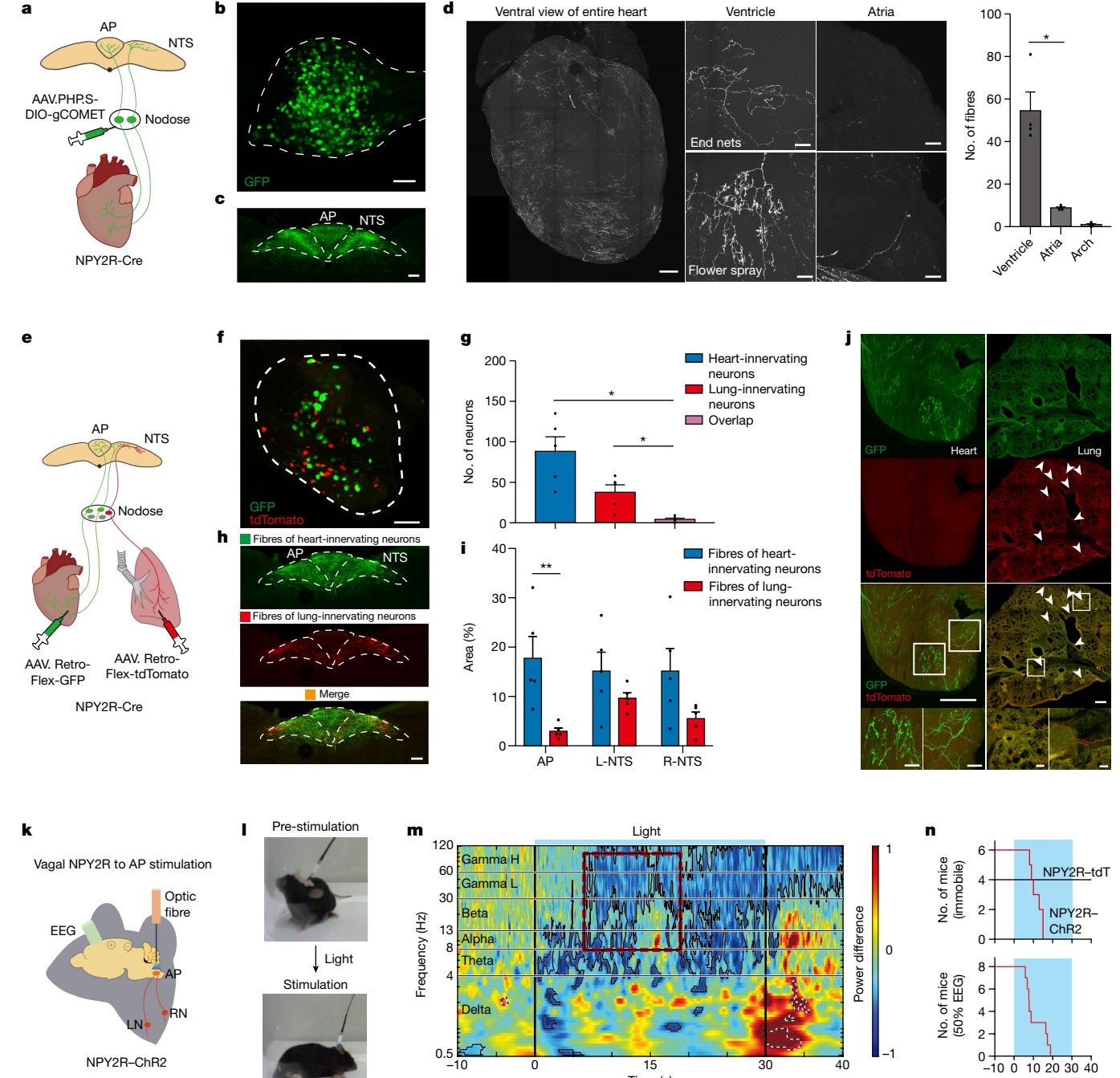

**Fig. 1 | Brainstem stimulation of NPY2R VSNs induces syncope. a**, Schematic of anterograde tracing of NPY2R VSNs. **b**, gCOMET-labelled neurons in a nodose ganglion of a NPY2R-Cre mouse (*n* = 4). **c**, Fibre distribution of NPY2R VSNs (green) in the AP and NTS (*n* = 4). **d**, Left, HYBRiD-cleared heart showing NPY2R VSN terminals in the heart ventricles and atria. Right, quantification of fibre distribution (*n* = 4, *P* = 0.0108). **e**, Schematic of retrograde tracing of NPY2R VSNs from heart and lung. **f**, Retrogradely labelled VSNs from the heart (green) and lung (red, *n* = 5). **g**, Quantification of overlap (*n* = 5, heart/overlap *P* = 0.0198; lung/overlap *P* = 0.0254). **h,i**, Spatial projection pattern of heart-innervating (green) and lung-innervating (red) NPY2R VSNs (**h**) and quantifications (**i**, *n* = 5, *P* = 0.0079). R, right; L, left. **j**, NPY2R VSN terminals in retro-labelled heart and lung. Arrowheads indicate lung terminals. (*n* = 5). **k**, Schematic of optogenetic stimulation of NPY2R VSN terminals in the AP with EEG preparation. **l**, Photostimulation (20 Hz) of freely moving mice causes them to fall over and become immobile. **m**, Power is plotted using wavelets on EEG recordings and normalized to baseline. Mean power during light-off trials was subtracted from light-on trials. Areas of significant drops in power (blue with black border) or increases (red with dashed black/white border) are indicated. Strong decreases (50%) in power were observed (red box, width indicates range of latencies), which indicated syncope (*n* = 12 sessions from 8 mice). H, high; L, low. **n**, Top, step plot showing latency to first bout of immobility in NPY2R–ChR2 mice (*n* = 6) and control NPY2R–tdTomato (tdT) mice (*n* = 4). Bottom, step plot showing latency to 50% power drop (*n* = 8). *\**P* < 0.05, *\*\**P* < 0.01 by two-way repeated measures analysis of variance (ANOVA) with Šidák multiple comparisons or repeated measures ANOVA Geisser–Greenhouse correction with Tukey multiple comparisons. All error bars show mean ± s.e.m. Scale bars, 100 μm (**b**,**c**,**d** (ventricle and atria), **f**,**h**,**j** (bottom four)) or 500 μm (**d** (whole heart), **j** (top six)).

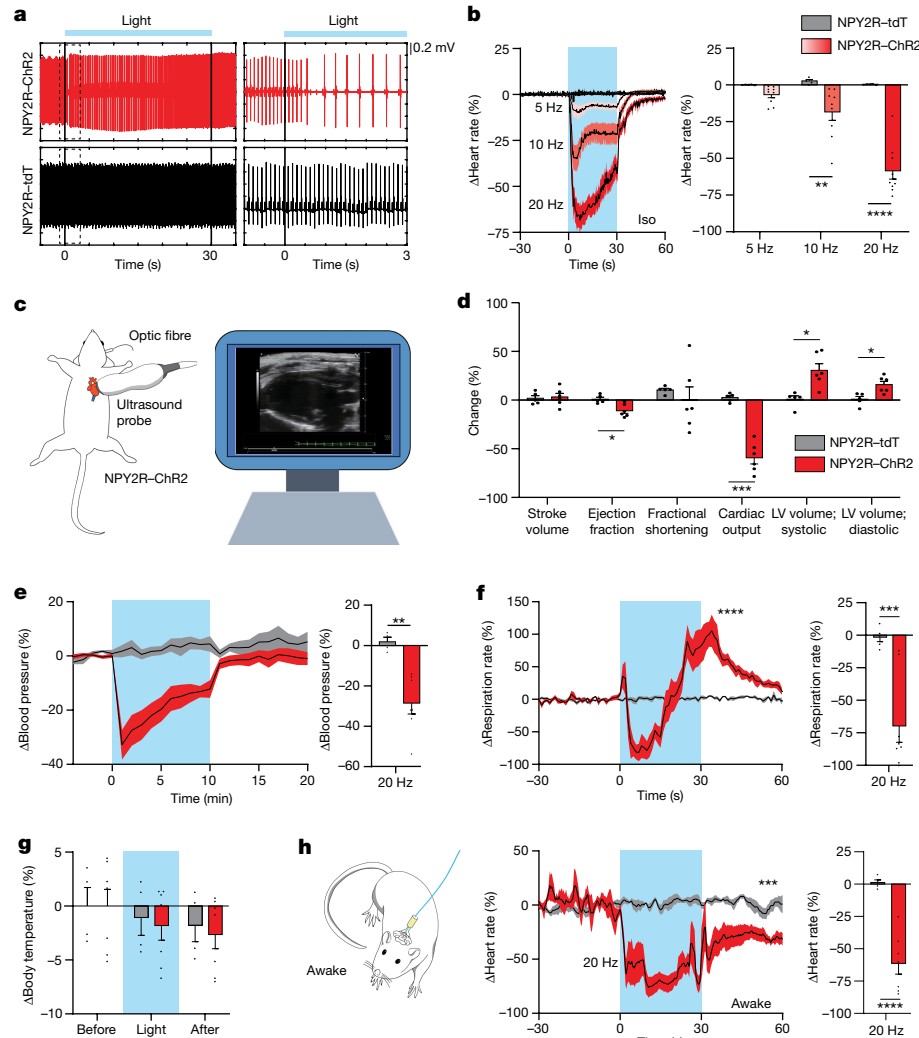

**Fig. 2 | vNAS suppresses cardiovascular function. a**, ECG waveforms recorded under 2% isoflurane. Photostimulation substantially lowered the heart rate. **b**, Average heart-rate traces with different stimulation frequencies (left) and quantification (right) under isoflurane (iso). Increasing photostimulation frequencies reduced heart rates in a scalable manner ($n = 9$ for ChR2 mice, $n = 5$ for tdTomato mice; 10 Hz, $P = 0.0079$; 20 Hz, $P < 0.0001$). **c**, Left, illustration of the opto-ultrasound strategy. During recording, the ultrasound probe was placed on the left chest with an implanted optic fibre above the AP. Right, an example of the parasternal long axis view display. **d**, Photostimulation induced changes in cardiovascular metrics. Both systolic and diastolic left ventricle (LV) volume increased, whereas cardiac output and ejection fraction decreased. Stroke volume and fractional shortening did not change ($n = 6$ for ChR2 mice, $n = 5$ for tdTomato mice; ejection fraction, $P = 0.0282$; cardiac output, $P = 0.0005$; volume; systolic, $P = 0.0282$; volume; diastolic, $P = 0.0282$). Colour key for mice

is used for **e**–**h**. **e**, Left, blood pressure decreased with photostimulation in a time-locked manner. Right, quantification ($n = 7$ for ChR2 mice, $n = 4$ for tdTomato mice; $P = 0.0029$). **f**, Left, respiration was significantly suppressed followed by an increase with photostimulation ($P < 0.0001$). Right, quantification ($n = 8$ for ChR2 mice, $n = 5$ for tdTomato mice; $P = 0.0001$). **g**, No changes in internal body temperature were observed during photostimulation ($n = 7$ for ChR2 mice, $n = 4$ for tdTomato mice). **h**, Left, diagram depicting photostimulation and ECG recordings under freely moving conditions. Middle, photostimulation also significantly decreased the heart rate under awake conditions and remained suppressed after light delivery ended ($P = 0.0001$). Right, quantification ($n = 7$ for ChR2 mice, $n = 5$ for tdTomato mice, $P < 0.0001$). *$P < 0.05$, **$P < 0.01$, ***$P < 0.001$, ****$P < 0.0001$ by two-way ANOVA with Šidák multiple comparisons, two-tailed unpaired $t$-tests with Holm–Šidák multiple comparisons or two-tailed unpaired $t$-tests. All error bars and shaded areas show the mean ± s.e.m.

syncope, we recorded the activity of thousands of CNS neurons using state-of-the-art Neuropixels depth electrodes[36] while capturing facial and eye movements with infrared cameras (Fig. 3a,b). Similar to our EEG results, we found stereotypical significant drops in 8–100 Hz band power in local field potentials (LFPs) from multiple brain regions across mice with vNAS (20 Hz). This power drop was not observed in mice with improper fibre position or with vNAS at lower frequencies (Extended Data Fig. 6). Pupil dynamics during vNAS-induced syncope in head-fixed mice showed a stereotypical pattern of rapid dilation followed by a characteristic eye-roll (Fig. 3c, Extended Data Fig. 7a,b and Supplementary Video 4). This effect mirrors pupil dilation and eye movements reported in human syncope[29,37]. Two independent metrics

of syncope—the latency to eye-roll and latency to LFP power drop—were closely aligned (Fig. 3d). Strong whisking was evident immediately after laser onset at all stimulation frequencies, but mice that fainted at 20 Hz had decreased whisking following initial increase (Extended Data Fig. 7c–f and Supplementary Video 4). Aggregate brain-wide spiking data collected during periods of syncope showed marked suppression of neural activity across a large regional distribution (Fig. 3e and Extended Data Fig. 7g–i). To quantify this effect, we analysed the spiking activity of each unit with respect to syncope onset as defined by an average 50% drop in LFP power (8–100 Hz) across the entire Neuropixels probe (Extended Data Fig. 8a). At syncope onset, most neurons were inactive; that is, they did not fire for a significant period, as defined

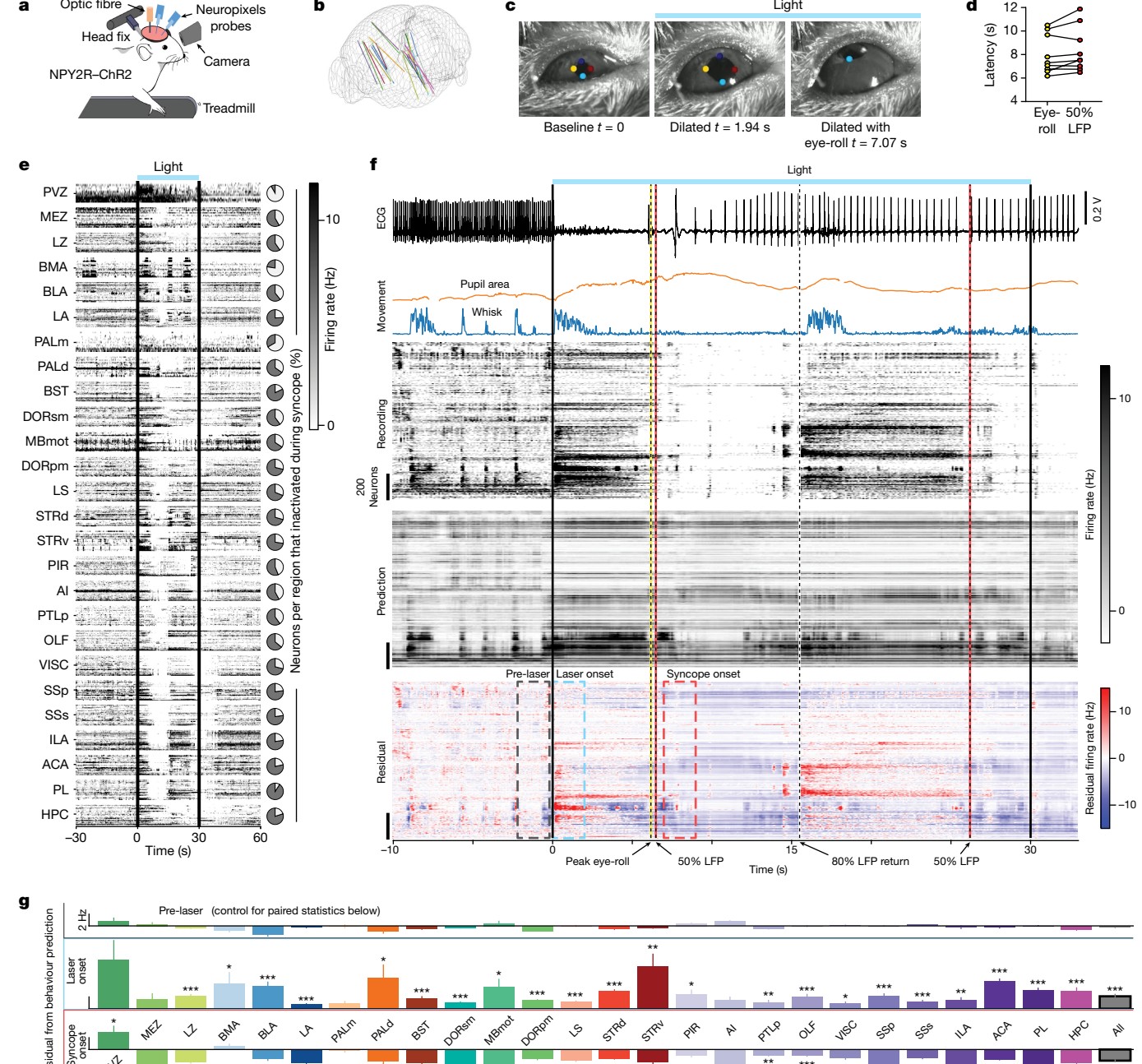

**Fig. 3 | vNAS-triggered syncope is associated with widespread suppression of brain activity. a**, Illustration of head-fixed Neuropixels experiments. Cameras were positioned to record pupil and facial movements. **b**, Neuropixels probe trajectories were registered using SHARPtrack[51] and plotted. **c**, Time course of eye dynamics during optogenetically triggered syncope. The pupil rapidly dilates (middle) followed by a characteristic eye-roll during syncope (right, *n* = 9 sessions from 5 mice). **d**, Latency to eye-rolling behaviour and 50% LFP power drop are correlated (*r* = 0.9596, *P* < 0.0001). **e**, Firing rates of units from all probes and mice that exhibited syncope during 20 Hz photostimulation. Left, neuronal spiking rates in most regions of the brain substantially decreased during syncope. Right, pie charts show the percentage of neurons that were inactive (grey area). **f**, Example of a Neuropixels recording session during 20 Hz photostimulation. Top to bottom: (1) recording showing reduction in heart-rate (ECG); (2) pupil area and whisking behaviour (Movement); (3) raw firing rates across two probes (Recording); (4) predicted firing rates based on movement

model (Prediction); (5) residuals after subtracting predicted firing rates from recorded firing rates (Residual). Relevant events during recording are noted below with arrows. Windows of analysis are shown with dashed colour-coded boxes. **g**, Quantification of changes in brain activity that are not predicted by movements (residuals). Averages of residual activity during the pre-laser time window (top, grey). Small deviations from 0 indicates good predictive capability of the facial movement model under baseline conditions. Residual firing rates increase after laser onset (middle, blue). Significant changes were determined by comparison to pre-laser. Syncope onset showed a widespread decrease in firing rate that was not predicted by the facial movement model, except for the PVZ, which still significantly increased during syncope (bottom, red). *P* < 0.05, **P* < 0.01, ***P* < 0.001 by two-tailed paired *t*-tests. All error bars show the mean ± s.e.m. See Methods for definitions of the abbreviations used for brain regions.

relative to their baseline firing rate. Inactive neurons were individually plotted by region (Extended Data Fig. 8a) and as percentage pie charts (Fig. 3e). A comparison of random baseline conditions (by region) to LFP-defined windows of syncope onset confirmed a substantial, widely distributed suppression of neural spiking (Extended Data Fig. 8b).

## Modelling neural activity during syncope

Spontaneous behaviours such as whisking and pupil dynamics can drive brain-wide neural activity[38–40]. Given that syncope is characterized by these behaviours, interpretation of electrophysiological brain data may be confounded. To address this issue, we built a multidimensional model of behaviour, trained on time periods during recording without vNAS, that can predict a significant proportion of the spiking variance recorded from individual neurons using facial movements (predicted firing rate). A comprehensive plot of recordings and computed measures from a single session is shown in Fig. 3f. vNAS generally elicited an initial increase followed by a substantial reduction in brain activity. Brain regions that were activated after laser onset were plotted in ascending order of latency. The periventricular zone (PVZ) had the shortest recorded latency (around 8 ms) whereas cortical regions generally showed longer latencies. Notably, the longest latency was still in the 60 ms range, which was much faster than physiological changes and indicating rapid brain-wide responses to vNAS (Extended Data Fig. 8c,d). Residual activity, defined as [recorded firing rate] – [predicted firing rate], was plotted across all recordings (Extended Data Fig. 8e). A positive residual shows that the model underpredicted recorded firing data and vice versa. As expected, during the pre-laser period, the model was more accurate at predicting firing data (residual ~ 0 Hz; Fig. 3g, top). At laser onset, however, the model underpredicted (residual > 0 Hz; Fig. 3g, middle), which suggested that neurons increased above their expected activity from the behaviour of the mouse. Notably, at syncope onset, there was overprediction (residual < 0 Hz; Fig. 3g, bottom), which implied that there was brain-wide suppression of activity that was not explained by the cessation of facial movements. Further categorization of unpredicted unit activity showed that the majority of neurons in each region, excluding the PVZ and the basomedial amygdala (BMA), were decreased below behaviour prediction during syncope ('syncope (↓)' in Extended Data Fig. 8f). Notably, the PVZ had the shortest latency to activation after photostimulation onset (about 8 ms; Extended Data Fig. 8c) and was the only brain region we recorded from that remained significantly activated during syncope (Fig. 3g, bottom). Thus, we sought to elucidate the role of the PVZ in syncope. To this end, we used chemogenetics to bidirectionally manipulate (hM4Di for suppression and hM3Dq for activation) the PVZ alongside vNAS (Extended Data Fig. 9a,b,h,i). Activation or suppression of the PVZ did not alter the inhibitory physiological effects of vNAS (Extended Data Fig. 9c,d,j,k). However, PVZ inhibition increased the duration of vNAS-induced syncope (Extended Data Fig. 9e–g). Additionally, pre-vNAS and post-vNAS gamma power was low (Extended Data Fig. 9p). Because gamma band (>30 Hz) activity has been associated with general increases in arousal and excitatory–inhibitory balance[41,42], this result indicated that there is an initial and sustained reduction in arousal state (Extended Data Fig. 9p, see hM4Di CNO (clozapine N-oxide, pre) compared with CNO (post)). PVZ activation during vNAS did not alter EEG power, but two mice that fainted when treated with vehicle ceased to faint (Extended Data Fig. 9l–n). However, PVZ activation by itself increased baseline arousal states (increased velocity (Extended Data Fig. 9o) and increased gamma power (Extended Data Fig. 9p, hM3Dq)) that led to quicker recovery after vNAS. Overall, our Neuropixels analyses demonstrate that syncope is linked to a widespread reduction in CNS activity that cannot be explained by changes in spontaneous behavioural movements. Additionally, manipulating the PVZ alone was sufficient to bidirectionally alter syncope dynamics and brain states without altering cardiovascular physiological changes.

Together, these results may indicate the presence of a central circuit mechanism that modulates these effects.

## Cerebral blood flow during syncope

It has been widely postulated that syncope is due to reduced cerebral blood flow (CBF) by overactivation of the parasympathetic nervous system[43]. Hence, we wanted to test this hypothesis with our vNAS model. Pretreatment with atropine, a parasympathetic blocker, significantly reduced bradycardia and hypotension associated with vNAS, whereas the respiration rate decrease was unaltered (Fig. 4a–c and Extended Data Fig. 10a,b). Notably, atropine also increased the latency to 50% EEG power drop (Fig. 4d–f) and eye-roll, which were indicative of delayed syncope onset (Extended Data Fig. 10c). Atropine by itself did not alter the baseline heart rate ($P = 0.10$). These results confirmed that the effects of vNAS are primarily mediated by the parasympathetic pathway[44]. We next assessed CBF associated with syncope. We developed a method that combined EEG–ECG recordings alongside CBF measurement with laser Doppler flowmetry (LDF) and pupil recordings during vNAS in awake behaving mice (Fig. 4g). vNAS caused asystole that lasted $4.72 \pm 0.75$ s (mean ± s.e.m.) on average, along with cerebral hypoperfusion that started about 200 ms after laser onset (Fig. 4h and Extended Data Fig. 11a, inset). This timing is much slower than brain activation (latency of about 8–60 ms; Extended Data Fig. 8c,d). Pretreatment with atropine prevented asystole and delayed cerebral hypoperfusion (Fig. 4i). With atropine, syncope was still observed but it was delayed (Fig. 4i–m). Atropine also caused blood flow to remain above baseline long after vNAS ended (Extended Data Fig. 11a,b); however, it had no effect on sustained EEG power drop after vNAS (Extended Data Fig. 11d, atropine × pre/post interaction $P = 0.56$). Lower frequency photostimulation (5 or 10 Hz) caused transient drops (<4 s) in CBF that was not sufficient to induce syncope (Fig. 4j,l and Extended Data Fig. 11a). Thus, vNAS may cause CBF to fall below a certain volume threshold (Fig. 4l and Extended Data Fig. 11c). Many physiological processes changed dynamically and simultaneously during vNAS; a 3D plot of heart rate, LDF and power depicting the time course in the heart rate–blood flow–EEG space is shown in Fig. 4n and Supplementary Video 5. Additionally, we observed a patterned sequence of four events during syncope induction, each of which was delayed by atropine: heart rate minima, CBF minima, 50% drop in power and eye-roll (Extended Data Fig. 11e and Fig. 4m). Next, we set out to analyse in further detail behavioural microstructures associated with syncope. To this end, we used an unsupervised machine-learning-based approach: motion sequencing (MoSeq)[45]. vNAS substantially increased the time spent in an immobile state and reduced rearing (Extended Data Fig. 12a–d). This sustained immobile state was not altered with atropine (Extended Data Fig. 12e), even though atropine caused a general increase in EEG power (main effect of atropine $P = 0.01$; Extended Data Fig. 12f). Overall, our findings provide evidence that syncope requires considerable activation of VSNs, which creates a global state that is simultaneously modulated by central circuit processes and cerebral hypoperfusion falling below a volume threshold.

## NPY2R VSNs in the BJR

vNAS is sufficient to cause bradycardia, hypotension and reduced respiration, all of which are trademarks of the BJR[2,12]. To check the necessity and specificity of NPY2R VSNs in mediating the BJR, we used four different approaches. Approach 1 involved a loss-of-function paradigm, whereby AAV.PHP.S-mCherry-Flex-DTA was injected into the nodose of NPY2R-Cre mice to ablate VSNs (Fig. 5a). This ablation abolished syncope and cardiovascular phenotypes associated with vNAS (Extended Data Fig. 13), whereas baseline physiology was unaltered (Extended Data Fig. 14a–d). This result suggests that NPY2R VSNs are

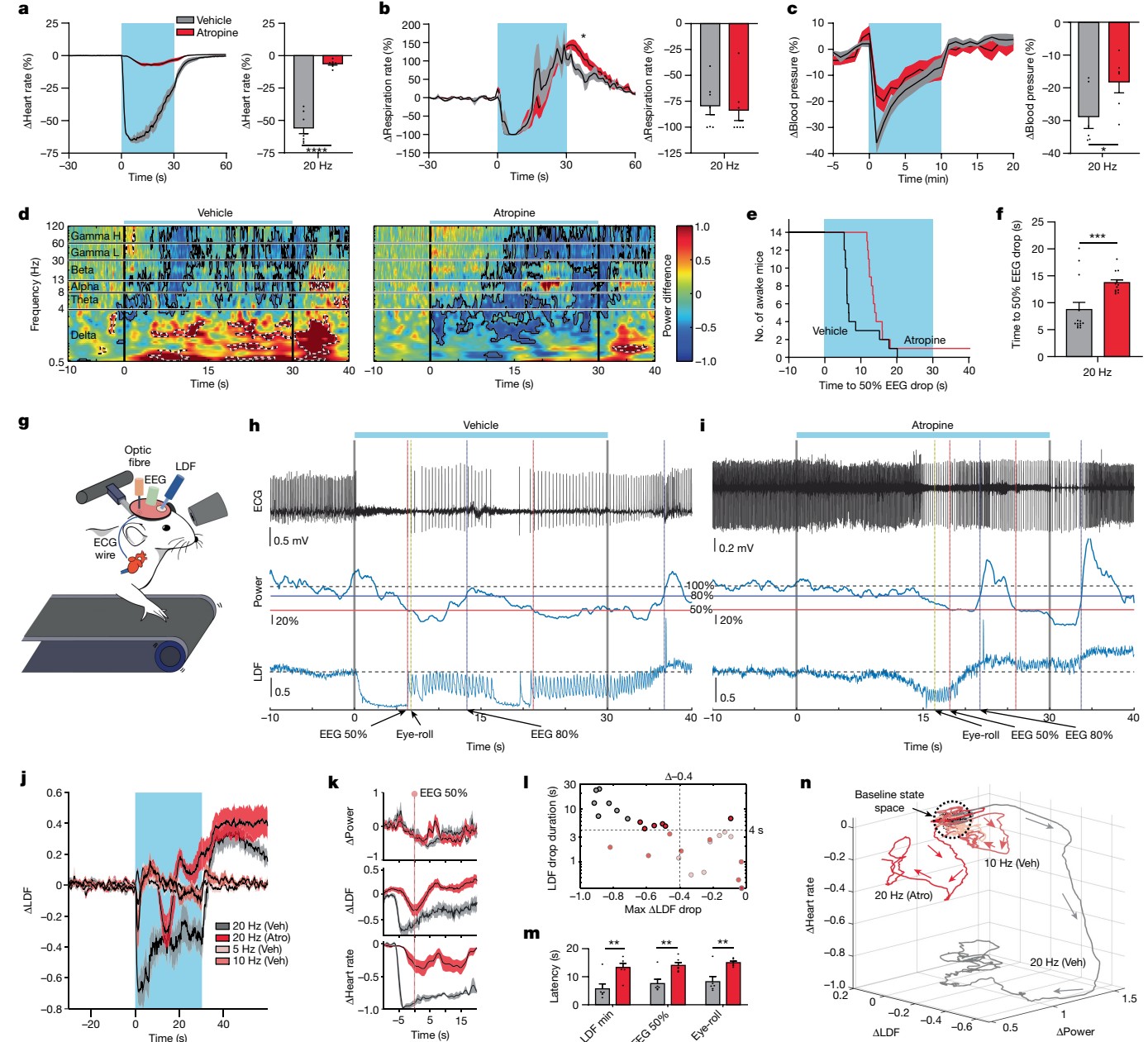

**Fig. 4 | Parasympathetic suppression delays syncope-related phenotypes, including substantial drops in CBF. a**, Left, average heart-rate traces with photostimulation under atropine (Atro) and vehicle (Veh) pretreatment. Right, atropine markedly suppressed heart-rate drops ($n = 7$, $P < 0.0001$). Colour key for mice is used for **b**–**m**. **b**, As for **a**, but for the respiration rate ($n = 7$, after stimulation, $P = 0.0110$; during stimulation, $P = 0.9695$), **c**, As for **a**, but for blood pressure. Atropine reduced drops in blood pressure ($n = 6$, $P = 0.0191$). **d**, Head-fixed mice showed EEG power drops when treated with vehicle (left). Treatment with atropine delayed the latency (right, $n = 14$ sessions from 14 mice). **e**, Step plot showing the delay in latency to power drop ($n = 14$ sessions from 14 mice). **f**, Quantification of **e** shows that atropine increases latencies ($n = 14$ sessions from 14 mice, $P = 0.0002$). **g**, Illustration of head-fixed experiments that recorded CBF using LDF, with ECG, EEG and pupil recording during vNAS. **h**,**i**, Example recordings from the same mouse treated with vehicle (**h**) and atropine (**i**). Top to bottom: (1) ECG; (2) average EEG power normalized to

baseline (Power); (3) LDF signal normalized to baseline reflecting CBF changes (LDF). Specific event markers are noted with arrows. **j**, Average traces of LDF across all experimental conditions ($n = 7$). **k**, Averaged data aligned to latency to reach 50% power drop ($n = 7$). **l**, Scatter plot between the maximum LDF drop during stimulation compared with the duration it remained below 50% of the maximum drop ($n = 7$). **m**, Quantification of the latency increases caused by atropine ($n = 6$; LDF min, $P = 0.0010$; EEG 50%, $P = 0.0041$; eye-roll, $P = 0.0027$). **n**, Plot illustrating the group-averaged dynamics of heart rate, LDF and power, with arrows denoting the progression over time. All groups started in the same state space at the top. After stimulation, groups rapidly diverged while following the same general pattern, even though their magnitudes were different ($n = 7$). *$P < 0.05$, **$P < 0.01$, ***$P < 0.001$, ****$P < 0.0001$ by two-way repeated measures ANOVA with Šidák multiple comparisons or two-tailed paired $t$-tests. All error bars and shaded areas show the mean ± s.e.m.

not involved in the continuous maintenance of resting baseline physiology. Next, we induced the baroreflex or the BJR through injection of the widely accepted substrates phenylephrine (PE; baroreflex), sodium nitroprusside (SNP; baroreflex)[8] or phenyl biguanide (PBG; BJR)[12].

PBG produced dose-dependent dips in heart rate (Extended Data Fig. 14e), which was abolished by NPY2R VSN ablation (Fig. 5c). The baroreflex response to PE–SNP was intact (Fig. 5b), as was other gut-mediated behaviour such as drinking (Extended Data Fig. 14f). This

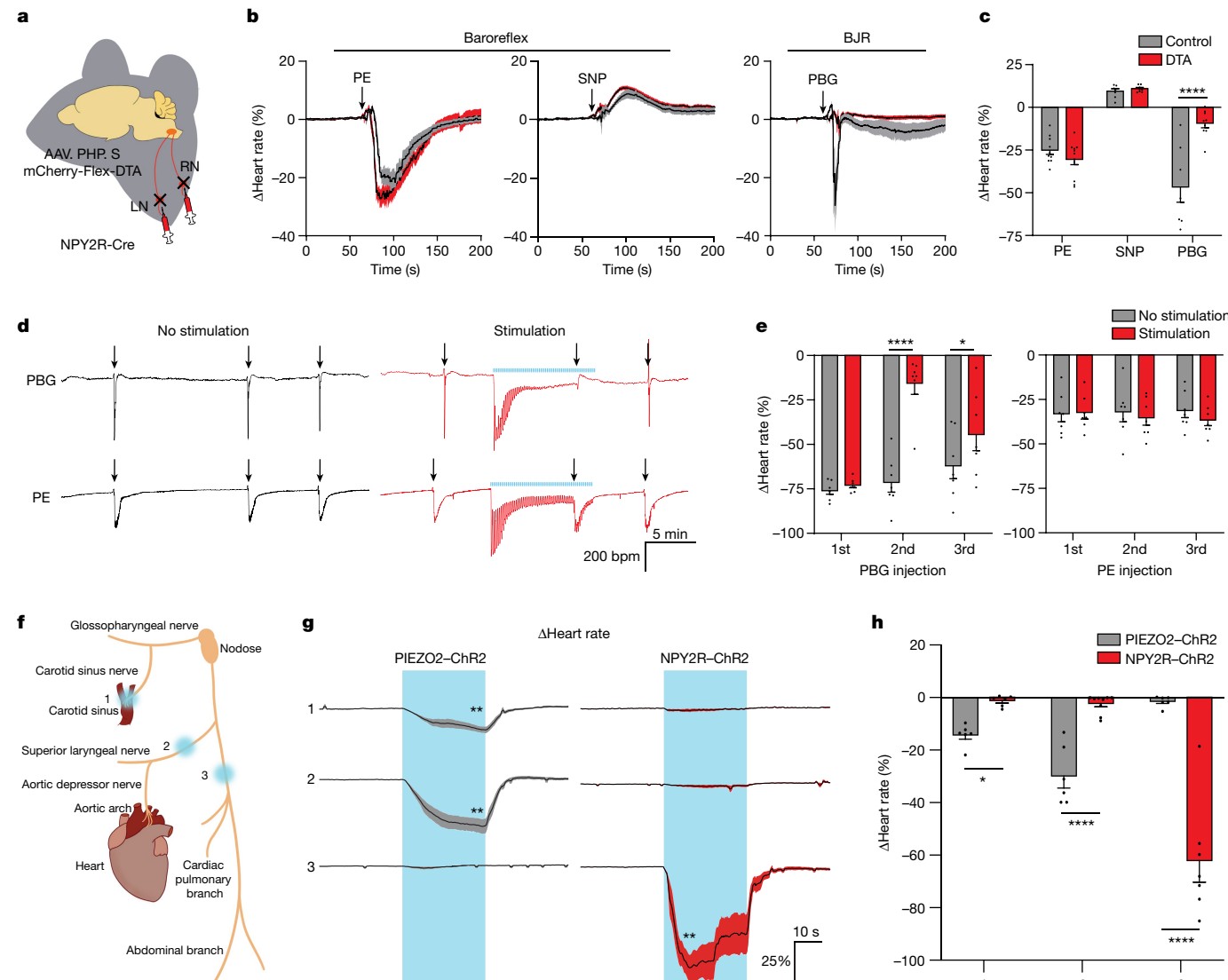

**Fig. 5 | NPY2R VSNs are required for the BJR but not the baroreflex.**
**a**, Schematic for ablating VSNs. **b**, Average heart-rate traces with PE, SNP and
PBG injection. **c**, Quantification of drug efficacy after NPY2R VSN ablation.
The baroreflex was still intact, whereas the BJR was abolished (PE, $n = 10$ for
mCherry, $n = 11$ for diphtheria toxin subunit A (DTA), $P = 0.4699$; SNP, $n = 7$
for mCherry, $n = 8$ for DTA, $P = 0.7969$; PBG, $n = 7$ for mCherry, $n = 9$ for DTA,
$P < 0.0001$). **d**, Heart-rate traces for vNAS adaptation with chemically induced
baroreflex and BJR. PBG and PE were injected before, during and after adaptation.
**e**, Quantification of PBG (second injection, $P < 0.0001$; third injection, $P = 0.0378$)
and PE responses. vNAS adaptation selectively inhibited the BJR but had no
effect on the baroreflex ($n = 7$). **f**, Illustration of the region-specific optogenetic
strategy. PIEZO2-positive baroreceptive vagal afferents are mainly located in
the carotid sinus and aortic arch. The following regions were stimulated: (1) the
carotid sinus; (2) the superior laryngeal branch; and (3) the vagus nerve trunk

above the cardiac branch. **g,h**, Average heart-rate traces (**g**, PIEZO2–ChR2,
regions 1 and 2, $P = 0.0030$; NPY2R–ChR2, region 3, $P = 0.0011$) and quantification
(**h**, region 1, $n = 6$, $P = 0.0386$; region 2, PIEZO2–ChR2, $n = 6$, NPY2R–ChR2, $n = 9$,
$P < 0.0001$; region 3, PIEZO2–ChR2, $n = 6$, NPY2R–ChR2, $n = 7$, $P < 0.0001$) with
region-specific photostimulation in NPY2R–ChR2 and PIEZO2–ChR2 mice.
Stimulation of either region 1 or region 2 did not change the heart rate in
NPY2R–ChR2 mice. By contrast, stimulating region 3 caused an immediate
heart-rate drop only in NPY2R–ChR2 mice. PIEZO2–ChR2 mice showed heart-rate
reductions only in regions 1 and 2, but not region 3, revealing functional
separation between NPY2R and PIEZO2 VSNs. *$P < 0.05$, **$P < 0.01$, ****$P < 0.0001$
by two-tailed paired $t$-tests with Holm–Šidák multiple comparisons, two-way
repeated measures ANOVA with Holm–Šidák multiple comparisons or two-way
ANOVA with Holm–Šidák multiple comparisons. All error bars and shaded areas
show the mean ± s.e.m.

result shows that NPY2R VSNs are specifically needed for the BJR and
not the baroreflex. For approach 2, we performed timed injections
of PE and PBG alongside long-term vNAS (Fig. 5d). Long-term vNAS
caused a reduction in response across time in the form of adaptation
(Extended Data Fig. 10d–g). Therefore, drug-induced reflexes that
share common pathways with vNAS should become occluded during
periods of vNAS adaptation. PE-induced bradycardia was unaffected,
whereas the PBG response was greatly reduced (Fig. 5e). This result con-
firmed that vNAS shares some common pathway with the PBG-induced
BJR and does not overlap with the baroreflex. For approach 3, we did

a comparative study of the effects of stimulating various branches of
the vagus nerve using optogenetics in PIEZO2–ChR2 mice (mediates
the baroreflex) and NPY2R–ChR2 mice (mediates the BJR) (Fig. 5f and
Extended Data Fig. 14g). As expected, photostimulation of baroreflex
afferents (carotid sinus and the superior laryngeal nerves that branch
into the aortic depressor nerve and innervate the aortic arch and the
airways, respectively[8,20]) caused bradycardia in PIEZO2–ChR2 mice
but had no effects in NPY2R–ChR2 mice (Fig. 5g,h, regions 1 and 2).
Stimulation of the vagal trunk at the level of the cardiac branch had
no effect in PIEZO2–ChR2 mice but induced substantial bradycardia

in NPY2R–ChR2 mice (Fig. 5g,h, region 3). Stimulation of abdominal branches had no effect on the heart rate (Extended Data Fig. 14h), which rules out gut-mediated effects. Consistent results were also obtained with respiration rate (Extended Data Fig. 14i,j). This result, combined with drug specificity, shows that NPY2R VSNs are specifically needed for the BJR and not the baroreflex. For approach 4, it was recently shown that stimulation of baroreflex-sensitive neurons in the NTS induces substantial bradycardia and is involved in sleep–wake brain-state regulation[46]. We reanalysed EEG data from that report and found that there was no drop in EEG power (within 8–100 Hz), thus indicating the lack of syncope (Extended Data Fig. 14k,l). Thus, bradycardia (and by extension reduced cardiac output) has different effects on the brain, syncope in the case of vNAS and sleep–wake regulation in the case of the baroreflex, thereby indicating that there are distinct central circuit mechanisms at play. Taken together, our results demonstrate the specificity, sufficiency and necessity of NPY2R VSNs for the BJR. We also show that genetically and anatomically segregated vagal pathways mediate the baroreflex (PIEZO2 vagal afferents in the carotid sinus and aortic arch) and the BJR (NPY2R vagal afferents predominantly in the heart ventricles). Moreover, separate cardiac reflexes drive distinct brain states and behaviour; that is, the baroreflex is associated with sleep–wake regulation, whereas the BJR is involved in syncope.

In the past few years, there has been a resurgence in genetic dissection of the vagus nerve innervating various organs[12]. Some of these VSNs also express NPY2R according to our scRNA-seq analysis (Extended Data Fig. 15). However, to date, syncope has not been reported with optogenetic stimulation of these other organ-innervating VSNs, thus highlighting the specificity of vNAS. Finally, the most prevalent hypothesis is that vigorous contraction of an underfilled ventricle due to decreased venous return activates mechanosensory ventricular VSNs to trigger the BJR and induce syncope[14,15]. Our study provides an entry point for future research to test this hypothesis by probing whether NPY2R VSNs act as these mechanosensitive blood-volume-sensing neurons. In fact, NPY2R VSNs express putative mechanosensors[47] (Extended Data Fig. 16). In summary, we identified a genetically defined neural pathway that induces syncope, a classical problem in neurology.

## Discussion

In this study, we identified a genetically defined neural pathway at the heart–brain interface that recapitulates many behavioural phenotypes of human syncope (fainting, pupil dilation, eye-roll and loss of motor tone)[29,37]. NPY2R VSNs innervated the heart ventricular wall with both flower sprays and end-net terminals. Notably, these heart ventricle VSNs projected to the AP in addition to the canonical NTS. Stimulation of these neurons (vNAS) led to syncope and associated cardiovascular changes, including bradycardia, hypotension, cerebral hypoperfusion and reduced respiration. These physiological changes are hallmarks of the nebulous BJR, which was first described more than 150 years ago and hypothesized to be mediated by vagal afferent c-fibres predominantly located in the ventricles[2,12,13] and triggers syncope[2,11,31]. Because NPY2R VSNs are unmyelinated c-fibres[21] and their ablation markedly suppressed the BJR, our experiments confirmed both of these hypotheses. NPY2R VSNs are probably the long-sought substrates of the BJR and induce syncope.

Cerebral hypoperfusion is a well-documented observation associated with syncope in human patients. LDF measurements in our model during vNAS confirmed a substantial decrease in CBF. Asystole and large CBF drops occurred shortly after the start of photostimulation, whereas latency estimates of syncope onset based on 50% EEG power and eye-rolling behaviour occurred much later (about 6–8 s). One explanation for the delay is that estimates of syncope onset coincided with the cessation of asystole in many cases along with simultaneous cerebral blood reperfusion (Fig. 4h,k). Another explanation could be

related to total blood volume loss and the time to reach a total volume threshold, as the combination of LDF drop magnitude and duration was able to segregate most fainters and non-fainters, even under conditions of atropine (Fig. 4l).

To investigate the role of heart rate and blood pressure in syncope induction, the parasympathetic nervous system was blocked with atropine. As expected, atropine reduced vNAS-induced bradycardia and hypotension. Accordingly, we proposed that if heart activity and blood pressure were maintained throughout vNAS, the expression of syncope may also be suppressed or even blocked. However, vNAS with atropine pretreatment did not prevent syncope from occurring. Instead, it only delayed the entire sequence of events, including a blunted drop in CBF as measured by LDF (Fig. 4i–m and Extended Data Fig. 11e). Furthermore, we observed an adaptation in heart rate with vNAS (Extended Data Fig. 10d–g) and an initial decrease followed by an increase in breathing (Fig. 2f). We speculate that the animal could recover from syncope because the sensory signal is diminished over time owing to these adaptation and homeostatic effects.

Neuropixels recordings revealed instant widespread activation of brain regions following initial vNAS (about 8–60 ms; Extended Data Fig. 8c,d), even quicker than changes in CBF, followed by rapid inactivation during syncope. In particular, the hypothalamic PVZ, which includes crucial nuclei for homeostatic regulation of body physiology such as the paraventricular nucleus (PVN), had the shortest latency to activation after photostimulation (around 8 ms) and is known to receive vagal inputs[25]. In all conditions (except for PVZ excitation), syncope induction led to long-lasting suppression of locomotor and gamma band activity, which indicated that there was a sustained reduced state of arousal compared with pre-stimulus baselines. Notably, in the case of persistent PVZ excitation, mice displayed increased baseline locomotor activity and gamma band power. These mice were also insensitive to long-term drops in gamma or locomotor activity after vNAS. By contrast, PVZ inhibition led to drops in gamma band power, which was then further reduced by vNAS. Thus, bidirectional manipulation of PVZ had a push–pull effect on syncope, with inhibition leading to a longer syncope state and activation causing arousal. Conceptually, this result further suggests that there may be a circuit mechanism in the induction and maintenance of syncope apart from the widely proposed reduced CBF. The AP sends its information to the parabrachial nucleus (PBN), which in turn projects to the hypothalamus (including the PVN, the amygdala and reciprocal connections back to the AP)[25,48,49]. Furthermore, the PVN also sends feedback to the AP[50], thereby possibly creating multiple nested circuit loops that are triggered by heart-innervating VSNs. We speculate that recurrent activity between the AP, PBN and PVN could be a central hub that integrates sympathetic and parasympathetic information and then signals other regions of the brain to inactivate and induce a brief state of syncope. In summary, these new lines of evidence suggest that direct neural circuit mechanisms of syncope may coincide with CBF reduction through parasympathetic overactivation.

Finally, this study lays the groundwork for genetic, anatomical and functional dissection of other cardiovascular reflex arcs at the heart–brain interface and their influence on mental processes and behaviour. Elucidating the neurobiology of the heart is not only a crucial basic scientific endeavour but also offers substantial translational promise for cardiovascular diseases that remain the leading cause of morbidity around the world.

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

# Methods

## Animals

All procedures were done according to Institutional Animal Care and Use Committee (IACUC) guidelines at Scripps Research. Experiments were conducted on adult mice, both male and female between 1.5 and 6 months of age. No randomization or blinding was performed. Animals were arbitrarily assigned to experimental groups. Sample sizes were similar to recently published papers[52,53]. The following mouse lines were purchased from the Jackson Laboratory: C57BL/6J, stock number 000664; Ai9, stock number 007909; Ai32; stock number 012569; SLC17A6-Cre (also known as VGLUT2-Cre), stock number 016963; NPYR2R-IRES-Cre, stock number 029285; and PIEZO2-eGFP-IRES-Cre, stock number 027719. Mice were maintained in temperature-controlled (around 22–23 °C) rooms with a 12–12-h light–dark cycle (6:00–18:00 light on) and ad libitum access to chow and water.

## Reagents

The following AAVs were purchased from Addgene: AAVrg-pCAG-Flex-eGFP-WPRE (51502, $1.2 \times 10^{13}$ gene copies (GC) per ml); AAVrg-Flex-tdTomato (28306, $2.3 \times 10^{13}$ GC per ml); AAV2-hSyn-DIO-mCherry (50459, $1.8 \times 10^{13}$ GC per ml); AAV5-hSyn-hM4D(Gi)-mCherry (50475, $2.5 \times 10^{13}$ GC per ml); and AAV8-hSyn-hM3D(Gq)-mCherry (50474, $5 \times 10^{12}$ GC per ml). Other reagents used were AAV.PHP.S-DIO-sfGFP (gCOMET, gift from D. Gibbs, AAV obtained from Janelia, $6.6 \times 10^{12}$ GC per ml), AAV.PHP.S-mCherry-Flex-DTA (gift from L. Ye, plasmid obtained from Addgene, $4.72 \times 10^{13}$ GC per ml), WGA647 (wheat germ agglutinin (WGA), Alexa Fluor 647 conjugate, Thermo Fisher Scientific, W32466) and WGA488 (WGA, Alexa Fluor 488 conjugate, Thermo Fisher Scientific, W11261). Virus and WGA were mixed with Fast Green (0.05%) before peripheral injection for visualization. The following primary antibodies were used: chicken anti-GFP (ab13970, Abcam, 1:500); rabbit anti-RFP (600-401-379, Rockland, 1:500); rabbit anti-NPY2R (RA14112, Neuromics, 1:500); and Alexa Fluor 647-conjugated GFP polyclonal antibody (A-31852, Thermo Fisher Scientific, 1:200). The following secondary antibodies were all from Jackson ImmunoResearch and used at 1:500 dilution: Alexa Fluor 647 donkey anti-rabbit (711-605-152); Alexa Fluor 488 donkey anti-chicken (703-545-155); and Cy3 donkey anti-rabbit IgG (H+L) (711-165-152).

**Drug administration.** Atropine (A0132, Sigma) was dissolved at 50 mg ml⁻¹ in ethanol stock solution, then diluted to 0.5 mg ml⁻¹ working solution in filtered PBS. Atropine was administered (10 mg kg⁻¹, intraperitoneally) 15 min before recording and at least 20 min before the first laser stimulation. CNO was dissolved in filtered PBS and was administered intraperitoneally (for hM3Dq, 1 mg kg⁻¹; for hM4Di, 10 mg kg⁻¹) 15 min before recording and at least 20 min before the first laser stimulation. PE (baroreflex, Sigma), SNP (baroreflex, Sigma) or PBG (BJR, Sigma) was dissolved in PBS and injected retro-orbitally (0.1 mg ml⁻¹, 50 µl for PE and SNP, 20 µl for PBG). Buprenorphine (0.1 mg kg⁻¹) was intraperitoneally injected.

## scRNA-seq analysis

Single-cell sequencing data for nodose, jugular and petrosal ganglia were obtained from the Gene Expression Omnibus database (identifier GSE145216)[20]. The data were filtered and re-plotted using the Seurat (v.3) package. In brief, cell outliers were filtered out based on the number of expressed genes and the proportion of mitochondrial genes as standard practice for single-cell transcriptomics analysis. Filtered cells were clustered using standard methods and used to plot gene expression levels.

## Surgical procedures

Anaesthesia was induced in a chamber at 3% isoflurane and then maintained through a nose cone with 1.5–2% isoflurane on a heating pad (36–38 °C). Flunixin (2.5 mg kg⁻¹) and buprenorphine (0.1 mg kg⁻¹) was injected intraperitoneally or subcutaneously before all procedures. Ophthalmic ointment was applied to the eyes of the animal to prevent drying. After surgery, animals were allowed to recover with oxygen infusion on a heating pad until awake and placed in their home cage.

**Retrograde organ tracing.** Mice were ventilated (tidal volume 0.18–0.20 ml) with a mouse ventilator (R405, RWD Life Science). Hair was removed from the left side of the chest using Nair cream and then disinfected with a 75% ethanol pad. For the heart, a lateral incision (about 1.5 cm) was made along the intercostal space. The heart was then exposed, and virus (5 µl, 700 nl min⁻¹) was injected into the walls through a glass pipette with a nanoinjector (Nanoliter 2020 Injector, 300704, World Precision Instrument). For the lung, the left lobe of the lung was exposed through the same incision site. Virus (2 µl, 700 nl min⁻¹) was injected at multiple sites (3–4) until the entire lobe became visibly blue. For the gut (stomach, small intestine and large intestine), target organs were exposed through an abdominal incision. Virus (1.5 µl for stomach, 1 µl each for the small intestine and the large intestine, speed was 500 nl min⁻¹) was injected into target organs (dorsal and ventral sides of both glandular and non-glandular stomach, around 1.5 cm of duodenum and colon) at multiple points. Mice were euthanized 3–4 weeks after surgery for tissue collection. WGA injection into heart, lung and trachea were similar except for volume and concentration: WGA647 (2 µl, 5 mg ml⁻¹) for heart injection, and WGA488 (1 µl, 5 mg ml⁻¹) for lung or trachea injection. Mice were euthanized 3 days after surgery for sample collection.

**Nodose ganglia injection.** Mice were prepared for surgery as described above. The hair on the neck was removed and disinfected. A mid-line incision (about 1.5–2 cm) was made, and nodose ganglia on both sides were carefully exposed. A volume of 400 nl AAV.PHP.S-DIO-gCOMET or 300 nl AAV.PHP.S-mCherry-Flex-DTA/AAV2-hSyn-DIO-mCherry was injected into the nodose bilaterally at a speed of 150 nl min⁻¹. Mice were euthanized 3–6 weeks after surgery for tissue collection.

## Stereotaxic surgery

**Fibre implantation.** Mice were prepared for surgery as previously described[52]. After animals were secured in a bite bar and placed on a stereotaxic apparatus (model 942, Kopf), isoflurane was set to 1.5% to maintain anaesthesia and adjusted as needed. The scalp was cleaned with an ethanol pad and removed to expose the skull. Neck muscles were retracted using hooks, and slight incisions were made to expose the skull just above the brainstem. In some cases, a 1-mm diameter hole was drilled for an anchor screw in the right occipital portion of the skull. The skull was then balanced using a glass pipette attached to a micromanipulator (MP-285, Sutter Instruments). Vetbond glue (3M) was applied to the surface of the skull to aid dental cement adherence. Another hole was drilled above the AP. An optic fibre was then slowly lowered starting from the surface of the brain (anterior–posterior: −7.4 mm; medial–lateral: 0 mm; dorsal–ventral: −3 mm). Dental cement (Jet Set-4 and Liquid) was applied and allowed to cure. Mice were monitored and allowed to recover for at least 3 days before any experiments were conducted.

**AP injection.** All procedures were similar to AP fibre implantation. Virus (AAVrg-Flex-GFP mixed with AAV5-hSyn-hM4D(Gi)-mCherry (to mark injection site) in a 9:1 ratio, 70 nl) was injected using a nanoinjector at 100 nl min⁻¹. Coordinates for AP injection were as follows: anterior–posterior: −7.4 mm; medial–lateral: 0 mm; dorsal–ventral: −3.1 mm. Post-surgery treatment was the same as described above.

**PVZ injection.** All procedures were similar to AP injection. Inhibitory (AAV5-hSyn-hM4Di-mCherry, 500 nl) or excitatory (AAV8-hSyn-hM3Dq-mCherry, 500 nl) DREADDs were injected using a nanoinjector at 100 nl min⁻¹. The coordinates for PVZ injection were as follows: anterior–

posterior: −0.5 mm; medial–lateral: ±0.35 mm; dorsal–ventral: −4.7 mm (from surface). Post-surgery treatment was the same as described above.

**Chronic ECG implant.** Mice were anaesthetized and placed on a stereotaxic frame as described above. Hair was removed at sites of incision. Skin incisions were made on the scalp, above the right shoulder, and on the left side of the abdomen as previously described[54]. Small Teflon tubes were tunnelled between the skin and the underlying tissue from both incision sites on the body up to the head. Teflon-insulated wires (32 AWG) soldered to a 3-channel pedestal (MS333/2, P1tech) were inserted through the tubes. Exposed portions of the wire were sutured to the underlying tissue with 2–3 sutures using a 6-0 nylon monofilament (1034505, McKesson). Wound clips were used to seal the wounds. Dental cement was then applied to the head and ECG pedestal and allowed to cure. Mice were monitored and allowed to recover as described above.

**Head-fixed Neuropixels implant.** Mice were prepared for stereotaxic surgery and the scalp removed as described above. A 3D printed well was first fixed to the skull using Vetbond superglue. A 1-mm ground screw was placed in the right occipital lobe attached to silver wire soldered to a gold pin. A custom-fabricated stainless steel headpost was lowered on the right side of the head of the mouse at an angle of 30° from the horizontal surface of the skull. The entire apparatus was then cemented with dental cement. Predetermined sites for craniotomies were marked with a felt-tip pen. A thin layer of Vetbond glue was applied to the entire exposed surface of the skull for protection until craniotomies were performed. A 3D printed cap was then glued to the top of the well for protection. Mice were allowed to recover as described above.

**Chronic EEG implant.** Mice were anaesthetized and placed on a stereotaxic frame as described above. EEG implants were placed as previously described[55]. Once the mouse was anaesthetized and the skull exposed, 1-mm holes were drilled into the skull overlying the right temporal cortex (+lead) and cerebellum (−lead) using visual landmarks. Three channel electrode posts (MS333/2, P1tech) were secured to 1-mm stainless steel screws (8L003905201F, P1tech). Screws were advanced until secure, and special care was taken not to advance the screws beyond the point of stable contact with the dura. Dental cement was applied around the screws, on the base of the post and on exposed skull.

**Combined chronic ECG and EEG implant.** Combined implants were made using the above descriptions with a few modifications: two screws over the temporal (+EEG and −ECG) and cerebellum (−EEG) served as leads for EEG recordings. A third connection to a wire sutured to the left abdomen (+ECG) and the same temporal screw (+EEG and −ECG) served as leads for ECG recordings.

**Neuropixels craniotomy.** The day before recording, Neuropixels-implanted mice were placed on a stereotaxic frame, the skull balanced and 2 holes (about 1.5–2 mm) were drilled in predetermined locations on the surface of the skull. Low-melting point saline-based agarose was prepared and allowed to cool just before the gelling point, and a thin layer was applied to the surface of the skull. After the agarose had set, a layer of silicone oil was applied to the surface to prevent drying. Finally, silicone elastomer (Kwik-Cast, WPI) was applied and then the 3D printed cap was super-glued back on top of the well. The animal was then given an analgesic cocktail (dexamethasone 1.25 mg kg$^{-1}$, Rimadyl 5 mg kg$^{-1}$ and enrofloxacin 5 mg kg$^{-1}$) and allowed to recover.

**Transcranial thinned skull window.** Mice were anaesthetized and placed on a stereotaxic frame under 1–3% isoflurane. The scalp was sterilized and removed, followed by gentle but thorough removal of

the periosteum on the parietal and occipital plates. The connective tissue over the skull sutures was gently removed using a scalpel and the sutures were then covered with low-viscosity cyanoacrylate glue (Loctite, no. 4104) to provide stability between the skull plates. A 3 × 3 mm transcranial window was made over the primary somatosensory cortex (2.5 mm caudal and 3 mm lateral from bregma) by thinning the skull with an electric vibration drill (250 μm burr diameter). Once dried, the thinned bone was covered with high-viscosity cyanoacrylate glue (Loctite, no. 401) and a glass coverslip. The rest of the exposed skull was covered with dental cement (C&B metabond). Buprenorphine was injected subcutaneously for analgesia (0.1 mg kg$^{-1}$) after surgery.

## Optogenetic stimulation
A diode laser (462 nm, Shanghai Laser & Optics Century) was attached to a patch cable set to 10 mW power at the fibre tip. Pulse widths were set to 20 ms and delivered in trains of 5, 10 or 20 Hz stimulation rates. In some cases, a 5 s on/off pattern of train delivery was used.

## Behaviour
**MoSeq.** Mice were placed in a large plastic black cylinder (43 cm diameter × 36 cm tall) and were recorded from above using a 3D camera (Xbox Kinect, Microsoft) in a dark room. Mice were attached to an optical commutator above the cylindrical chamber. The experiment lasted for 5 days: 3 days of 30 min of habituation; 1 day of 45 min of baseline; and 1 day of 15 min of baseline and 30 min of laser stimulation (5 s on/off, 20 Hz). Analysis was done using the MoSeq[45] Python pipeline available through the Datta Laboratory. All videos were used (including baseline videos) to compute principal component analysis of the mouse body. The first 10 principal components extracted were able to account for 90.36% of variance. Multiple iterative models adjusting Kappa values were tested and the most optimal model was selected for syllable extraction. All extracted syllables were categorized on the basis of visual assessment of crowd videos along with average velocity measures to determine slow, mid and fast movements, as well as rearing behaviour using velocity along the z axis. Six syllables were identified in which the optical tether clearly obstructed the line of sight of the camera of the mouse and were labelled artefact. Ethograms were calculated for each mouse exhibiting syllables that were grouped into nine behavioural categories across time. The percentage of time spent exhibiting each behaviour was then calculated for baseline and stimulation periods during experimental sessions. Individual control baselines were then subtracted from stimulation for each category and then averaged to quantify mean differences for each behaviour between genotypes.

**Open-field syncope.** To measure freely moving syncope behaviour, implanted mice were attached to an optic fibre tether and commutator and placed inside a transparent acrylic cylinder (20 cm diameter, 15 cm tall). Cameras were placed directly above, below (through a transparent floor) and, in some cases, from the side. Mice were given 2 min of baseline before laser stimulation was delivered, and 5 min after laser stimulation ended. Videos were manually scored for immobility during 1 min of photostimulation (20 Hz). Latency to first immobility bout, duration of first immobility bout and the number of bouts during laser stimulation were recorded.

**Locomotion tracking.** Locomotion was tracked in the open field (above) using open-source ToxTrac[56] software. Videos were downsampled to matching resolution (1,280 × 960) and frame rate (20 f.p.s.). Cameras were automatically calibrated for distance using a chequered grid placed on the floor of the arena. Maximum and minimum size thresholds of pixel counts were adjusted to isolate the mouse body from tethers. Instantaneous speed per frame was extracted and mean smoothed over a sliding 800 sample window. Average velocities were calculated in 3-min-long windows. Pre-window = −4 min until −1 min

before stimulation onset; post-window = 1 min until 4 min after stimulation onset.

**Drinking behaviour assays.** Water restriction experiments were conducted as previously described[52]. Animals were trained for 3 days before experiments. During training sessions, animals were deprived of water for 24 h and then given 1 ml of water in behaviour chambers. Ad libitum water was provided for recovery after training. In the experimental session, animals were water-deprived for 48 h followed by 30 min (measured, in chamber) of water access.

## Histology
Mice were anaesthetized with isoflurane and perfused with 15 ml cold 1× PBS followed by 15 ml ice-cold 4% paraformaldehyde (PFA). Brains, visceral organs and nodose ganglia were dissected and fixed in 4% PFA at 4 °C overnight. Brain samples were rinsed and kept in PBS before vibratome sectioning. Samples for cryosectioning were kept in 30% sucrose solution for 24–36 h at 4 °C until they sank and were then frozen in OCT at −80 °C.

**Immunohistochemistry.** Sections (100 µm for vibratome sections, 10 µm for nodose ganglion cryosections, 100 µm for heart cryosections, 70–80 µm for lung and gut cryosections) were washed with PBS, put in blocking buffer (10% donkey serum, 0.2% Triton in 1× PBS) for 1 h at room temperature and then incubated with primary antibody (1:500 in blocking buffer) for 48 h at 4 °C. Sections were then washed 3× 15 min with 0.2% PBST (0.2% Triton in 1× PBS) and incubated with secondary antibody (1:500 in blocking buffer) for 3 h at room temperature. This was followed by 3× 15 min washes with 0.2% PBST and 30 min DAPI incubation.

**Image acquisition and analysis.** Images were acquired with either an A1 or C2 confocal microscope (Nikon) using a CFI plan apochromat lambda ×10 (NA = 0.45, WD = 4.00) or ×20 (NA = 0.45, WD = 4.00) objective. Imaging settings were optimized for each individual experimental set. Images were analysed as previously described[57] (Extended Data Figs. 1e and 2g). The following abbreviations were used for labelling regions: AI, agranular insular area; AMB, nucleus ambiguus; ARH, arcuate hypothalamic nucleus; BNST, bed nuclei of the stria terminalis; CEA, central amygdala nucleus; DMH, dorsomedial nucleus of the hypothalamus; EW, Edinger–Westphal nucleus; LH, lateral hypothalamus; MARN, magnocellular reticular nucleus; PAG, periaqueductal grey; PSTN, parasubthalamic nucleus; PVN, paraventricular hypothalamic nucleus; PVT, paraventricular nucleus of the thalamus; SNR, substantia nigra, reticular part; VTA, ventral tegmental area.

## HYBRiD clearing
Whole mouse heart, lung and gut samples were stained and cleared using a recently developed clearing protocol[4]. Samples were imaged using an Olympus FV3000 confocal microscope with a ×10, 0.6 NA, water-immersion objective (XLUMPlanFI, Olympus).

**Light-sheet imaging.** Samples were imaged with a light-sheet microscope (SmartSPIM, LifeCanvas). Samples were imaged using a ×3.6, 0.2 NA objective (LifeCanvas). Images were sampled at full resolution (2,048 × 2,048, 1.79, 1.79, 4 µm *xyz* voxel size). Supplementary videos were made using light-sheet microscopy.

## Physiology
**ECG and respiration.** ECG and respiration recordings were obtained using a BioPac MP160 system (Biopac Systems) with an ECG100C amplifier unit. The acquisition hardware was set to 1 Hz high-pass and 150 Hz low-pass filters. ECG output data were collected with gain maintained the same (5,000×) between all recordings. A piezo electric transducer was placed on the heating pad under the chest of the mouse to measure

its respiration rate. Data were sampled at a rate between 2 and 10 kHz using AcqKnowledge software. Positive and negative recording needle electrodes were inserted under the skin above the right shoulder (−lead) and on the left side of the abdomen (+lead). When necessary, a ground electrode was inserted under the skin at the base of the tail. For awake recordings, the ECG implant was connected to a commutator (SL3C/SB, P1tech). For analysis, ECG signals were bandpass-filtered between 10 and 50 Hz to aid peak detection. Positive peaks of the ECG waveforms were detected using AcqKnowledge software and converted into heart rates (bpm). Respiration rates (RRs) were extracted using custom MatLab scripts. Smoothed bpm and RR traces were calculated using a 1-s sliding window along the entire recording. Rates during each stimulation were normalized to their average 30 s to 1 min baseline before laser stimulation so that the average baseline value = 1 for each stimulation window. Heart-rate changes during laser stimulation were calculated by taking a 6–24 s time window in the centre of the laser stimulation period and RR changes using a 6–15 s window. Heart-rate changes during post-laser stimulation were calculated by taking a 30–60 s time window relative to onset of the laser stimulation period and RR changes using a 36–45 s window. For ECG recordings during adaptation with simultaneous retro-orbital drug injection, data were shifted by 10 s (for PBG) and 40 s (for PE), then subtracted from the original time-series. This process isolates heart-rate changes due to drug effects from direct effect of photostimulation. In cases when the raw data were outside the parameters of standardized scripts, the data were manually curated and analysed.

**Region-specific optogenetic stimulation of the vagus nerve.** NPY2R–ChR2 or PIEZO2–ChR2 mice were anaesthetized by 2% isoflurane and placed on a heating pad. The carotid sinus and vagus nerve were surgically exposed. An optic fibre (200 µm core, NA = 0.22, RWD) was placed above the following regions: carotid sinus, superior laryngeal nerve, vagal nerve trunk and bilateral abdominal branches of the vagus nerve. Surrounding areas were carefully covered while stimulating each individual region (20 ms pulses, 10 mW intensity, 20 Hz, 30 s). An average of a 3 s time window around the minimum heart rate was used for quantification. The RR was calculated as described above.

**EEG recording and fast-Fourier transform.** EEGs were acquired in the same manner as for awake ECGs described above, but instead using screw electrode leads in the right temporal cortex (+lead) and left cerebellum/occipital cortex (−lead). Fast-Fourier transform (FFT) analysis was conducted in MatLab after applying a 59–61 Hz notch filter to eliminate line noise. FFTs were calculated in 2 s segments over the total analysis window, and power was further averaged into standard frequency bins: delta (1–4 Hz), theta (4–8 Hz), alpha (8–13 Hz), beta (13–30 Hz), low gamma (30–59 Hz) and high gamma (61–120 Hz).

**Blood-pressure recording.** Blood pressure was recorded in 1.5% isoflurane-anaesthetized animals on a warmed plate using a tail cuff (Kent Scientific). Blood-pressure measurements were obtained every 1 min. Data were normalized to the first 5 min of baseline. Blood pressure change during photostimulation was calculated by averaging the first 3 min.

**Internal temperature recording.** Anaesthetized animals were put on pad and their internal temperature was monitored using a rectal probe (ThermoStar, RWD). The baseline temperature was recorded for 1 min, followed by photostimulation for 1 min and a 1 min post-stimulus period. The internal temperature was recorded in the centre of each time interval.

**Ultrasonography.** Animals were anaesthetized with 3–4% isoflurane for induction then maintained at 2% isoflurane. An optic cable was connected for photostimulation. After the left side of the chest was

shaved, animals were placed on a heating platform without any disruption for about 10 min. Images of the left ventricle (B-mode, Pslax view) and carotid arch (B-mode and Doppler mode) were acquired using a Vevo 3100 micro-ultrasound imaging system with a MX550D linear array transducer (VisualSonics). Data were analysed using Vevo Lab (v.5.5.0) software. Left ventricle functional parameters were analysed using end-diastolic and end-systolic measurements in B-mode. For pre-stimulation baseline recording, frames without respiration within a consistent 30 s window were chosen for analysis. For recording during stimulation, frames around the time of minimal heart rate were chosen. The same frame selection guidelines were followed for Doppler mode analysis and three sequential peaks from each recording session were chosen for analysis. The following parameters were measured: ENDOmajr;d (B-mode): endocardial major in diastole; ENDOmajr;s (B-mode): endocardial major in systole; EPImajr;d (B-mode): epicardial major in diastole; EPImajr;s (B-mode): epicardial major in systole; IVS;d (B-mode): inter ventricular septum in diastole; IVS;s (B-mode): inter ventricular septum in systole; LVID;d (B-mode): left ventricular internal diameter in diastole; LVID;s (B-mode): left ventricular internal diameter in systole; LVPW;d (B-mode): left ventricular posterior wall in diastole; LVPW;s (B-mode): left ventricular posterior wall in systole; AAT (PW Doppler mode): aortic acceleration time; AET (PW Doppler mode): aortic ejection time; AoV Diam (B-mode): ascending aorta diameter; AV Peak Vel (PW Doppler mode): aortic valve peak velocity.

**LDF in awake mice.** Non-invasive LDF (Moor instruments) was used to measure CBF over the contralateral temporal cortex through a reinforced thinned skull window. The probe was held over the transcranial thinned skull window and data were sampled at 10 kHz using a BiopacM160.

**LDF signal processing.** Raw LDF signals were processed using custom MatLab scripts. All the trials were first normalized to a baseline of 30 s before stimulation, followed by median filtering over a 500 ms window. For plotting the haemodynamics relative to the drop in EEG power during stimulation, the trials were averaged with respect to the syncope onset (defined as the time at which 50% EEG power drop is observed). For inter-group comparison (5 Hz, 10 Hz, 20 Hz vehicle and 20 Hz atropine), we calculated the following parameters from the LDF signal across all animals as described below.

(1) Latency to 50% drop in LDF, calculated as the time to decrease to 50% of baseline LDF. (2) Minimum LDF during stimulation, calculated as the minimum LDF during a stimulation duration of 30 s. (3) A 50% drop in LDF transit time, with the duration calculated as the full width at half-minima of LDF. (4) Late-phase mean LDF, calculated as the average LDF over a 100 s window before the end of a trial. (5) Post-stimulus LDF recovery duration, calculated as the time taken to recover to baseline after a 30 s stimulus window. (6) Rate coefficient, calculated as the ratio of rate of increase of LDF (positive slope) to rate of decrease of LDF (negative slope) during the 30 s stimulation window.

## Neuropixels

**Habituation.** The habituation protocol took place over 7 days. On day 1, mice were handled by an experimenter for 5 min in the morning and another 10 min session in the afternoon, during which mice were allowed to explore and move freely on the treadmill and head fixation setup. On successive days, mice were head-fixed on the treadmill for increasing amounts of time (up to 2 h) until the day of Neuropixels recordings.

**Targeting brain regions.** Before surgery, a freely available MatLab toolbox (allenCCF and SHARPtrack)[51,58] was used to plan probe insertion angles through desired brain regions. After the probe position in the virtual Allen brain atlas trajectory planner was satisfactory,

probe angles, depth of insertion and brain surface entry coordinates were recorded. Surface entry points were marked on the skulls of implanted mice before craniotomy, and the micromanipulator system (MPM, Newscale) that holds the Neuropixels probes was adjusted to match the planned probe insertions.

**Probe preparation.** On the day of recording, lipophilic dyes (DiI, DiO or DiD) (V22889, Invitrogen) were manually applied to the shank of the Neuropixels probes to be recorded by using a droplet at the tip of a pipette under a dissecting microscope and allowed to dry. A different dye was used for each day of recording to keep track of probe recording sites across multiple days. Penetrations were spatially far enough apart such that dye tracks using the same dye could be identified anatomically. After recording, probe shanks were soaked in a 1% tergazyme (Alconox) solution for at least 3 h to remove debris and dye, followed by soaking in DI water overnight.

**Probe insertion.** Neuropixels probes were lowered until just above the silicone oil and slowly advanced in the *z*-direction while monitoring channel activity. Once the probe made contact with the surface of the brain and visible spikes were detected at the tip, the probe was inserted at a rate of 200 μm min⁻¹ until the targeted depth was reached. Once the final depth was reached, the probe was retracted by 100 μm and allowed to settle for 15 min before recording. At the conclusion of the first recording session, probes were retracted, and a silicone elastomer (Kwik-Cast, WPI) was applied to the entire surface of the silicone oil–agarose and allowed to dry. A 3D printed cap was then super glued to the top of the well. The mouse was then returned to its home cage for subsequent recordings on subsequent days. On the final day of recording, the mouse was euthanized and the brain collected and prepared for histological analysis.

**Neuropixels recording.** Data were collected using SpikeGLX software specifically designed to acquire multi-channel data from Neuropixels probes. Data from the NP action potential bands (0.3–10 kHz) were sampled at 30 kHz. LFP bands (0.5–500 Hz) were sampled at 2.5 kHz. Digital and analog inputs were sampled at about 12.5 kHz. Sampling frequencies were slightly modified based on signal calibration and synchronization from individual NP head-stage parameters using SpikeGLX calibration during setup.

**Video acquisition.** During head fixation, two main cameras were used (BFS-U3-13Y3M-C, FLIR): one with a zoom lens for pupillometry (×3.3 macro zoom lens no. 56-524, Edmund Optics) and another for recording facial movements (HF25XA-5M, Fujinon). Cameras were directly linked for synchronous acquisition using GPIO cables connected to each camera and TTL trigger pulses generated by one of the cameras. Trigger pulses generated by the master camera (30 f.p.s.) were also simultaneously recorded to a digital input channel. Videos were recorded using SpinView software, available through the Spinnaker SDK (FLIR) package. Three infrared LED arrays (B075F7NV56, Univivi) were setup at corners of the head-fixed recording chambers, and illumination was adjusted so that the video images were not oversaturated[59].

**Probe insertion histology.** Probes were registered using freely available SHARPtrack software[51]. In brief, images from each slice were preprocessed, aligned and transformed according to the Allen Mouse Brain Common Coordinate Framework. Individual probe tracks were then manually marked along all slices and reconstructed in 3D. The final region output from SHARPtrack was then saved and matched with electrophysiological data collected at electrode site depths. The following list are region labels used from the Allen Mouse Brain Common Coordinate Framework: ACA (anterior cingulate area): ACAd1, ACAd2/3, ACAd5, ADAd6a, ACAv1, ACAv2/3, ACAv5, ACAv6a, ACAv6b and ACAd6a; AI (agranular insular area): AId6a, AIp6a, AIp6b, AIv6a,

CLA and GU6a; BLA (basolateral amygdalar nucleus): BLAa, BLAp and CTXsp; BMA (basomedial amygdalar nucleus): BMAa and BMAp; BST (bed nuclei of the stria terminalis): BST; DORpm (thalamus, polymodal association cortex related): AD, AMd, AMv, AV, CL, CM, Eth, IAM, IMD, LD, LH, LP, MD, MH, PCN, PF, PO, PT, PVT, RE, RH, RT, SMT, SubG, TH, Xi, IAD and SGN; DORsm (thalamus, sensory-motor cortex related): LGd-co, LGd-ip, LGd-sh, LGv, VAL, VM, VPL, VPLpc, VPM, VPMpc, PoT and SPFp; HPC (hippocampal formation): CA1, CA2, CA3, DG-mo, DG-po, DG-sg, HPF, ProS, SUB and IG; ILA (infralimbic area): ILA1, ILA2/3, ILA5, ILA6a and ILA6b; LA (lateral amygdalar nucleus): LA; LS (lateral septal nucleus): LSc, LSr, LSv and SF; LZ (hypothalamic lateral zone): FF, HY, LHA, LPO, PSTN and ZI; MBmot (midbrain, motor related): APN, MB, MRN, MT, NOT, PAG, PPT, PR, RN, RR, SNr and VTA; MEZ (hypothalamic medial zone): MPN, PH, PMv, PMd, TMv, TU, VMH and AHN; OLF (olfactory areas): AON, COApl, COApm, DP, EPd, EPv, NLOT3, OLF, PAA, TTd and TR; PALd (pallidum, dorsal region): GPe, GPi and PAL; PALm (pallidum, medial region): MS, NDB and TRS; PIR (piriform area): PIR; PL (prelimbic area): PL1, PL2/3, PL5, PL6a and PL6b; PVZ (periventricular zone): PVH, PVHd and PVi; SSp (primary somatosensory area): SSp-n4, SSp-n5, SSp-n6a, SSp-ul6a, SSP-ul6b, SSp-bfd6a, SSp-bfd6b, SSp-ll2/3, SSp-ll4, SSp-ll5, SSp-ll6a, SSp-ll6b, SSp-n1, SSp-n2/3, SSp-tr1, SSp-ul1, SSp-ul2/3, SSp-ul4, SSp-ul5, SSp-ul6b, SSp-un2/3, SSp-un4, SSp-un5, SSp-un6a and SSp-un6b; SSs (supplemental somatosensory area): SSs5, SSs6a and SSs6b; STRd (striatum dorsal region): CP, FS and STR; STRv (striatum ventral region): ACB and OT; VISC (visceral area): VISC6a and VISC6b; PTLp (posterior parietal association areas): VISa1, VISa2/3, VISa4, VISa5, VISa6a and VISa6b.

**LFP or EEG wavelet analyses.** Raw LFPs were first averaged across all channels of a Neuropixels probe into a single LFP. EEGs were calculated from a single channel over the temporal cortex. Using custom MatLab scripts, a 59–61 Hz notch filter was first applied to the data to remove any residual 60 Hz line noise. LFP or EEG time-series data during windows that included baseline, laser stimulation and post-laser periods were then converted into power values using 80 logarithmically spaced Morse wavelets from 0.5 Hz to 120 Hz using functions provided in the MatLab wavelet toolbox. Power was then normalized for each frequency by dividing power across the entire window by the average power during pre-stimulus baseline. To determine significant changes in power in the time × frequency domain owing to light stimulation, a Monte Carlo permutation approach was utilized, which shuffles trials of light and no-light conditions to create a null-distribution of the size (area) of contiguous time × frequency clusters that are due to chance and comparing them to the size of clusters from actual light conditions. If the size of clusters from the actual light conditions were greater at $P = 0.025$ on either tail (positive or negative) of the null distribution, this was considered a significant result due to light stimulation and the direction of change was noted[60]. Because significant drops in LFP or EEG power were found between 8 and 100 Hz, we used a 50% drop in average power (after smoothening) across this frequency band to determine an estimate for latency to syncope onset. When average power returned to above 80%, this was considered an estimated end of the syncope bout.

**Spike sorting.** Each day after recording was completed, raw data from the NP action potential channels were run through the spike sorting software Kilosort3 (ref. 61). This separates 'good' isolated units from 'multi-unit activity'. After spike clusters were identified, the quality of automatic unit separation was manually inspected using Phy software. Kilosort3 performs automated drift correction of the electrode. We still observed some units that were more active at the beginning or the end of the recording, which suggesting there was some remaining drift. To determine drifting units, each unit's firing rate in 100-ms bins was smoothed by a Gaussian of width 30 s. If this smoothed firing rate varied by more than a factor of 5 during the recording, then the unit was

defined to be drifting and removed from further analyses. This removed on average 5.7% of units across recordings (range of 1 to 15%). Units with a firing rate less than 0.25 Hz were also excluded from further analyses.

**Video processing.** We used open-source software FaceMap[38,62] and DeepLabCut[63]. For FaceMap, three regions of interest (ROIs) were chosen: around the eye, the entire face and the whisker pad. A saturation threshold was set on the eye ROI such that only pixels in the eye were darker than the threshold. Blinks were determined to be times when the eye area went below a certain number of pixels, which was set to the same value across mice. From the ROIs around the face and the whisker pad, the single value decomposition (SVDs) of the absolute motion energy and the raw frames were computed. The raw movie was a matrix of size $N_{pixels}$ by $N_{time points}$. The absolute motion energy at each time was computed as the absolute value of the difference between consecutive frames. Because the matrices were too large to decompose in their raw form, we performed the SVD in segments on both the raw movie and the motion energy[38]. From this, we obtained a matrix $\mathbf{X}_{movie}$ and $\mathbf{X}_{motion}$, each of size 250 by $N_{time points}$, where 250 is the number of top SVD components used. The whisking (Extended Data Fig. 7c–f) was defined as the L2 norm of the top 10 components of the whisker $\mathbf{X}_{motion}$ at each time point. The whisking baseline for laser onset was computed for each laser stimulation using the mean across 30 s before laser onset. The whisking baseline for syncope onset was computed using the mean across −6 s to −4 s relative to syncope onset. The whisking was normalized by this baseline and averaged across mice and sessions (Extended Data Fig. 7c,e) and summarized using the mean across 5 s post-laser onset (Extended Data Fig. 7d) and across 2 s post-syncope (Extended Data Fig. 7f). To determine pupil area, a DeepLabCut model was trained with 140 frames using 20–40 frames per mouse to track 4 key points (top, bottom, left and right) around the pupil of the mouse. The $x$ and $y$ coordinates of the key points were median filtered with a window of 5 time points, corresponding to about 160 ms. The lengths of the two axes of the pupil were computed using the four key points and the pupil area was approximated as the area of an ellipse computed with the lengths of those two axes. Frames for which the likelihood estimated by DeepLabCut was below 0.75 were set to be undefined, along with frames for which the mouse was blinking, as defined above. The pupil area baseline was computed for each laser stimulation using the mean across 30 s before laser onset. The pupil area was normalized by this baseline and averaged across mice and sessions (Extended Data Fig. 7a, b).

**Neural prediction from behaviour.** We built a model to predict neural activity from the behaviour of the face. The face behaviour used for the model were the movie and motion SVDs ($\mathbf{X}_{movie}$ and $\mathbf{X}_{motion}$, respectively), which were collected at 30 Hz. The neural activity was binned in 100 ms bins (10 Hz). The behaviour and neural activity were first split into train, validation and test sets consisting of 50, 50 and 63 s segments, respectively, to avoid contamination of the test set with the train set owing to the autocorrelation timescale of behaviour. One test segment (the control period) consisted of a period of 63 s without laser stimulation, to quantify the performance of the model without laser stimulation, and the other test segments were during the laser stimulation, in a window −3 s to 60 s at laser stimulation onset. The rest of the data were divided into segments of length 50 s, and 80% were placed in the train set and 20% in the validation set. Thus, the training and validation sets did not contain any of the laser stimulation periods as we aimed to quantify the discrepancies of the model during stimulation. Next, a multi-layer network model was fit to predict neural activity from behaviour. The model consisted of a linear layer, a one-dimensional convolutional layer (temporal convolution), a ReLU non-linearity, a linear layer with smaller dimensionality (the output of this is defined as the latents) and then a linear output layer. The output of the convolutional layer was subsampled at the time points of neural activity because the behaviour

was collected at 30 Hz whereas the neural activity was binned at 10 Hz. This model was fit on the training data using the optimizer AdamW, and the validation data were used for early stopping of the model fitting[64,65]. The model was then applied to the test behaviour segments. This produced a prediction of the neural activity and latents of the model. The latents were then used to find time points in the training set that most resembled the test set latents to create a nearest neighbour prediction: the top 50 neighbours of each test latent from the training set were averaged to compute the test set prediction. This nearest neighbour prediction approach outperformed the direct prediction method from behaviour, with a median variance explained across neurons of 10.3% compared with 5.0%. The residual between the neural activity and the behaviour prediction was defined as the neural activity subtracted by the nearest neighbour prediction (Fig. 3f,g and Extended Data Fig. 8e, f).

The average correlation between the model and neural activity across all brain regions, binned in 1 s time bins, was $r = 0.75$ in the control period, whereas the average correlation was $r = 0.44$ during the 20 Hz, 30 s of laser stimulus. For brain area PVZ, it was $r = 0.91$ for the control, and $r = 0.36$ for the 20 Hz, 30 s of stimulation period (Extended Data Fig. 8b, bottom).

**Syncope quantification.** Syncope onset was defined using the EEG–LFP for the 20 Hz, 30 s of laser stimulation (see section 'LFP–EEG wavelet analyses'). We defined a neuron to be inactive at syncope onset if it had a period of inactivity starting within 250 ms of the syncope onset. The period of inactivity had to be sufficiently long such that, according to its firing rate, it had a random probability of occurring of less than 1%. Assuming a Poisson distribution of spiking for a neuron, the probability of no spikes in a given number of time bins $k$ is $P = e^{-\lambda k \, dt}$, where $\lambda$ is the average firing rate of the neuron excluding the test set as defined above, and where $dt$ is the time bin size, which in this case is 1 ms. Setting this probability $P$ to 1%, the time period corresponds to $k = -\frac{\ln(0.01)}{\lambda \, dt}$ — time bins of inactivity. The maximum number of time bins used for the inactivity definition (for lower firing units) was 4,000 time bins (4 s). The number of inactive neurons was also computed for a random time window of 250 ms. The percentage of active neurons per brain area for the control time window and for syncope onset were compared using a paired $t$-test (Extended Data Fig. 8b, top). For inactive neurons, time off during the total syncope time window was defined as the percentage of time bins the neuron was inactive, where the time bins were defined as above (Extended Data Fig. 8b). Time off was not computed for active neurons.

**Laser response quantification.** For each mouse and each session, there were at least four laser stimulation periods (at frequencies of 5 Hz, 10 Hz and 2 × 20 Hz). To identify laser-activated neurons, the laser onset period was defined as the first 800 ms following laser stimulation onset, and the pre-laser period was defined as the 800 ms preceding laser stimulation onset; the residual from the behaviour prediction was averaged within each of these periods for each neuron. A neuron was defined as activated by the laser if it fulfilled three criteria: (1) the behaviour prediction residual of the neuron during the laser onset period averaged across the four stimulation periods was positive; (2) the residual of the neuron during the laser onset period exceeded the residual during the pre-laser period for all four laser stimulations; and (3) the neuron responded to the laser with a latency of less than 250 ms (see latency definition below; Extended Data Fig. 8c,d). The first two criteria ensured that the neuron responded to the laser rather than to behaviours induced by the laser. The last criterion focused on neurons that respond quickly and therefore the activities of these neurons were more likely to be related directly to the laser onset, although this criterion did not substantially decrease the number of neurons defined as activated neurons: 91% of neurons that satisfied the first two criteria satisfied the last criterion. The latencies of the responses of the neurons to the laser stimulation were computed using

equation 1 from a previous paper[66]. Assuming a Poisson distribution of spiking for a neuron, the probability of observing at least $n$ spikes in a window $t$ is:

$$P_t(\geq n) = 1 - \sum_{m=0}^{n-1} \frac{(N\lambda t)^m e^{-N\lambda t}}{m!} \tag{1}$$

where $\lambda$ is the firing rate of the neuron excluding the test set as described above, $N$ is the number of repetitions (in this case four laser onsets), and the time windows $t$ started from the time of laser onset and were increased with a step size of 1 ms. The latency to spike was defined as the time when this probability is reduced below $1 \times 10^{-6}$; that is, when the response observed is unlikely to occur by chance from the spontaneous firing rate of the neuron[66]. The latency per brain area was defined as the median of the latencies of the neurons in the region (Extended Data Fig. 8d).

## Statistics

All statistical analyses were carried out using Prism (GraphPad). All tests used are reported in the figure legends. Data sheets with the analyses of statistical tests from Prism reporting estimates of variance within each group, comparison of variances across groups are available upon request. No sample size statistics, blinding or randomization procedures were performed. Representative data were based on at least three independent observations. Animals with proper viral expression or implant placement were included in the analyses. These criteria were pre-established.

## Reporting summary

Further information on research design is available in the Nature Portfolio Reporting Summary linked to this article.

## Data availability

The single-cell sequencing data were obtained from the Gene Expression Omnibus database (identifier GSE145216). Source data are provided with this paper. All other data are large and available from the corresponding author upon reasonable request.

## Code availability

Code has been uploaded to GitHub and is available at https://github.com/AugustineLab/Syncope.

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

**Acknowledgements** We thank G. Lippi and N. Zolboot for sharing VGLUT2-Cre mice; A. Patapoutian and Y. Oka for comments on an initial draft; M. Pachitariu for advice on Neuropixels data analysis; H. Cline, E. Topol and L. Stowers for resources; Y. Wang, M. R. Servín Vences and M. Loud for viruses; J. Goins, S. Leutgeb and T. Komiyama for help with setting up Neuropixels recordings; V. Nudell for tissue clearing guidance; K. Marshall for advice on image analysis; K. Spencer and staff at the Scripps Department of Animal Resources for support; D. Gibbs for the gift of AAV.PHP.S-DIO-sfGFP (gCOMET); and Y. Yao and Y. Dan for sharing EEG data. This work was supported by funds from Scripps Research, the Helen Dorris Foundation, the University of California San Diego, NIH 5UL1TR002550-03,the American Heart Association Early Faculty Independence Award and the Mallinckrodt Foundation grant to V.A., R35 NS097265 and U19 NS123717 to D.K., a Dorris Scholarship to J.W.L. and S.Y., a Dorris-Skaggs Fellowship and a Shurl and Kay Curci Fellowship to J.M., and a Merck Fellowship of the Damon Runyon Cancer Research Foundation to Y.Z.

**Author contributions** J.W.L., J.M. and V.A. conceived the research programme and designed experiments. J.W.L. and J.M. carried out the experiments and analysed data with help from S.Y., H.S., R.S., Y.Z., T.B., S.T., Y.L. and V.A. C.S. performed Neuropixels data analysis and behavioural modelling. K.C. and D.K. assisted with the LDF data collection and analysis. Z.P., T.Q. and L.Y. helped with tissue clearing data collection. T.V. and G.R. helped with Neuropixels setup. J.W.L., J.M. and V.A. wrote the paper. V.A. supervised the entire work.

**Competing interests** The authors declare no competing interests.

**Additional information**

**Correspondence and requests for materials** should be addressed to Vineet Augustine.

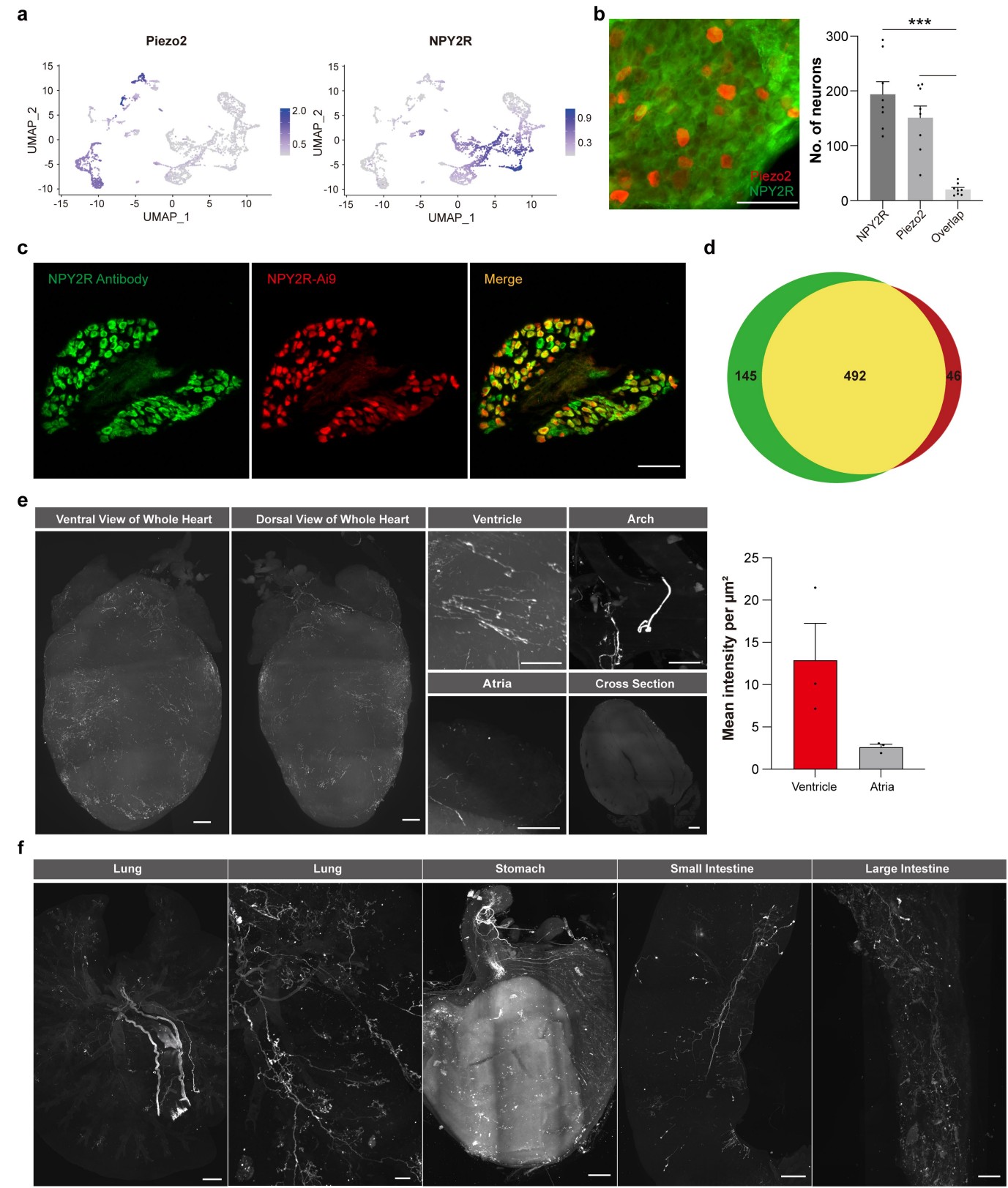

**Extended Data Fig. 1** | See next page for caption.

**Extended Data Fig. 1 | Projection patterns of NPY2R vagal sensory neurons (VSNs) across multiple organs. a**, Single cell RNA sequencing of nodose ganglia shows separation of NPY2R and Piezo2 expression (reanalysis from previous report[20]). **b**, Immunohistochemistry of nodose ganglia from PIEZO2-Cre mice infected by AAV2-DIO-mCherry showed minimal overlap between NPY2R and Piezo2 VSNs (left). Quantification of overlap (right, n = 8 nodose from 4 mice, NPY2R/overlap p = 0.0005, PIEZO2/overlap p = 0.0008). **c**, Immunohistochemistry of nodose ganglia from NPY2R/Ai9 mice showed transgenic tdTomato highly overlapped with endogenous NPY2R expression (n = 3). **d**, Quantification of overlap (77.45 ± 0.39%, n = 3). **e**, Light sheet images of a cleared whole heart (HYBRiD) with NPY2R VSNs transduced with AAV-PHP.S-DIO-gCOMET, dorsal and ventral views (left). Distribution of NPY2R VSN terminals across different heart regions (ventricles, atria, aortic arch). No labeling was observed in the interventricular septum (cross section). In the heart, NPY2R VSNs predominantly innervated the ventricular wall (n = 3). **f**, Light sheet images of cleared lung, stomach, small and large intestines showing NPY2R VSN innervation. p < 0.001\*\*\*, by one-way repeated measures ANOVA with Geisser-Greenhouse correction and Tukey multiple comparisons. All error bars and shaded areas show mean ± s.e.m. Scale bars, 100 μm (**b**, **c**) 500 μm (**e**, **f**).

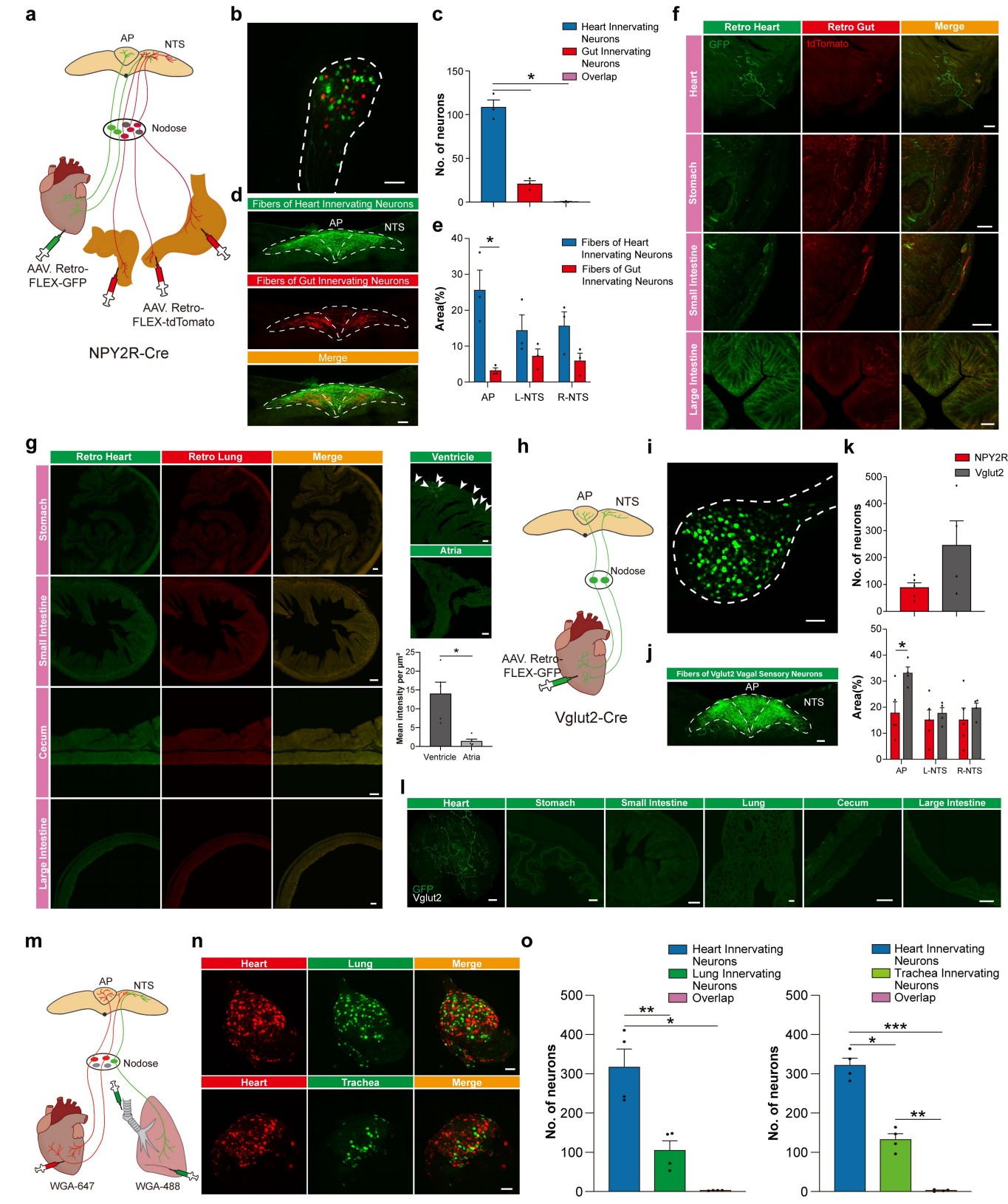

**Extended Data Fig. 2 |** See next page for caption.

**Extended Data Fig. 2 | One-to-one map of vagal sensory neurons (VSNs).**
**a**, Schematic of dual retrograde tracing of NPY2R VSNs from heart ventricles and gut (stomach, small and large intestine). **b**, Retrogradely labeled NPY2R VSNs from the heart ventricles (green), or gut (red, n = 3), in the nodose. **c**, Quantification of overlap. Nearly all of the heart and gut innervating neurons are non-overlapping, indicating that specific NPY2R VSNs project to the heart and gut separately (n = 3, heart/overlap p = 0.0112, heart/gut p = 0.0115). **d**, Spatial projection pattern of heart ventricles and gut innervating NPY2R VSNs in the AP and NTS (n = 3). **e**, Fiber density from heart ventricles or gut innervating NPY2R VSNs in the AP and NTS. The majority of fibers in the AP originate from heart ventricle innervating NPY2R VSNs (n = 3, p = 0.0102). **f**, NPY2R VSN terminals in retro-labeled heart ventricles and gut. Retro-labeled heart ventricle or gut fibers were only observed within heart ventricle or gut respectively (n = 3). **g**, NPY2R VSN terminals were not observed in stomach, small intestine, cecum, and large intestine after heart ventricle/lung dual retrograde tracing (n = 5). NPY2R VSN terminals were predominantly observed in the ventricle after atria/ventricle dual retrograde tracing (right, n = 5, p = 0.0145). **h**, Schematic of retrograde labeling (AAVrg-FLEX-GFP) of Vglut2 VSNs from the heart ventricles. Vglut2 is expressed by all VSNs. **i**, Retrogradely GFP labeled Vglut2 VSNs from the heart ventricles in the nodose ganglia (n = 4). **j**, Fiber distribution of retrogradely labeled GFP expressing Vglut2 VSNs from the heart ventricles in the AP and NTS (n = 4). **k**, Comparison of retrogradely labeled VSNs from the heart ventricles in Vglut2-Cre and NPY2R-Cre animals (top). Quantification of fibers from the heart ventricles innervating Vglut2 or NPY2R VSNs in the AP and NTS (bottom, n = 4 for Vglut2, n = 5 for NPY2R, p = 0.0225). **l**, Retrogradely labeled Vglut2 VSN terminals from the heart ventricles. Terminals were not observed in the lung, stomach, small and large intestine, or cecum (n = 4). **m**, Schematic of retrograde tracing of VSNs with wheat germ agglutinin (WGA) from heart ventricles (WGA 647), lung or trachea (WGA 488, n = 4). **n**, Heart ventricle (red), lung or trachea (green) innervating VSNs in the nodose ganglia after retrograde labeling (n = 4). **o**, Quantification of overlap after dual retro labeling. Similar to retrograde AAV tracing in NPY2R-Cre mice (Fig. 1e–g), most of the WGA488 and WGA647 labeled VSNs did not overlap (n = 4, heart/lung p = 0.0056, heart/overlap p = 0.0134; heart/trachea p = 0.0110, heart/overlap p = 0.0008, trachea/overlap p = 0.0068). p < 0.05* by one-way repeated measures ANOVA with Geisser-Greenhouse correction and Tukey multiple comparisons, two-way repeated measures ANOVA with Šidák multiple comparisons or two-way ANOVA with Šidák multiple comparisons. All error bars show mean ± s.e.m. Scale bars, 100 μm.

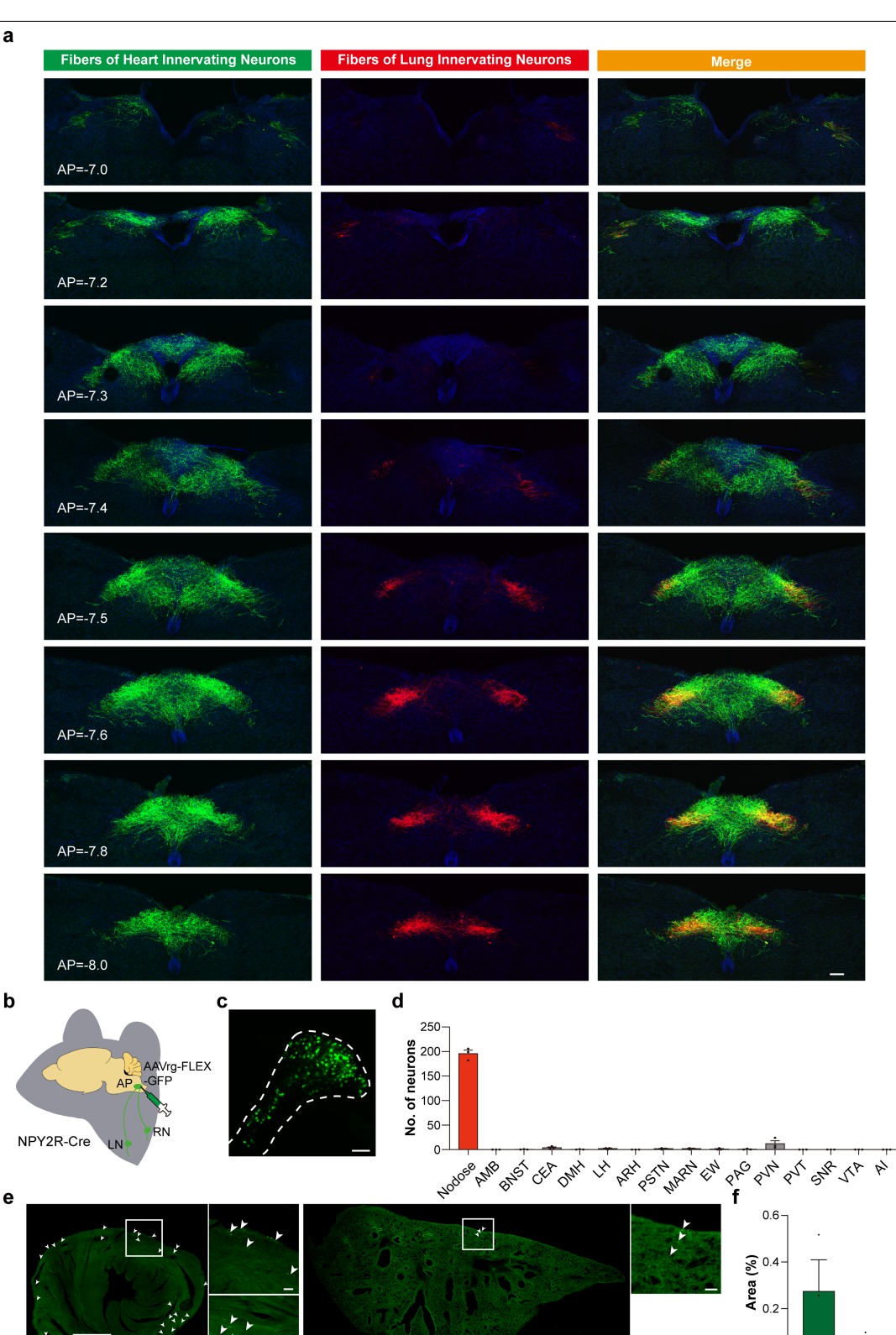

**Extended Data Fig. 3 | Organ specific spatial mapping of NPY2R vagal sensory neuron (VSN) fiber projections to the brainstem. a**, Brainstem images showing projection patterns of retrogradely labeled heart ventricle (green) or lung (red) innervating NPY2R VSNs with indicated bregma coordinates. Lung fibers were more prominent in caudal sections. Area postrema (AP) is predominantly labeled by heart ventricle projecting VSNs (n = 5) **b**, Schematic of retrograde tracing from AP in NPY2R-Cre mice. **c**, Nodose ganglia with retrogradely labeled NPY2R neurons projecting to the AP. **d**, Quantification of retrogradely labeled NPY2R neurons in nodose and the brain. Almost all NPY2R projections to the AP were from the nodose ganglia (n = 3). **e**, NPY2R VSN terminals in the heart ventricles and lung after retro-labeling from the AP. **f**, Quantifications of fiber density of NPY2R VSNs in heart ventricles and lung after retro-labeling from the AP (n = 3). All error bars show mean ± s.e.m., Scale bars, 100 μm.

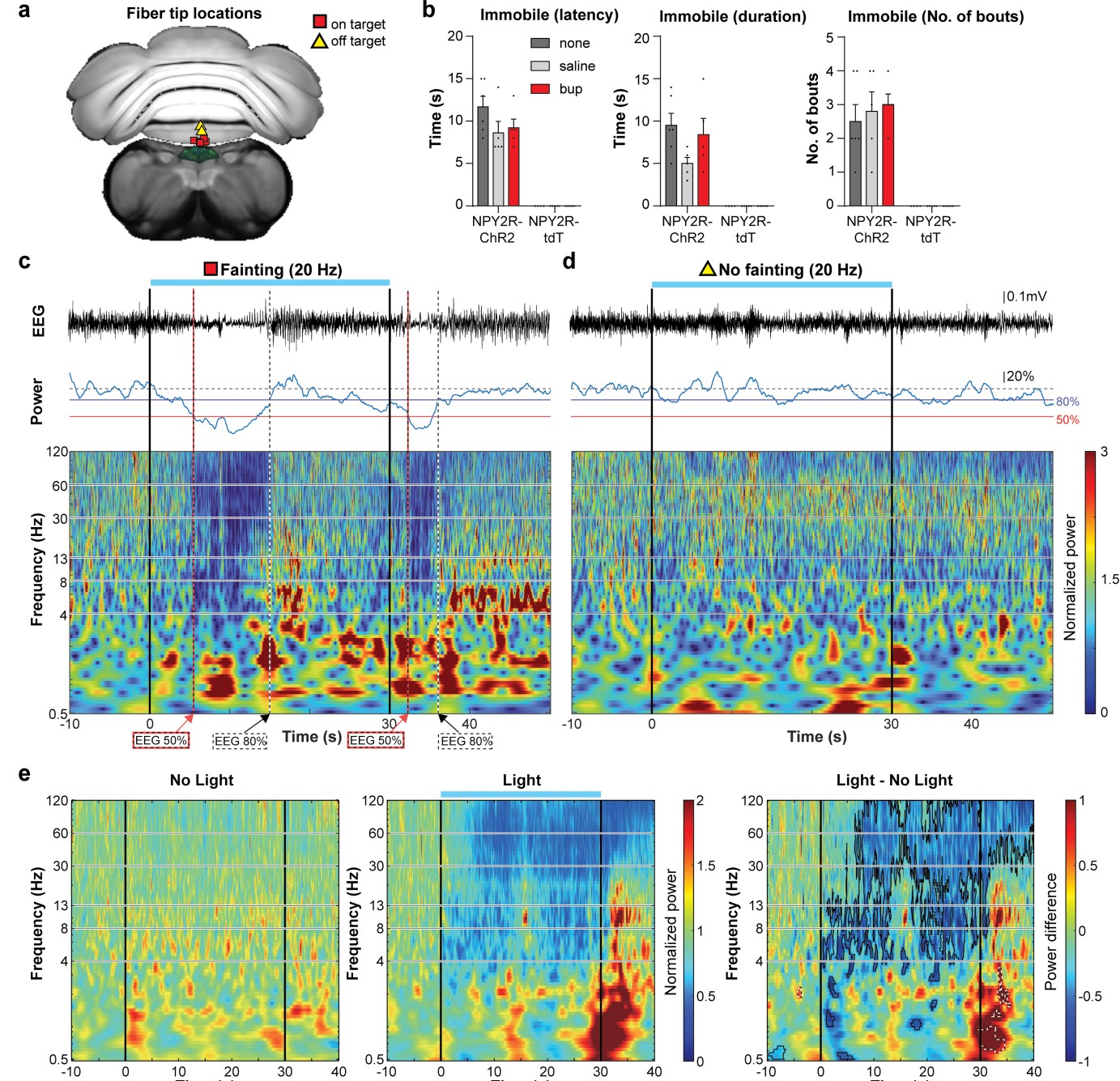

**Extended Data Fig. 4 | Syncope is associated with a drop in EEG power.**
**a**, Postmortem optic fiber tip locations targeted above the area postrema (AP) labeled on the corresponding (−7.4 mm AP) Allen Mouse Brain Common Coordinate Framework (CCF)[58]. Off-target fiber implants (yellow triangles) did not evoke behavioral phenotypes with photostimulation. Proper fiber placement is indicated by red squares. **b**, Quantification of behavior from video assessment. Buprenorphine was administered prior to vagal NPY2R to AP photostimulation to rule out a pain response and controlled by saline or no injection. Latency to first immobility (left), duration (middle), and number of bouts (right, n = 5 except for none/ChR2 = 6, none/tdTomato = 4). **c**, Single trial EEG analysis with on-target vagal NPY2R to AP stimulation (20 Hz) during freely moving recording. Raw EEG trace from temporal cortex is shown on top, average

power between 8–100 Hz is shown in the middle with 50% and 80% thresholds, and the Morse wavelet power transformation is shown below as a heatmap. The latency to when baseline normalized average power between 8–100 Hz dropped below 50% is indicated by dashed red/black lines, the latency to when average power returned to 80% is marked by dashed white/black lines. The period between is considered one "bout". **d**, Single trial EEG analysis of a mouse with off-target fiber placement (yellow triangle in "**a**"). Note, the average power trace does not cross the 50% threshold. **e**, Photostimulation produced a delayed, but stark power drop in the alpha-gamma range. The subtraction plot on the right is the same as Fig. 1m. This drop in the alpha-gamma range can be used as biomarker for syncope (n = 12 sessions across 8 mice). All error bars show mean ± s.e.m.

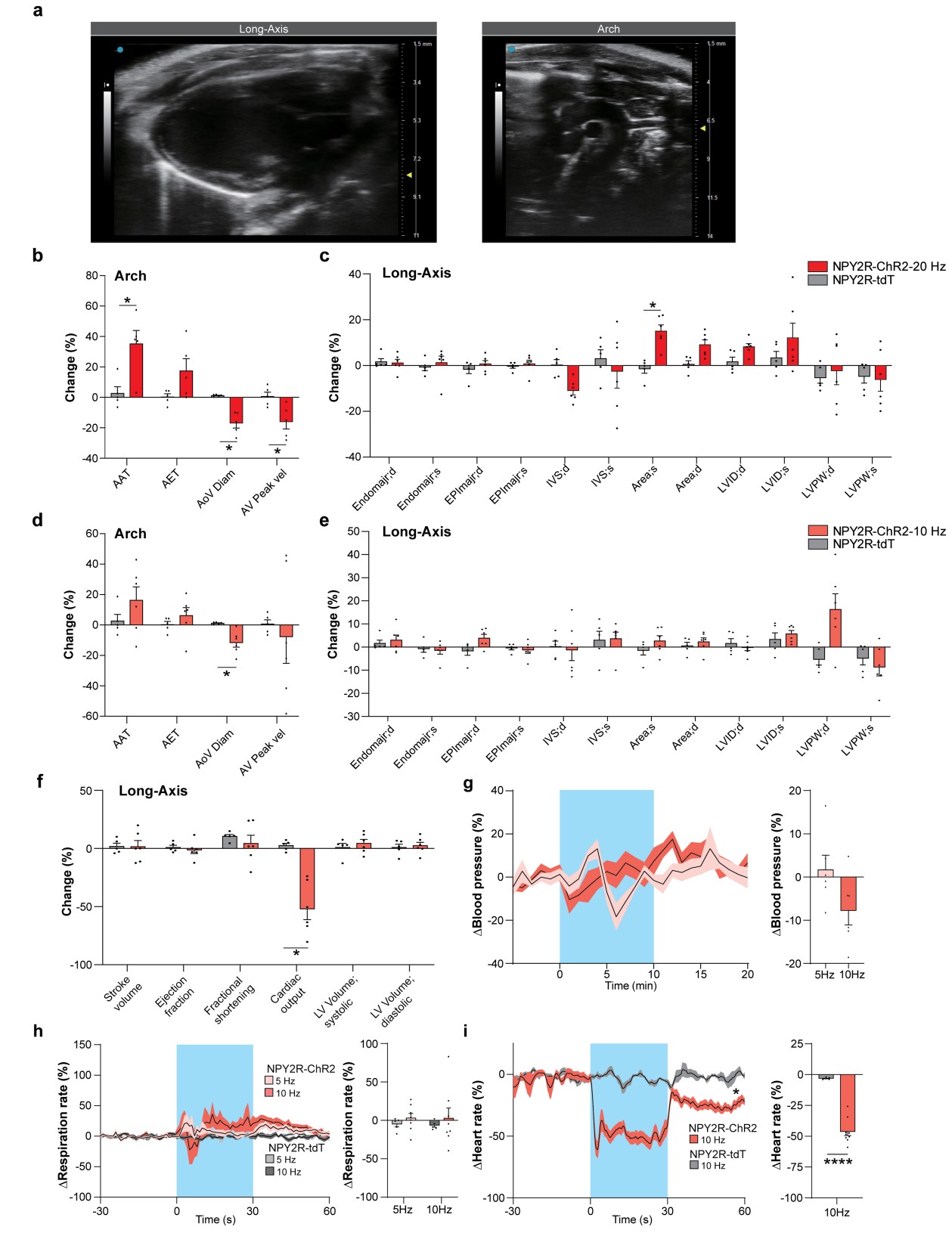

**Extended Data Fig. 5** | See next page for caption.

**Extended Data Fig. 5 | Physiological characterization of vagal NPY2R to area postrema (AP) photostimulation. a**, Representative ultrasound images of left ventricle (long-axis view) and aortic arch. **b**, 20 Hz photostimulation induced changes in the arch. After stimulation, the ascending aorta diameter decreased (AoV Diam, p = 0.0197). Aortic blood flow was also measured using a color Doppler function. Peak blood velocity in the ascending aorta (AV Peak vel, p = 0.0499) dropped. Aortic acceleration time (AAT, p = 0.0499) showed an increase induced by stimulation, indicating longer latency to reach peak velocity in the ascending aorta (right, n = 5). **c**, 20 Hz photostimulation induced changes in the left ventricle. After stimulation left ventricle end-systolic area increased significantly (Area;s, p = 0.0108), but no obvious change was observed in other parameters (left, n = 6 for ChR2, n = 5 for tdTomato). **d**, 10 Hz photostimulation induced changes in the aortic arch. After 10 Hz stimulation, only AoV Diam decreased (n = 6 for ChR2, n = 5 for tdTomato, p = 0.0346). **e-f**, 10 Hz photostimulation induced changes in the left ventricle. No obvious change was observed except for decreased cardiac output (n = 6 for ChR2, n = 5 for tdTomato, NPY2R/tdT mice with 20 Hz stimulation from Fig. 2d were reused as control, p = 0.0101). **g**, Blood-pressure during 5 Hz and 10 Hz photostimulation did not change compared to baseline (n = 6). **h**, Respiration-rates during 5 Hz and 10 Hz light stimulation showed no changes (n = 8 for ChR2, n = 5 for tdTomato). **i**, Heart-rate with 10 Hz stimulation during freely moving conditions showed sustained reduction during (left, p = 0.0353) and after laser delivery (n = 9 for ChR2, n = 5 for tdTomato, p < 0.0001). p < 0.05*, p < 0.0001**** by two-tailed unpaired t-tests with Holm-Šidák multiple comparisons or two-way ANOVA with Šidák multiple comparisons. All error bars and shaded areas show mean ± s.e.m.

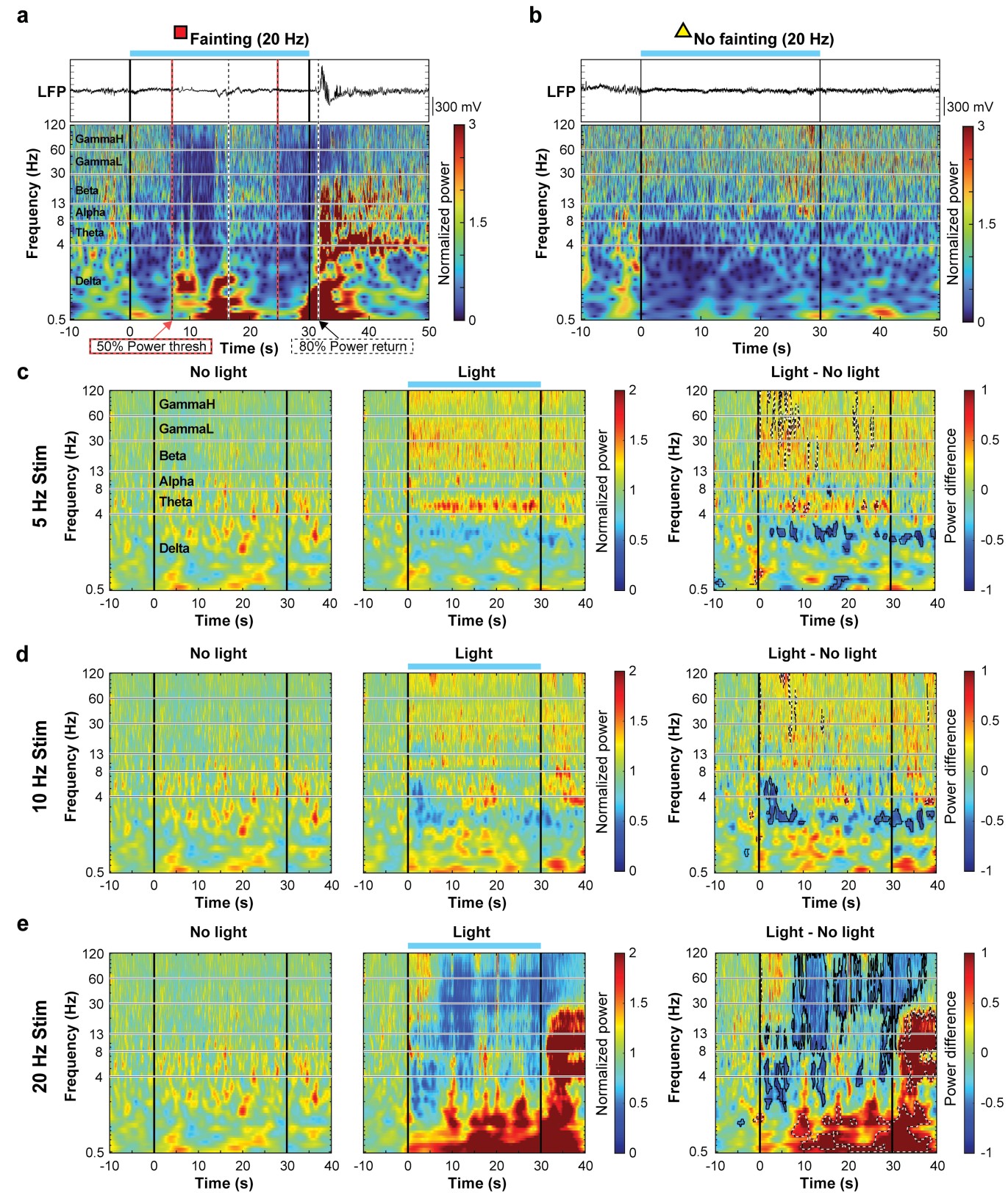

**Extended Data Fig. 6** | See next page for caption.

**Extended Data Fig. 6 | Syncope is only triggered by 20 Hz vagal NPY2R to area postrema (AP) photostimulation. a**, Single trial LFP analysis with on-target vagal NPY2R to AP stimulation (20 Hz) during head-fixed Neuropixels recording. Average raw LFP trace is shown on top, the Morse wavelet power transformation is shown below as a heatmap. The latency to when baseline normalized average power between 8–100 Hz dropped below 50% is indicated by dashed red/black lines, the latency to when average power returned to 80% is marked by dashed white/black lines. The period between is considered one "bout". **b**, Single trial LFP analysis of a mouse with off-target fiber placement. Note, no appreciable drop in 8–100 Hz power during the 20 Hz laser stimulation period. **c**, Group averaged power spectra in no light (left) and 5 Hz light conditions (middle). Mean power differences between conditions show scattered drops in power in the delta range and an increase in the beta-gamma range (right, n = 14 sessions across 8 mice for no light, n = 14 sessions across 7 mice for light). **d**, Same as in "**c**" but 10 Hz light, scattered drops in lower frequencies are observed, with minor effects in the gamma range (n = 14 sessions across 8 mice for no light, n = 14 sessions across 7 mice for light). **e**, 20 Hz photostimulation produced a delayed, but stark power drop in the alpha-gamma range which coincided with increased power in the lower delta range. This drop in the alpha-gamma range can be used as biomarker for syncope (n = 14 sessions across 8 mice for no light, n = 14 sessions across 7 mice for light).

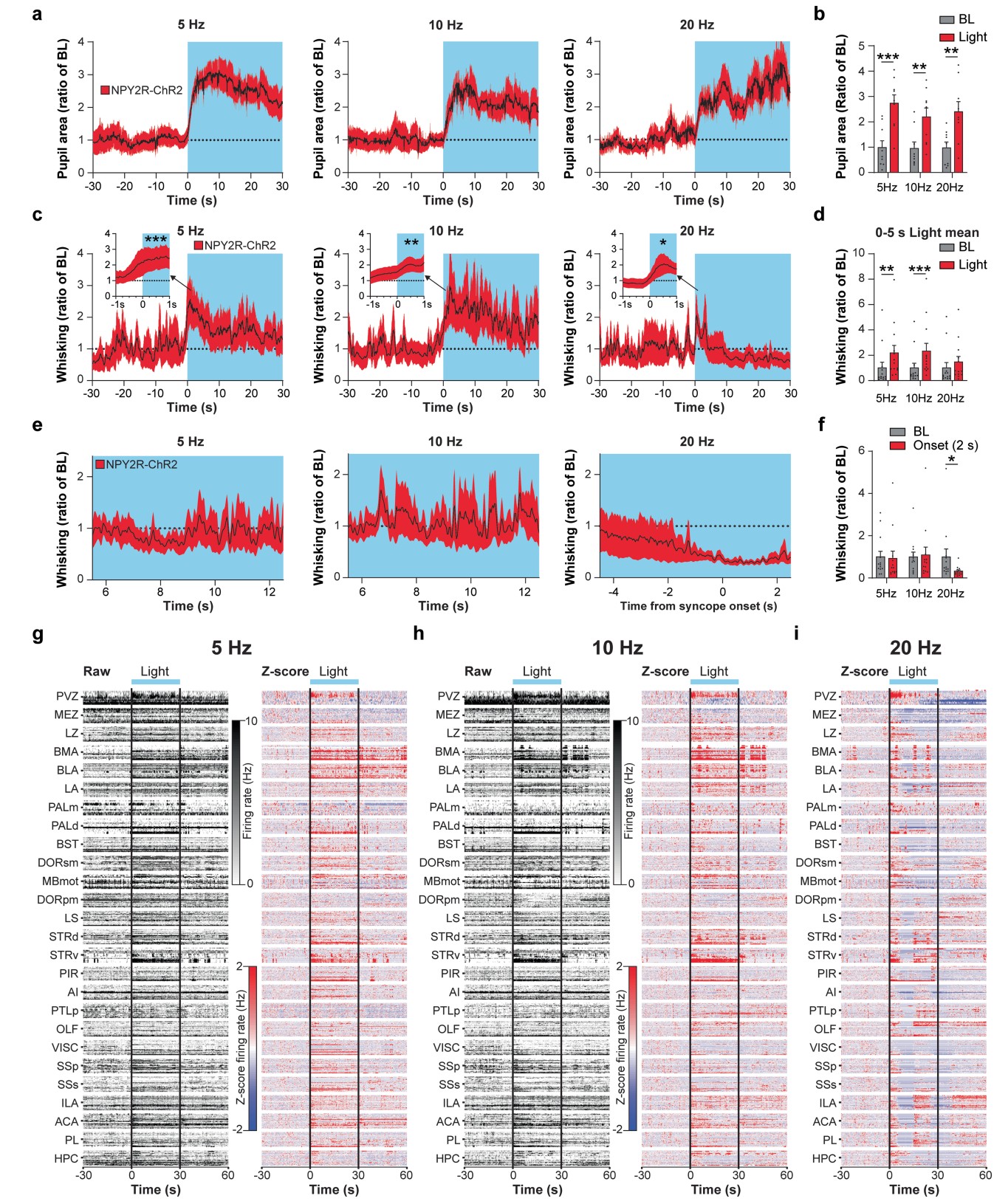

**Extended Data Fig. 7 |** See next page for caption.

**Extended Data Fig. 7 | Facial dynamics and Neuropixels data visualization during vagal NPY2R to area postrema (AP) stimulation. a**, Pupil area was tracked during photostimulation and averaged across sessions. **b**, Pupil area was significantly increased from baseline during photostimulation (n = 9 sessions from 5 mice, 5 Hz p = 0.0002, 10 Hz p = 0.0067, 20 Hz p = 0.0018). **c**, Max whisking values were collected in a 0-1 s time window after laser onset (insets) and averaged. These values were compared to normalized baseline whisking. Under all stimulation frequencies, mice quickly began whisking. **d**, Average whisking behavior increased during a 5 s window after laser onset at 5 and 10 Hz, but not the syncope inducing 20 Hz (n = 13 sessions from 7 mice, 5 Hz p = 0.0024, 10 Hz p = 0.0007). **e**, Aligning whisking behavior to syncope onset showed decrease of movement at 20 Hz (right) which was not observed at lower frequencies **f**, Quantification of whisking 2 s after syncope onset (n = 13 sessions from 7 mice, 20 Hz, p = 0.0254), and between 10–12 s after laser onset. **g-h**, Raw firing rates (left) and baseline z-scored firing rates (right) from all recordings across all animals at 5 Hz and 10 Hz stimulation in 100 ms time bins. **i**, Baseline z-scored firing rates from all recordings at 20 Hz stimulation in 100 ms time bins. Note the disruption of ongoing activity in most regions shortly after light onset under the 20 Hz condition as compared to 5 or 10 Hz: an average change of −0.17 in z-scored firing rate. p < 0.05*, p < 0.01**, p < 0.001*** by repeated measures two-way ANOVA with Šidák multiple comparisons. All error bars and shaded areas show mean ± s.e.m.

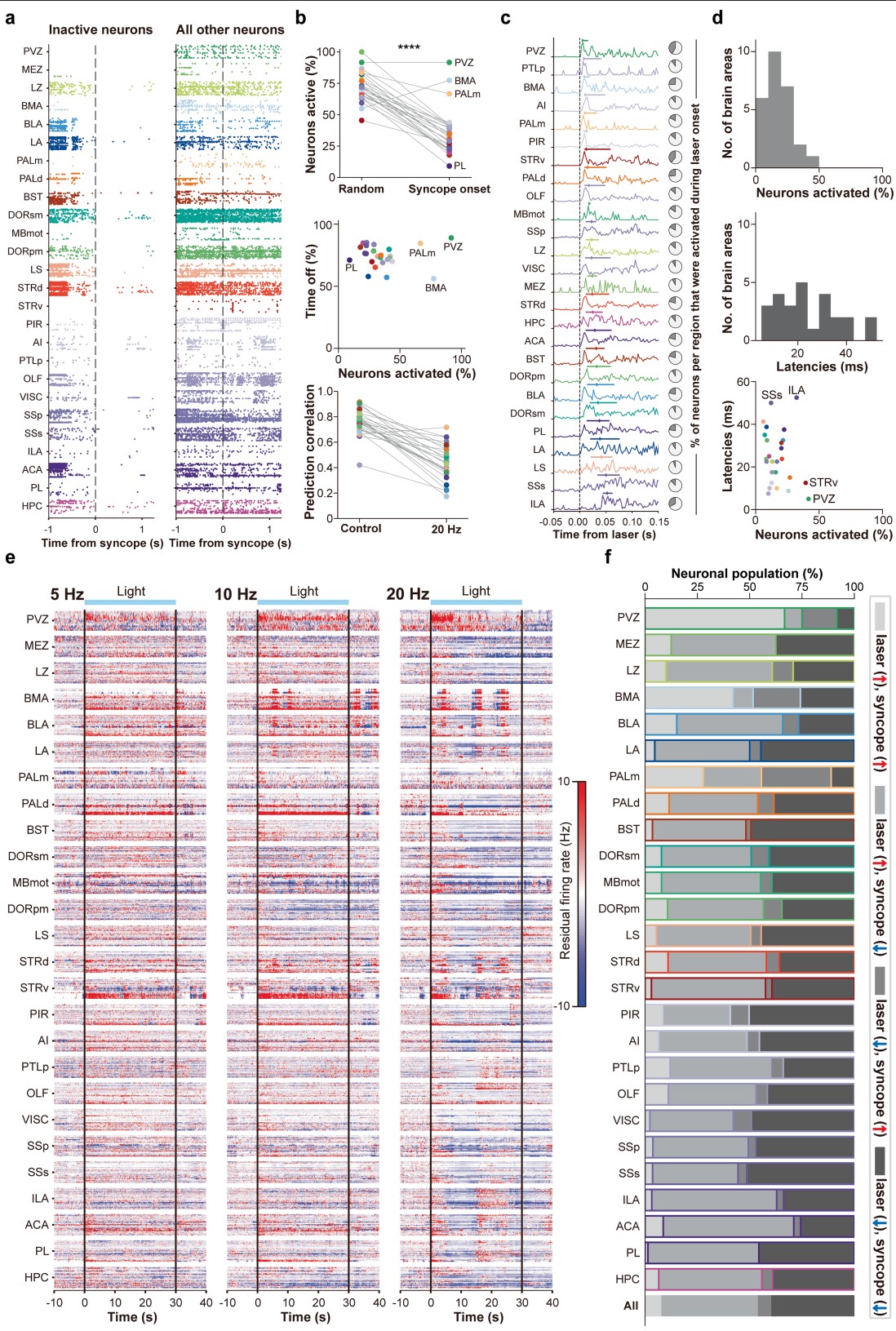

**Extended Data Fig. 8 |** See next page for caption.

**Extended Data Fig. 8 | Regionwise neuronal activity after vagal NPY2R to area postrema (AP) stimulation controlling for spontaneous movements.** **a**, Spike rasters of neurons aligned to syncope onset (as defined by 50% LFP drop) and divided into groups of neurons that became inactive vs all other neurons. **b**, Quantification of neural activity during syncope. The percentage of active neurons dropped dramatically during syncope compared to a random time period (top, p < 0.0001). Scatter plot for individual brain regions (middle). Correlation between the prediction and raw data across all timepoints per area, in the control segment and the 20 Hz laser segment, in time bins of 1 s (bottom). **c**, Quantification of laser-activated neurons in each region. Activity traces of neurons at laser onset are normalized to individual maximum firing rate and averaged. Horizontal lines above each trace denote 25–75% quartiles, and the tick along the line is the median value by which regions have been sorted in ascending order (shortest to longest latency). Pie charts indicate the proportion of recorded cells in each region that were activated in response to laser onset (right, gray shaded area). **d**, Quantification of brain-wide neuronal activation in response to laser onset. All regions are activated within 60 ms of laser onset (bottom). **e**, Residual firing rates at 5, 10, and 20 Hz photostimulation in 100 ms bins. **f**, Region-wise residuals from 20 Hz laser stimulation separated into 4 response property categories. "laser" and "syncope" labels indicate the time window of response (laser and syncope onset, respectively), the "(↑)" and "(↓)" indicates the direction of change in the residual i.e. ( ↑ ) means the behavioral model was under-predicting neuronal firing, ( ↓ ) means overprediction. Most brain regions are dominated by neurons that include "syncope (↓)". This indicated that the behavioral prediction model was mostly unable to predict reductions in neural activity during syncope. p < 0.001*** by two-tailed paired t-test.

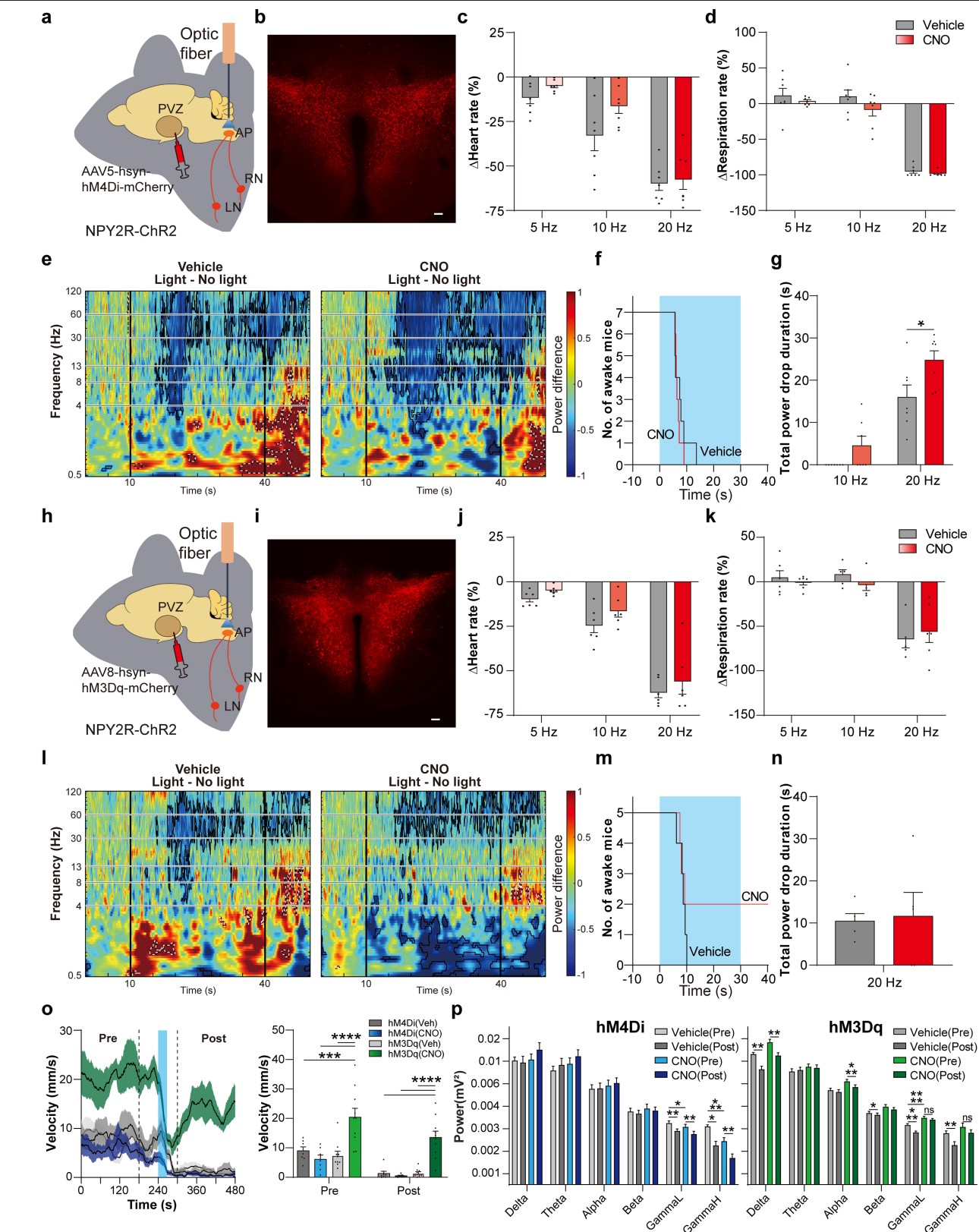

**Extended Data Fig. 9** | See next page for caption.

**Extended Data Fig. 9 | Bidirectional manipulation of the hypothalamic periventricular zone (PVZ) modifies syncope expression and states of arousal. a**, Illustration of vNAS with PVZ chemogenetic inhibition experiments. **b**, Representative image of hM4Di-mCherry expression in PVZ area. **c-d**, PVZ inhibition induced by CNO injection did not change the heart-rate reduction (**c**) or respiration-rate reduction (**d**) caused by photostimulation (n = 7). **e**, Subtraction plots showing replication of syncope triggered changes in EEG power under vehicle (left, n = 9 for no light, n = 7 for light) and an expanded area of power reduction under PVZ inhibition with CNO (right, n = 7). **f**, Step plot showing latency to first bout of immobility in CNO (red line, n = 7) and Vehicle (black line, n = 7) group with 20 Hz photostimulation (blue area). **g**, Total duration of EEG power drop was increased when the PVZ was inhibited during 20 Hz photostimulation (n = 7, p = 0.0449). At 10 Hz under CNO, >50% power drops were observed in 3 out of 7 mice, while that threshold was not reached under Vehicle. **h**, Illustration of vNAS with PVZ chemogenetic activation experiments. **i**, Representative image of hM3Dq-mCherry expression in PVZ area. **j-k**, PVZ excitation induced by CNO injection did not change the heart-rate reduction (**j**) or respiration-rate reduction (**k**) caused by photostimulation (n = 6). **l**, Subtraction plots showing replication of syncope triggered changes in power under vehicle (left) and unchanged power difference in the gamma range, but a decrease in delta (1–4 Hz) under CNO triggered PVZ activation (right, n = 10 sessions across 5 mice). **m**, Step plot showing latency to first bout of immobility in CNO (red line) and Vehicle (black line) group with 20 Hz photostimulation (blue area, n = 5). Note that 2 of the 5 mice did not faint under the CNO condition, suggesting a suppression of vNAS triggered syncope. **n**, Total duration of 50% power drop was unchanged between Vehicle and CNO conditions (n = 5). **o**, Average plots showing locomotor activity 4 min before 20 Hz stimulation (pre) and after (post) with PVZ chemogenetic manipulation under CNO. Mice with hM3Dq expression in the PVZ (green) showed dramatic increases in baseline locomotor activity (pre, hM4Di(Veh) p = 0.0001, hM4Di(CNO) & hM3Dq(Veh) p < 0.0001) under CNO and continued to move around the arena after recovering from 20 Hz induced syncope (post, hM4Di(Veh), hM4Di(CNO) & hM3Dq(Veh) p < 0.0001, n = 10 sessions from 5 mice for hM3dq, n = 7 sessions from 7 mice for hM4Di). **p**, Average fast-fourier transform (FFT) of EEG recording before and after 20 Hz photostimulation with PVZ chemogenetic manipulation under CNO. In hM4Di mice (left, blue n = 7) pairwise pre/post comparisons reveal significant drops in the gamma range after stimulation (GammaL(Veh) p = 0.0072, GammaH(Veh) p = 0.0101, GammaL(CNO) p = 0.0018, GammaH(CNO) p = 0.0012), indicating potential sustained reduction in arousal state, which is also reflected in their locomotor activity. In addition, CNO compared to Vehicle dropped gamma power even before 20 Hz stimulus was delivered (GammaL(pre) p = 0.0128, GammaH(pre) p = 0.0004). In hM3Dq mice (right, green, n = 10 session across 5 mice) pairwise pre/post comparisons reveal significant drops in delta (Vehicle p = 0.0019, CNO p = 0.0034) and alpha power (CNO p = 0.0006), and interestingly, no observable drops in the gamma range under CNO (GammaL(pre/post) p = 0.2377, GammaH(pre/post) p = 0.1449). Taken together, 20 Hz photostimulation may cause a sustained reduction in arousal state, while ongoing PVZ activation (CNO) causes an increase in baseline (pre) arousal state (GammaL p < 0.0001) which is maintained even after 20 Hz photostimulation. p < 0.05*, p < 0.01**, p < 0.001***, p < 0.0001**** by paired two-tailed t-tests, paired two-tailed t-tests with Holm-Šidák multiple comparisons, two-way repeated measures ANOVA with Šidák multiple comparisons or two-way repeated measures ANOVA with Holm-Šidák multiple comparisons. Scale bars, 100 μm. All error bars and shaded areas show mean ± s.e.m.

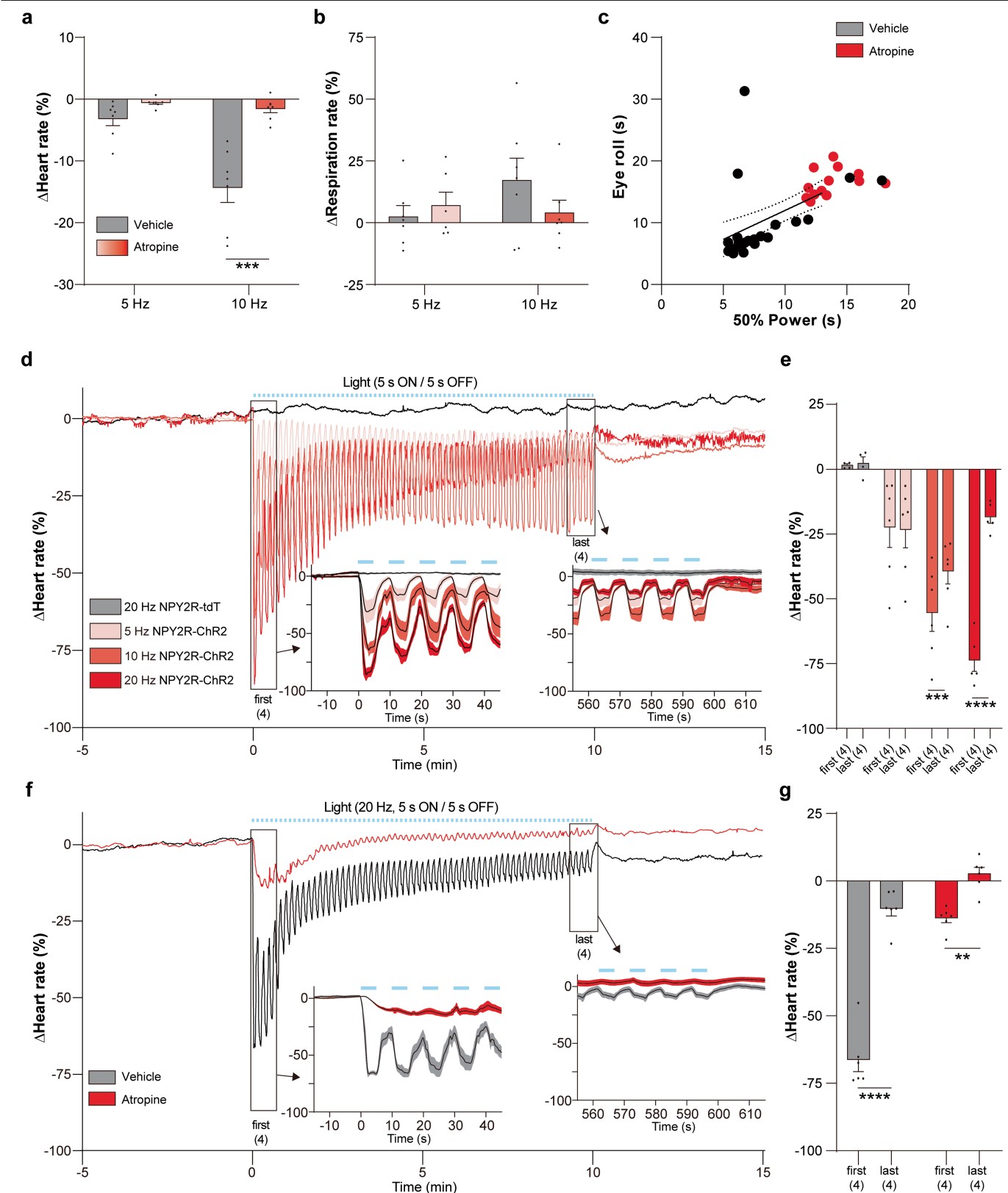

**Extended Data Fig. 10** | See next page for caption.

**Extended Data Fig. 10 | Atropine selectively attenuates vNAS induced drops in heart-rate, but does not alter sensory adaptation patterns.**
**a**, Effects of atropine on 5 and 10 Hz vNAS induced heart-rate reduction. Heart-rate reduction was significantly attenuated by atropine during 10 Hz (n = 7, p = 0.0006) but not 5 Hz photostimulation. **b**, Effects of atropine on 5, 10 Hz photostimulation induced respiration-rate reduction. No significant difference was observed between vehicle and atropine group (n = 7). **c**, Latency to eye-roll and 50% LFP/EEG drops were still correlated under atropine, but displayed longer latencies compared to vehicle (all combined head-fixed experiments, n = 22 for control, n = 13 for atropine, combined Pearson's r = 0.6099, p = 0.0001). **d**, Plot of group averaged heart-rate using 5, 10 and 20 Hz 5 s on/off photostimulation. Heart-rate rapidly changed during each 5 s on/off transition. Insets depict heart-rate adaptation across the first and last 4 light bursts over a 10 min period (n = 5 for 20 Hz, n = 6 for 5 & 10 Hz, n = 4 for NPY2R/tdT). **e**, The average of the first 4 minimum peaks was compared to the average of the last 4 minimum peaks. Robust adaptation was observed at both 20 Hz (p < 0.0001) and 10 Hz (p = 0.0005), with no adaptation at 5 Hz (n = 5 for 20 Hz, n = 6 for 5 & 10 Hz, n = 4 for NPY2R/tdT). **f**, Plot of group averaged heart-rate using 20 Hz 5 s on/off photostimulation with atropine. Atropine blunted rapid heart rate changes during the 5 s on/off stimulation pattern (n = 6). **g**, The average of the first 4 minimum peaks was compared to the average of the last 4 minimum peaks in vehicle (p < 0.0001) and atropine (p = 0.0039) condition. While atropine reduced the magnitude of heart-rate reduction, strong adaptation still occurred by the end of the train (n = 6). This indicates that adaptation to the 20 Hz photostimulation does not occur at the motor-output level but more likely at the sensory-input level. p < 0.01**, p < 0.001***, p < 0.0001**** by two-way repeated measures ANOVA with Šidák multiple comparisons, or Pearson's correlation. All error bars and shaded areas show mean ± s.e.m.

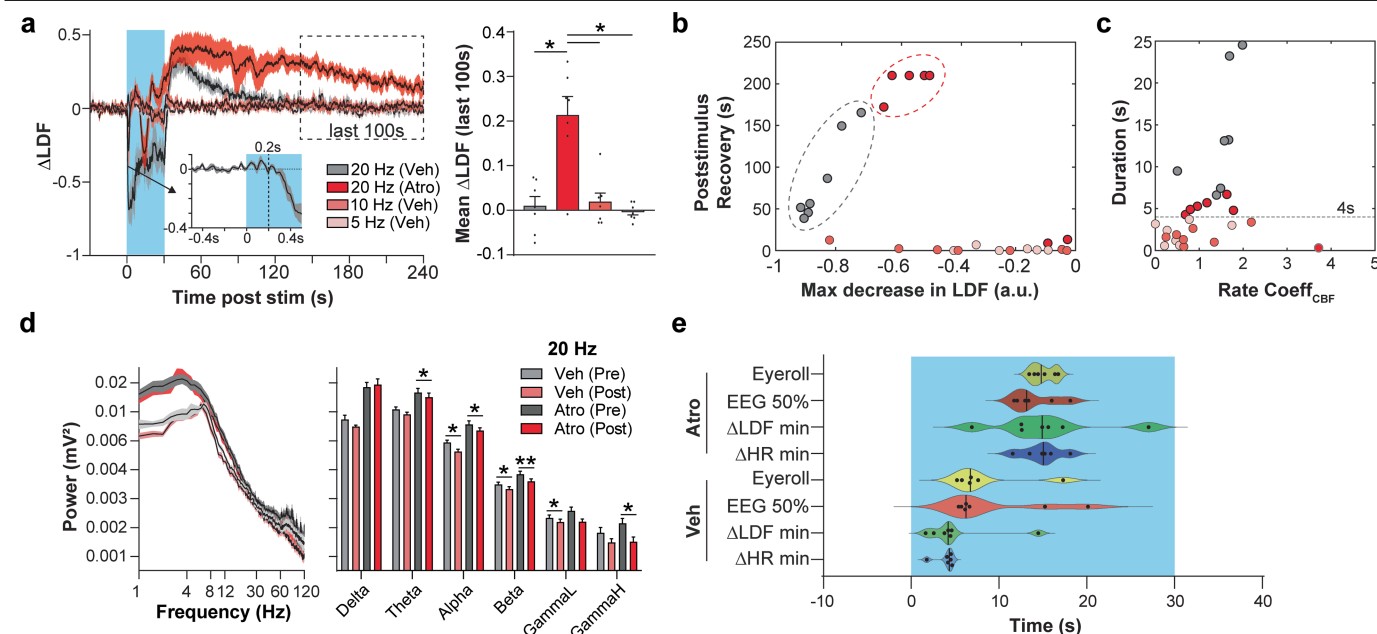

**Extended Data Fig. 11 | Atropine augments sustained changes in cerebral blood flow (CBF) and delays other metrics in response to vNAS. a**, Extended time window of LDF measured CBF (Fig. 4j) with vNAS (left). Atropine caused a sustained higher CBF for minutes following 20 Hz photostimulation. A 'late recovery phase' is marked as dashed box during last 100 s. Inset depicts at least 200 ms delay in changes in CBF following laser onset. Average CBF measurements during the last 100 s remained higher under atropine compared to all other conditions (right, n = 7, 20 Hz(Veh) p = 0.0444, 10 Hz(Veh) p = 0.0283, 5 Hz(Veh) p = 0.0219). **b**, Scatter plot of the time CBF takes to recover to baseline post vNAS, compared to the CBF minima during 30 s vNAS. Note that CBF under the atropine condition in 4 out of 7 mice, did not return to baseline during the recording window. **c**, Scatter plot of the duration of time spent below 50% of the baseline CBF compared to the rate coefficient, (i.e., rate of CBF decrease/rate of CBF increase, which captures the shape of CBF change across time, a higher rate coefficient indicates a rapid decrease followed by a slower increase)

during vNAS stimulation. The dashed line on the scatter plot marks a line (4 s duration CBF drop) that segregates fainters vs non-fainters. **d**, Mean fast-fourier transform (FFT) of EEG recording before and after 20 Hz vNAS (left). Atropine caused a general increase in power (three-way ANOVA, main effect of Atropine, p < 0.0002). Pairwise pre/post comparisons reveal drops in the beta-gamma range after stimulation (Beta(Veh) p = 0.0424. Beta(Atro) p = 0.0065, GammaL(Veh) p = 0.0327, GammaH(Atro) p = 0.0209), indicating potential sustained reduction in arousal state under head-fixed conditions (right, n = 7). **e**, Violin plot depicting the time-course of recorded events during 20 Hz vNAS. Note that atropine causes a delay in observed sequence of events (HR, LDF, EEG, and eye-roll, n = 6 for all except n = 7 for LDF min and EEG 50% vehicle). p < 0.05*, p < 0.01** by two-tailed paired t-tests with Holm-Šidák multiple comparisons or repeated measures ANOVA with Holm-Šidák multiple comparisons. All error bars and shaded areas show mean ± s.e.m.

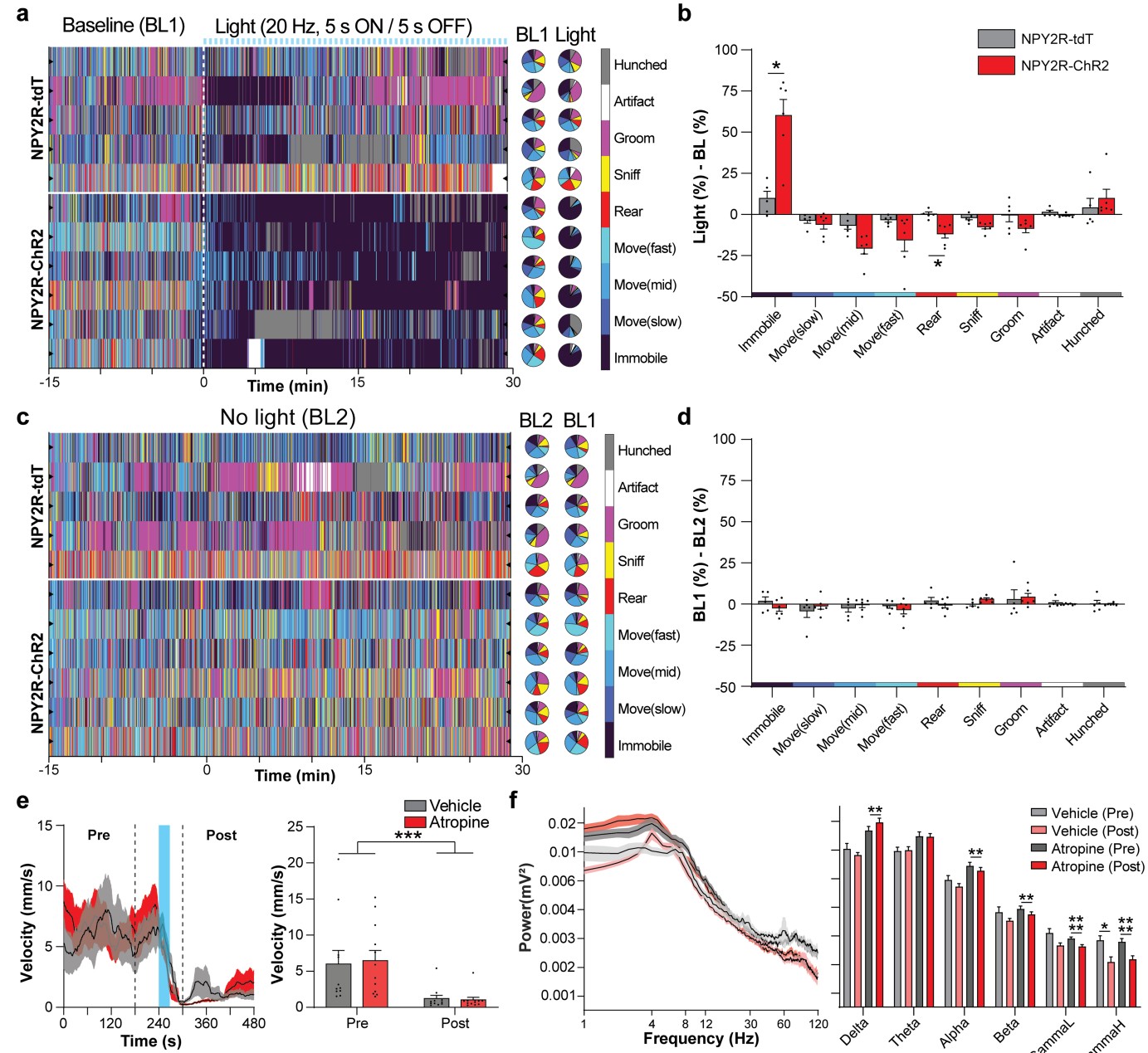

**Extended Data Fig. 12 | Behavioral and state changes caused by vNAS.**
**a**, Color-coded ethogram using the MoSeq pipeline (1 mouse per row).
Photostimulation promotes immobility. To the right of the ethogram, pie
charts show the proportion of time individual mice spent engaged in each
behavior during baseline (BL1) and photostimulation (Light) periods.
**b**, Quantification of the % change in behavior during light and baseline
conditions. Photostimulation significantly increased immobility (n = 6 for
ChR2, n = 5 for tdTomato, p = 0.0155) and decreased rearing (p = 0.0254).
**c**, Moseq data shown is the final day of habituation data one day prior to
experimental photostimulation. On the right, pie charts show the proportion
of time individual mice spent engaged in each behavior during the full
habituation period the day before (BL2) and during the 15 min baseline period
on the following photostimulation experimental day (BL1, **a**). **d**, Quantification
of baseline behavior during habituation day (BL2) and experimental baseline
day (BL1). Baseline behavior was the same across mice during both periods
(n = 6 for ChR2, n = 5 for tdTomato). **e**, Average plots showing locomotor activity

4 min before 20 Hz stimulation (pre) and after (post) with either vehicle or
atropine. In addition to acute syncope, brief 20 Hz vNAS for 30 s causes sustained
reduction in mouse locomotion which is not altered by atropine (n = 12 sessions
across 7 mice for atropine, n = 11 sessions across 7 mice for vehicle, two-way
repeated measures ANOVA, main effect of pre/post, p = 0.0002). **f**, Fast-Fourier
transform (FFT) of EEG recording before and after 20 Hz photostimulation.
Average traces of FFT profile (left) show a general increase in power under
atropine (n = 12 sessions across 7 mice, three-way ANOVA, main effect of Atropine,
p = 0.0108, replicated in Extended Data Fig. 11d). Pairwise pre/post comparisons
(right) reveal significant drops in the beta-gamma range after stimulation
(Beta(Atro) p = 0.0015, GammaL(Atro) p < 0.0001, GammaH(Veh) p = 0.0301,
GammaH(Atro) p < 0.0001), indicating potential sustained reduction in arousal
state, which is also reflected in their locomotor activity. p < 0.05*, p < 0.01**,
p < 0.001***, p < 0.0001**** by two-tailed paired or unpaired t-tests with
Holm-Šidák multiple comparisons or two-way repeated measures ANOVA. All
error bars and shaded areas show mean ± s.e.m.

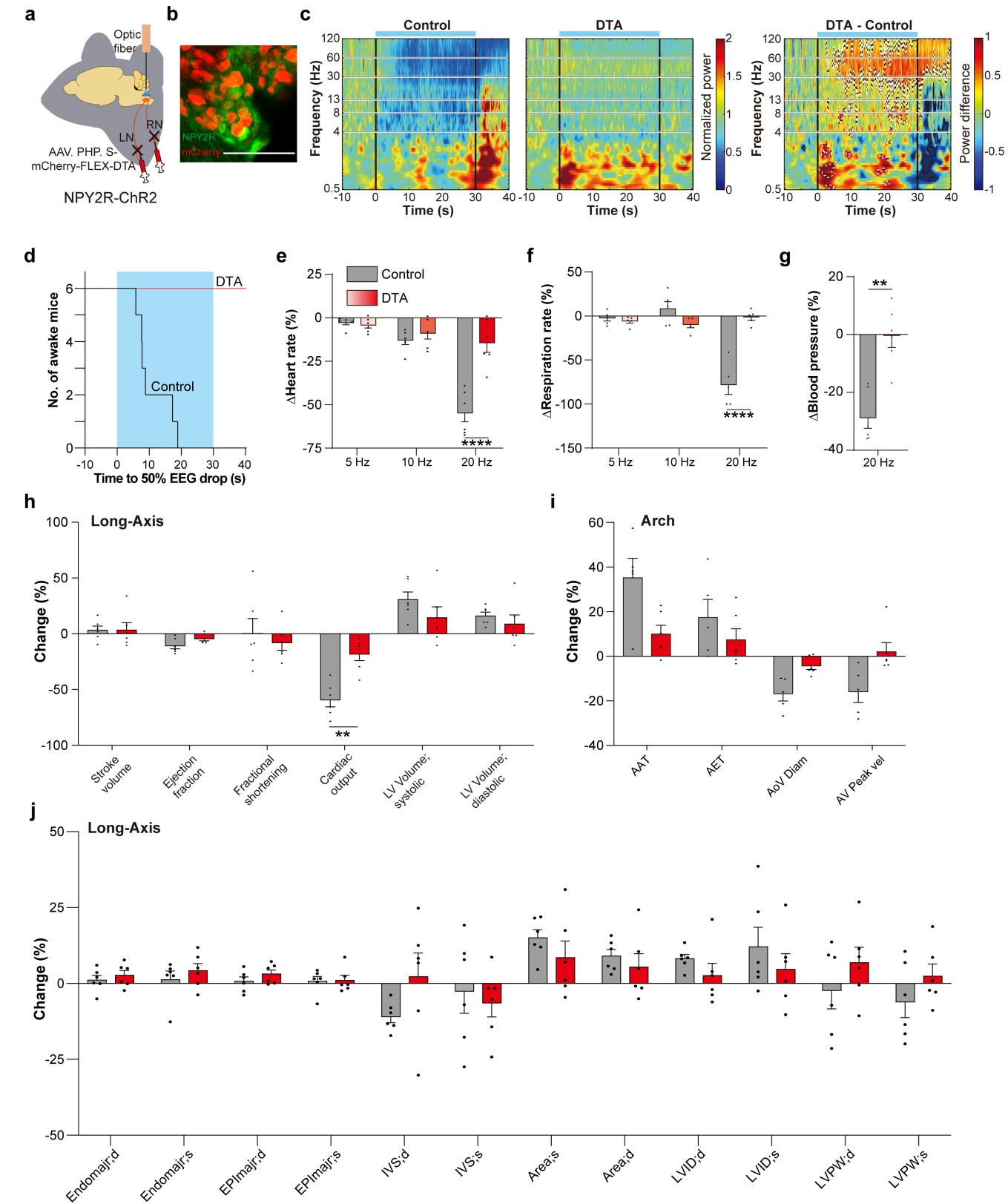

**Extended Data Fig. 13** | See next page for caption.

**Extended Data Fig. 13 | Ablation of NPY2R VSNs abolishes vNAS induced syncope and physiological changes. a**, Schematic of vNAS with diphtheria toxin subunit A (DTA) mediated ablation of NPY2R VSNs. **b**, Representative image of nodose ganglia injected with AAV-mCherry-FLEX-DTA virus and stained with NPY2R antibody (n = 13). Scale bar, 100 μm. **c**, Mice were tested for syncope and EEG power drops before DTA injection (left, n = 11 sessions across 7 mice). After DTA ablation of NPY2R VSNs, the same mice showed no appreciable drops in EEG power (middle, n = 12 sessions across 6 mice). Subtraction plots show significant differences in expected time x frequency ranges that are associated with syncope. **d**, Step plot showing latency to 50% mean power drop relative to baseline between 8–100 Hz with photostimulation (blue area). Notably, no mice reached the 50% power threshold after DTA ablation (n = 6). **e**, vNAS induced heart-rate reduction in control and DTA group. DTA mice showed less robust reduction compared to control group with 20 Hz stimulation (n = 6, p < 0.0001). **f**, vNAS induced respiration-rate changes in control and DTA group. During 20 Hz vNAS, respiration-rate reduction was markedly suppressed in DTA group (n = 5, p < 0.0001). **g**, Blood-pressure did not decrease with 20 Hz vNAS in DTA group (n = 6, p = 0.0019). **h-j**, Cardiac metrics of left ventricle (n = 6) and aortic arch (DTA n = 6, control n = 5) measured by ultrasound in DTA mice with 20 Hz vNAS. Reduction in cardiac output (p = 0.0043) was blocked. Changes in other parameters also showed a trend of being blunted compared to control group (control group data was reused from 20 Hz NPY2R/ChR2 mice Fig. 2d, Extended Data Fig. 5b, c). p < 0.01**, p < 0.0001**** by two-tailed paired or unpaired t-tests with Holm-Šidák multiple comparisons or two-way repeated measures ANOVA with Šidák multiple comparisons. All error bars show mean ± s.e.m.

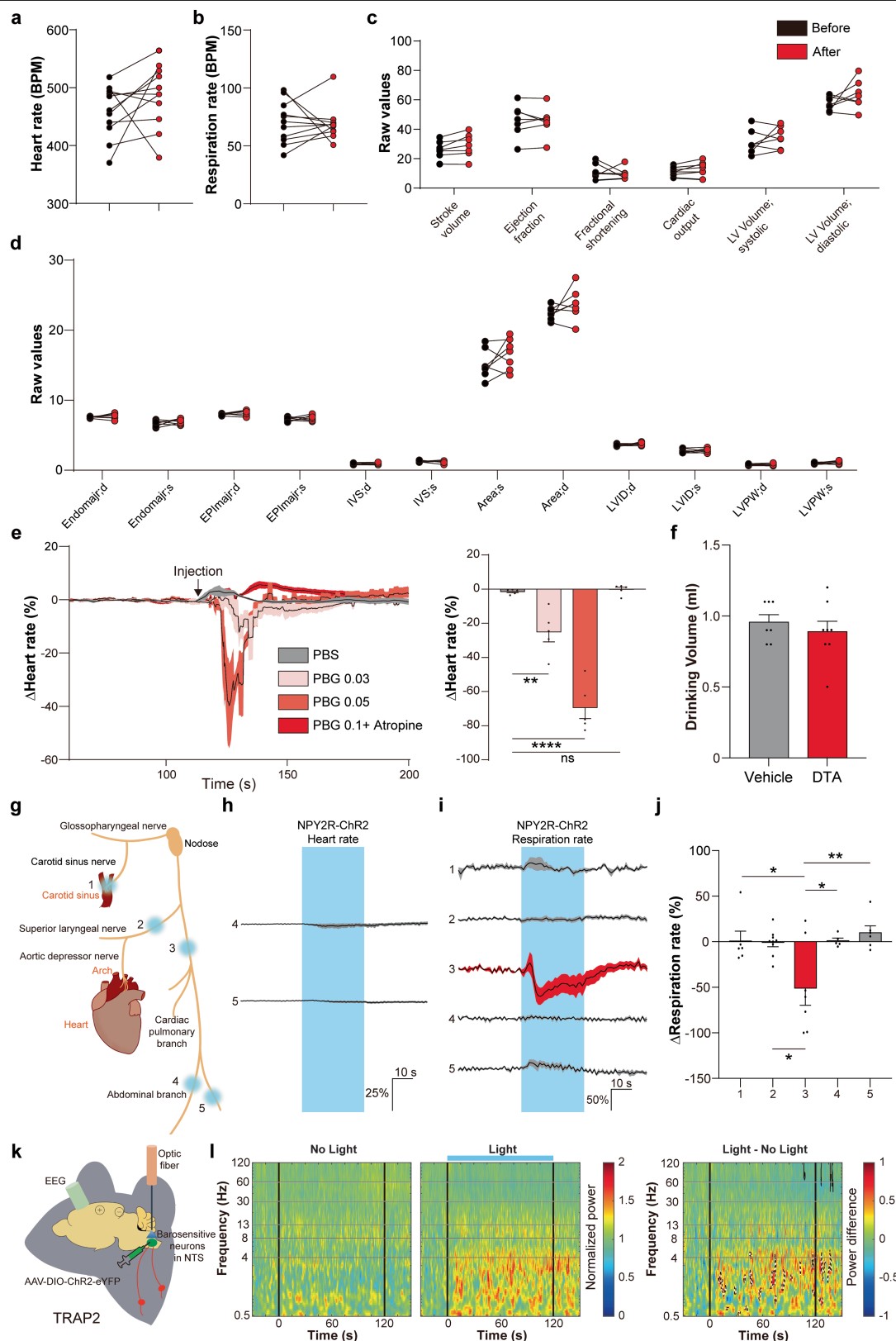

**Extended Data Fig. 14** | See next page for caption.

**Extended Data Fig. 14 | Specificity of the vagal bed in the Bezold-Jarisch reflex. a-d**, DTA ablation of NPY2R VSNs did not affect basic cardiac functions (**a-b** n = 11, **c-d** n = 7). **e**, PBG induced a dose dependent dip in heart-rate which was blocked by atropine (left) and quantification (right, n = 5, PBS compared with PBG 0.03 p = 0.0030, PBG 0.05 p < 0.0001, PBG 0.1+atropine p = 0.8466). **f**, DTA ablation did not affect water intake after 48-hour water deprivation (DTA n = 8, control n = 7). **g**, Illustration of region-specific vagal optogenetic strategy. **h**, Heart-rate did not change after abdominal branch photostimulation (n = 6). **i**, Average traces of respiration-rate with region-specific stimulation in NPY2R-ChR2 mice. **j**, Respiration-rate only decreased while stimulating the vagal trunk (region 3) above the abdominal branch (region[1] n = 6, [2] n = 9, [3] n = 7, [4] n = 5, [5] n = 6, region[3] compared to region[1] p = 0.0193, [2] p = 0.0106, [4] p = 0.0269, [5] p = 0.0044). **k**, Schematic of TRAP mediated optogenetic stimulation of barosensitive neurons in the NTS. **l**, Average power plots from a representative mouse using 17 trials of no light (left) or photostimulation (middle) of TRAPed barosensitive neurons of the NTS. Subtraction plots show no evidence of syncope-like changes in gamma power observed in vNAS (right) despite large dips in heart-rate. In addition, as published increases in the delta range were observed, consistent with previous observations (data from a published report[46] provided by Dr. Yang Dan). p < 0.05*, p < 0.01**, p < 0.0001**** by one way ANOVA with Holm-Šidák multiple comparisons or Tukey multiple comparisons. All error bars and shaded areas show mean ± s.e.m.

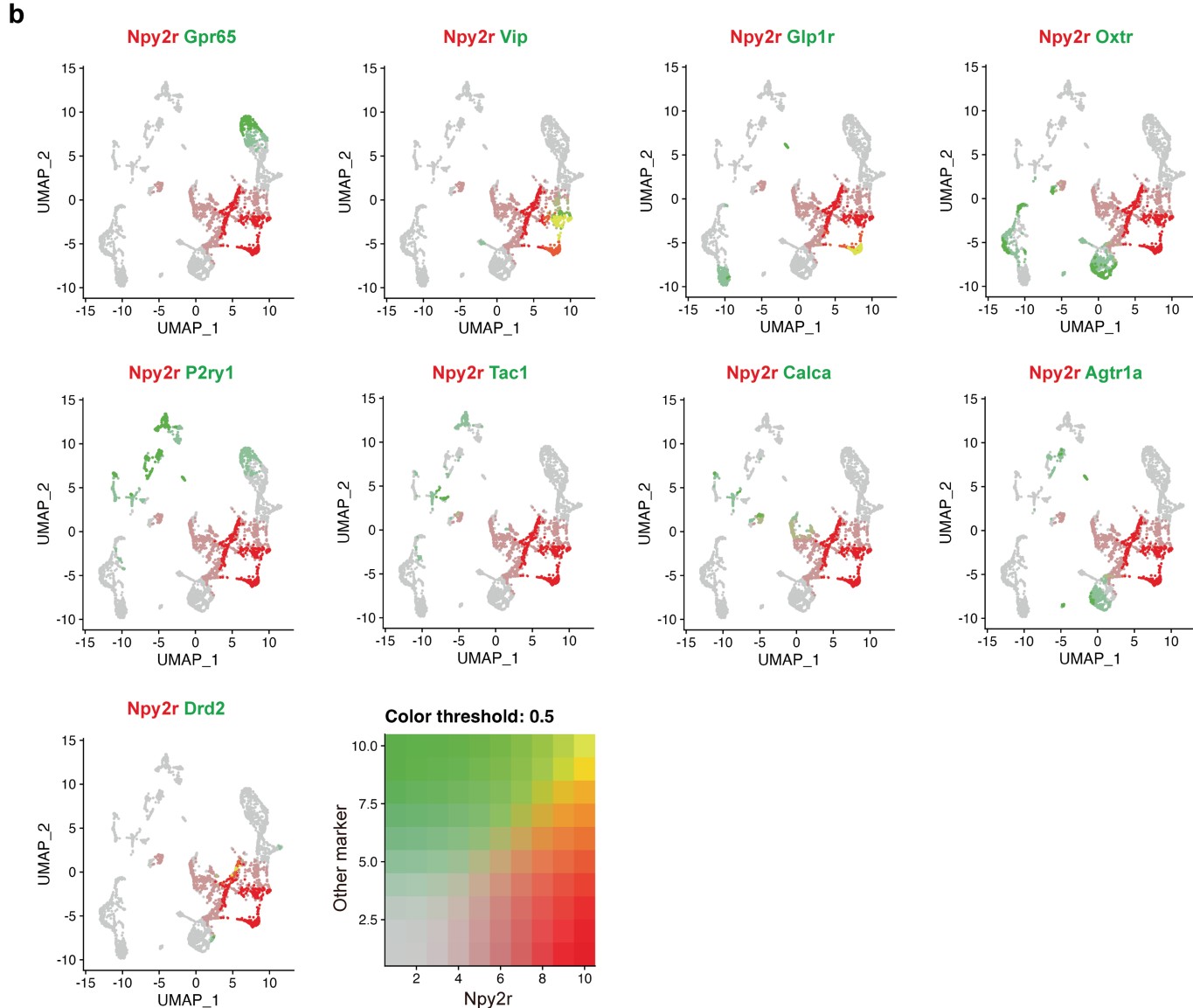

**a**

| Marker | Organs | Studies |
|--------|--------|---------|
| Gpr65 | Stomach, small intestine, colon | Zhao et al. 2022, Bai et al. 2019 |
| Vip | Small intestine | Bai et al. 2019 |
| Glp1r | Stomach, small intestine | Bai et al. 2019 |
| Oxtr | Stomach, small intestine, colon | Bai et al. 2019 |
| P2ry1 | Lung, trachea, stomach | Chang et al. 2015, Prescott et al. 2020, Zhao et al. 2022 |
| Tac1 | Lung | Liu et al. 2021 |
| Calca | Lung, stomach | Bai et al. 2019, Kupari et al. 2019, Zhao et al. 2022 |
| Agtr1a | Lung, colon, small intestine, heart | Zhao et al. 2022, Kupari et al. 2019 |
| Drd2 | Heart, stomach | Zhao et al. 2022 |

**b**

**Extended Data Fig. 15 | Target defined scRNA seq data analysis of the nodose ganglia. a**, A list of genetic markers and their primary anatomical targets which have been previously reported. Syncope has never been reported with photostimulation of these various genetically defined VSNs. **b**, Single cell RNA sequencing of nodose ganglia, showing comparison of NPY2R (red) and other markers (green) expression in VSN clusters (reanalyzed from previous reports[20,21,24,67–69]).

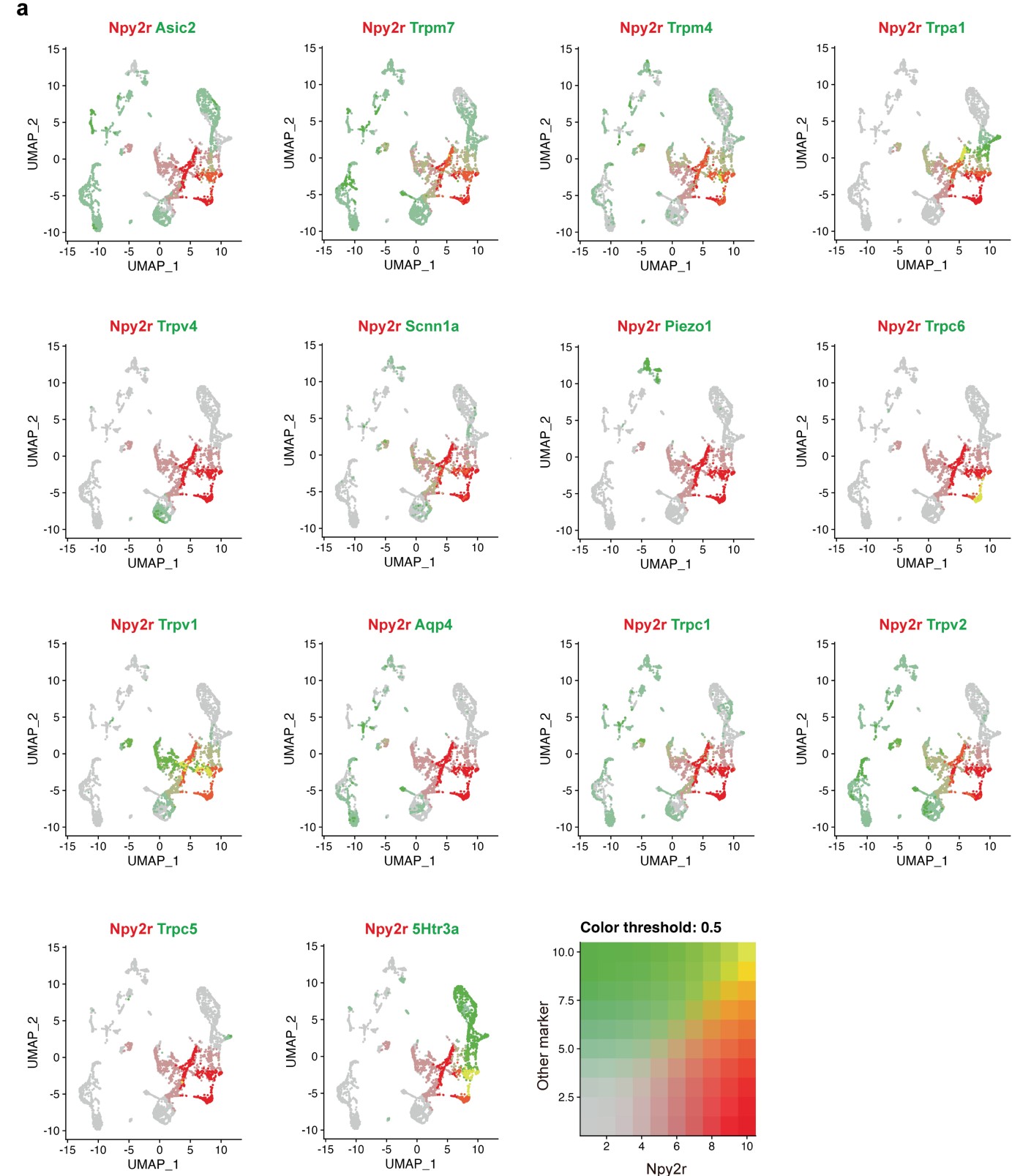

**Extended Data Fig. 16 | Putative receptors and channels expressed by NPY2R VSNs. a**, Single cell RNA sequencing of nodose ganglia, showing comparison of NPY2R (red) and some putative receptor[47] and channel (green) expression in VSN clusters.

# Reporting Summary

## Statistics

For all statistical analyses, confirm that the following items are present in the figure legend, table legend, main text, or Methods section.

| n/a | Confirmed | |
|---|---|---|
| ☐ | ☒ | The exact sample size (*n*) for each experimental group/condition, given as a discrete number and unit of measurement |
| ☐ | ☒ | A statement on whether measurements were taken from distinct samples or whether the same sample was measured repeatedly |
| ☐ | ☒ | The statistical test(s) used AND whether they are one- or two-sided<br>*Only common tests should be described solely by name; describe more complex techniques in the Methods section.* |
| ☐ | ☒ | A description of all covariates tested |
| ☐ | ☒ | A description of any assumptions or corrections, such as tests of normality and adjustment for multiple comparisons |
| ☐ | ☒ | A full description of the statistical parameters including central tendency (e.g. means) or other basic estimates (e.g. regression coefficient) AND variation (e.g. standard deviation) or associated estimates of uncertainty (e.g. confidence intervals) |
| ☐ | ☒ | For null hypothesis testing, the test statistic (e.g. *F*, *t*, *r*) with confidence intervals, effect sizes, degrees of freedom and *P* value noted<br>*Give P values as exact values whenever suitable.* |
| ☒ | ☐ | For Bayesian analysis, information on the choice of priors and Markov chain Monte Carlo settings |
| ☒ | ☐ | For hierarchical and complex designs, identification of the appropriate level for tests and full reporting of outcomes |
| ☐ | ☒ | Estimates of effect sizes (e.g. Cohen's *d*, Pearson's *r*), indicating how they were calculated |

*Our web collection on statistics for biologists contains articles on many of the points above.*

## Software and code

Policy information about availability of computer code

| Data collection | FLUOVIEW 2.4.1.198, SmartSPIM GUI 2.1, SmartSPIM Destriping and Stitching 2.0, NIS-Elements 5.21.03, SpikeGLX 3.0, MoSeq2, AcqKnowledge, Vevo 3100, SpinView, CODA software 4.2 and custom Matlab 2022b, 2023a/ Python 3.6 code for collecting data. |
|---|---|
| Data analysis | Microsoft Excel, Premiere Pro CC 2022 (22.0), GraphPad Prism 9.4.1, 9.5.1 and 10.0.0, Imaris 9.2.1 and 10.0.0, ImageJ 1.8.0, Vevo LAB 5.5.0, AcqKnowledge 5.0, Kilosort3, Phy v2.0b1, SHARP-track, MoSeq2, Facemap, DeepLabCut 2.2, ToxTrac2.98, and custom R/Matlab 2022b, 2023a/ Python 3.6 code for analyzing behavior, Neuropixels and single cell RNA sequencing data (Seurat v3). |

For manuscripts utilizing custom algorithms or software that are central to the research but not yet described in published literature, software must be made available to editors and reviewers. We strongly encourage code deposition in a community repository (e.g. GitHub). See the Nature Portfolio guidelines for submitting code & software for further information.

## Data

Policy information about availability of data

All manuscripts must include a data availability statement. This statement should provide the following information, where applicable:
- Accession codes, unique identifiers, or web links for publicly available datasets
- A description of any restrictions on data availability
- For clinical datasets or third party data, please ensure that the statement adheres to our policy

> All numerical data are included in Supplementary Information. All other data is large and available from the corresponding author upon reasonable request. SIngle cell sequencing data was obtained from GEO (GSE145216).

## Human research participants

Policy information about studies involving human research participants and Sex and Gender in Research.

| | |
|---|---|
| Reporting on sex and gender | NA |
| Population characteristics | NA |
| Recruitment | NA |
| Ethics oversight | NA |

Note that full information on the approval of the study protocol must also be provided in the manuscript.

# Field-specific reporting

Please select the one below that is the best fit for your research. If you are not sure, read the appropriate sections before making your selection.

☒ Life sciences    ☐ Behavioural & social sciences    ☐ Ecological, evolutionary & environmental sciences

For a reference copy of the document with all sections, see nature.com/documents/nr-reporting-summary-flat.pdf

# Life sciences study design

All studies must disclose on these points even when the disclosure is negative.

| | |
|---|---|
| Sample size | No sample size statistics was performed. Sample sizes were similar to recent papers (Augustine, V. et al. Nature 555, 204-209, 2018 and Lee, S. et al. Nature 568, 93-97, 2019). |
| Data exclusions | Animals with proper viral expression or implant placement were included in the analysis. For ECG and respiration, recordings with excessive noise were excluded. For facial features analysis, sessions where pupils were obscured were excluded. |
| Replication | All attempts at replication were successful. N numbers are indicated in the legends. |
| Randomization | No randomization was performed. Animals were arbitrarily assigned to experimental groups. |
| Blinding | No blinding procedures were performed. Representative data was based on at least 3 independent observations. Control and experimental groups were tested using similar conditions. Control and experimental data was analyzed using same criteria. |

# Reporting for specific materials, systems and methods

We require information from authors about some types of materials, experimental systems and methods used in many studies. Here, indicate whether each material, system or method listed is relevant to your study. If you are not sure if a list item applies to your research, read the appropriate section before selecting a response.

## Materials & experimental systems

| n/a | Involved in the study |
|-----|----------------------|
| ☐ | ☒ Antibodies |
| ☒ | ☐ Eukaryotic cell lines |
| ☒ | ☐ Palaeontology and archaeology |
| ☐ | ☒ Animals and other organisms |
| ☒ | ☐ Clinical data |
| ☒ | ☐ Dual use research of concern |

## Methods

| n/a | Involved in the study |
|-----|----------------------|
| ☒ | ☐ ChIP-seq |
| ☒ | ☐ Flow cytometry |
| ☒ | ☐ MRI-based neuroimaging |

## Antibodies

| | |
|---|---|
| Antibodies used | Primary antibodies used were:<br>Chicken anti-GFP (ab13970, Abcam, 1:500); rabbit anti-RFP (600-401-379, Rockland, 1:500); rabbit anti-NPY2R (RA14112, Neuromics, 1:500); Alexa Fluor™ 647 conjugated GFP polyclonal antibody (A-31852, Thermo Fisher Scientific, 1:200).<br>Secondary antibodies used were:<br>Alexa Fluor® 647 AffiniPure Donkey Anti-Rabbit (711-605-152, Jackson ImmunoResearch, 1:500); Alexa Fluor® 488 AffiniPure Donkey Anti-Chicken (703-545-155, Jackson ImmunoResearch, 1:500); Cy™3 AffiniPure Donkey Anti-Rabbit IgG (H+L) (711-165-152, Jackson ImmunoResearch, 1:500). |
| Validation | All antibodies used were previously published, validated, and purchased from commercial vendors in the USA.<br>Chicken anti-GFP (ab13970, Abcam) has been referenced in at least 3182 publications according to vendor's website. From vendor's website: "Our GFP antibody does cross-react with the many fluorescent proteins that are derived from the jellyfish Aequorea victoria. These are all proteins that differ from the original GFP by just a few point mutations (EGFP, YFP, mVenus, CFP, BFP etc.)." Citation: Wang G et al. Somatic genetics analysis of sleep in adult mice. J Neurosci 42:5617-40 (2022). IHC; Mouse.<br>Rabbit anti-RFP (600-401-379, Rockland) has been referenced in at least 928 publications according to vendor's website. From vendor's website: "This product was prepared from monospecific antiserum by immunoaffinity chromatography using Red Fluorescent Protein (Discosoma) coupled to agarose beads followed by solid phase adsorption(s) to remove any unwanted reactivities. Expect reactivity against RFP and its variants: mCherry, tdTomato, mBanana, mOrange, mPlum, mOrange and mStrawberry. Assay by immunoelectrophoresis resulted in a single precipitin arc against anti-Rabbit Serum and purified and partially purified Red Fluorescent Protein (Discosoma). No reaction was observed against Human, Mouse or Rat serum proteins." Citation: Zhang C et al. A brainstem circuit for nausea suppression. Cell Rep. (2022) Applications: IHC, ICC, Histology; Mouse.<br>Rabbit anti-NPY2R (RA14112, Neuromics) has been referenced in at least 14 publications according to vendor's website. From vendor's website: "This NPY Y2 antibody has been extensively characterized in the spinal cord, brain, and distal colon." Citation: Neuropeptide Y in the Medial Habenula Alleviates Migraine-Like Behaviors through the Y1 Receptor. IHC; Mouse.<br>Alexa Fluor™ 647 conjugated GFP polyclonal antibody (A-31852, Thermo Fisher Scientific) has been referenced in at least 30 publications according to vendor's website. From vendor's website: "This Antibody was verified by Relative expression to ensure that the antibody binds to the antigen stated." Citation: Pyk2 suppresses contextual fear memory in an autophosphorylation-independent manner. IHC; Mouse.<br>Alexa Fluor® 647 AffiniPure Donkey Anti-Rabbit (711-605-152, Jackson ImmunoResearch) has been referenced in at least 434 publications according to vendor's website.<br>Alexa Fluor® 488 AffiniPure Donkey Anti-Chicken (703-545-155, Jackson ImmunoResearch) has been referenced in at least 546 publications according to vendor's website.<br>Alexa Fluor® 555 AffiniPure Donkey Anti-Rabbit (703-545-155, Jackson ImmunoResearch) has been referenced in at least 546 publications according to vendor's website.<br>Cy™3 AffiniPure Donkey Anti-Rabbit IgG (H+L) (711-165-152, Jackson ImmunoResearch) has been referenced in at least 1306 publications according to vendor's website.<br>We also validated all antibodies listed above with different tissues. |

## Animals and other research organisms

Policy information about studies involving animals; ARRIVE guidelines recommended for reporting animal research, and Sex and Gender in Research

| | |
|---|---|
| Laboratory animals | Experiments were conducted on adult mice, both male and female between 1.5 - 6 months of age. The following mice lines were purchased from the Jackson Laboratory: C57BL/6J, stock number 000664; Ai9, stock number 007909; Ai32; stock number 012569; Slc17a6-Cre (also known as Vglut2-Cre), stock number 016963; Npy2r-ires-Cre, stock number 029285; Piezo2-EGFP-ires-Cre, stock number 027719. Mice were maintained in temperature (around 22-23C) controlled rooms with a 12 h:12h light:dark cycle (6am to 6 pm light on) and ad libitum access to chow and water. |

| | |
|---|---|
| Wild animals | No wild animals were used. |
| Reporting on sex | Both female and male mice were used. |
| Field-collected samples | No field-collected samples were used. |
| Ethics oversight | All procedures were done according to NIH and Institutional Animal Care and Use Committee (IACUC; protocol # 19-0018) guidelines at Scripps Research. |

Note that full information on the approval of the study protocol must also be provided in the manuscript.

