## [Peer Review File · Nature]

Manuscript Title: Vagal sensory neurons mediate the Bezold-Jarisch reflex and induce syncope

Reviewer Comments & Author Rebuttals

Reviewer Reports on the Initial Version:

Referees' comments:

Referee #1 (Remarks to the Author):

Summary and significance:

In this study Lovelace et al provide beautiful and convincing data showing that Neuropeptide Y2 (NPY2R) expressing vagal sensory neurons (VSNs) project to organs including the ventricles (but not atria) of the heart and also the lung, and that these NPY2R-VSNs form distinct populations within the nodose ganglion, supporting the notion of a one-to-one map for organ innervation by VSNs. While both subsets project to the NTS, the ventricle innervating NPY2R VSNs also strongly project to the area postrema (AP).

The authors then use an optogenetic approach to investigate the potential physiological role of these ventricle-projecting NPY2R-VSNs. Optogenetic stimulation of the AP in NPY2R-CHR2 mice (vNAS) induces a striking cluster of responses that include syncope (Fig 1), aversion/negative valence (Fig 2), Bezold-Jarisch reflex like responses (bradycardia, hypotension, reduced ventilation rate; Fig 3), and widespread inactivation of central neural activity (Figs 4).

Although quite impressive, the experiments described in Figure 5 (including the computational analysis) do not add substantially to the major findings of the paper. Although they do provide additional evidence for neuronal inactivation during the syncope phase and indicate this inactivation is not indirectly related to facial and eye movements also caused by vNAS, they do not shed direct light on the downstream pathways that causally mediate the overt cardiac effects, other phenotypes, or the basis for neuronal inactivation. The authors do indicate in their discussion that "Beyond clinical application, utilization of this 18 optogenetic "switch" could benefit the study of historically difficult questions regarding conscious and unconscious brain state transitions". Although this is true, it is still completely unclear if the inactivation results from the recruitment of a hypothetical "syncope network" or simply reduced blood flow associated with the reduced cardiac output.

If consolidated by additional experiments (see below), these data would provide the first evidence for a genetically- and anatomically defined sensory pathway involved in a ventricular-cardiac reflex arc. Although the efferent components of the reflex remain to be determined, this is a terrific tour de force that dramatically improves our understanding of vagal sensory afferents, cardiovascular function and syncope. This would be of widespread interest to the readers of Nature.

Critique:

Although the phenotypes induced by vNAS were very dramatic and interesting, it appears that "NPY2R-CHR2" mice were simply produced by crossing NPY2R-Cre mice to the Ai32 (cre-dependent CHR2) line. As such, AP neurons expressing NPY2R (if present) would also express CHR2, as would the axon terminals of any central NPY2R-expressing afferents to AP neurons, and therefore

optogenetic stimulation of the AP area in NPY2R-CHR2 mice (“vNAS”) would not only activate the terminals of NPY2R-VSNs, but also local AP neurons that may be unrelated (not synaptically connected) to the NPY2R-VSN input. If this is the case, it cannot be concluded that the effects caused by vNAS are specifically mediated by ventricular NPY2R-VSNs because the AP is well known to be involved in the genesis of nausea and cardiac effects. Thus, additional data should be provided to prove the specificity of the vNSA approach. Ideally this would include experiments in mice which express CHR2 exclusively in the ventricular-projecting VSNs (e.g. using viral delivery via the ventricle or other equivalent approach).

Referee #2 (Remarks to the Author):

Lovelace and colleagues utilize genetics to specifically manipulate subtypes of vagus sensory neurons (VSNs) and study their function by observing animal’s behavior as well as physiology and neural activity. They first found that a subset of Npy2r-expressing VSNs project to the AP and innervate the heart but not the lung. Furthermore, they found that optogenetic activation of these neurons induce syncope-like phenotypes in mice, suggesting that this could serve as model to study syncope in mice. Recent studies to genetically characterize the vagus sensory neurons (VSNs) has provided new insight to the coding logic of the autonomic sensory system. Classification of VSNs permits for manipulation and observation the physiological functions of these neurons. While many studies have investigated VSNs that innervate the lung or stomach, little was known about how VSNs regulate the cardiac interoceptive system. While the authors elegantly conduct experiments to characterize the behavioral, physiological, and neural phenotype of heart innervating VSN stimulation, the overall significance of the findings requires additional experiments and analysis to improve the manuscript. Below are 4 major critiques related to the significance of the findings (Major critique 1 and 2) or related to the experiment and analysis design (Major critique 3 and 4).

Major critiques

1. While the manuscript provides a new model to study syncope in mice, the paper does not deliver much new insights in how syncope is regulated or how the vagus nerve regulates heart activity. Dysfunction of the heart, or preventing blood flow to the brain, is very likely to cause syncope. Therefore, demonstrating that activation of cardiac VSNs, which likely disrupts heart activity, causes syncope is not surprising. On the other hand, the authors have identified a lot of interesting small details of syncope with this model. One example is the latency to syncope after laser stimulation. What is causing this latency? This is more interesting since the effect of laser stimulation to the hearts function and delta/theta brain waves are acute (Figure 1O and 3). Another example is the recovery from syncope. While some of the physiological dysfunction of the heart is stable throughout the laser stimulation, the animals remain in syncope for only around 10 seconds and before the offset of the stimulation. Why does the animal recover from syncope even when the heart remains dysfunctional? The authors also highlight PVZ as a specific brain area that increases neural activity during syncope. What is the function of PVZ? Does the PVZ regulate syncope? Further analysis to address at least one of these questions will significantly increase the impact of the paper. Not only will it provide new insight in how syncope is regulated, but it will also demonstrate that the optogenetic stimulation of cardiac VSN can be a good model to study syncope.

2. The authors claim that they have identified a novel genetic marker for cardiac VSNs, but the fact that Npy2r neurons innervate the heart and optogenetic activation affects heart function was demonstrated previously (Chang et al. 2015). It has already been suggested that Npy2r-expressing VSNs can be subcategorized into multiple cell types (Chang et al. 2015, Kupari et al. 2019). With the current standards, to claim that you have genetically identified a subset of neurons with a specific function, it is necessary for the authors to examine whether there is a genetic marker that separates the heart innervating VSNs from the other Npy2r+ VSNs.

3. The manuscript uses optogenetic activation of VSN fibers at AP since heart innervating VSNs project to this area while the lung VSNs do not. While this successfully evokes syncope-like phenotypes, which is a phenotype strongly associated to heart dysfunction, the authors lack control experiment to show that this stimulation is specific to the heart. The authors do not show that this stimulation method does not affect lung-related phenotypes. The manuscript also does not demonstrate how AP-projecting Npy2r+ VSNs innervate other organs such as the intestine and the stomach, making it difficult to deny the possibility of syncope could be caused by stomach or intestine innervating VSNs. The authors could use a AAVrg-FLEX-tdTomato virus to label VSNs projecting to AP, and observe if tdTomato fibers in the stomach and intestine are seen. While these control experiments are critical to evaluate the specificity of this manipulation method, the authors can just use a retrograde virus to specifically label the heart innervating VSNs as they do in Figure 1E. They have already demonstrated that heart innervating VSNs specifically project to the heart. Therefore, if they can replicate the syncope phenotype by stimulating the retrogradely labelled heart innervating VSNs, that will guarantee the phenotype is caused by heart manipulation specifically.

4. The analysis of neuropixel data is very preliminary. The analysis conducted in Figure 5 can be done in Figure 4. The pre-laser period can be treated as baseline and the change in firing rate pre vs post laser or syncope onset could be quantified. In addition, the analysis in Figure 5 is not ideal to test how light stimulation impacts neural representation of syncope in the brain. Moreover, constructing a model based on neural activity during optogenetic stimulation to predict activity during optogenetic epochs, is basically examining the quality of the constructed model. To address the authors question, they should construct a model using behavior and activity during a non-light stim period.

Furthermore, the authors should conduct more basic evaluation of the constructed model. The quality of the model is critical to interpret the analysis. The authors can evaluate the performance of the model by comparing it to a control model constructed using a shuffled dataset.

Referee #3 (Remarks to the Author):

This MS describes a genetically defined cardiac afferent nerve system that can induce bradycardia, hypotension and syncope when stimulated within the area postrema. The authors performed a tremendous amount of work, different state-of-the-art techniques and it is a very good example how tissue clearing, gene transfer and optogenetic stimulation can be used to identify and determine organ cross-talk. Since the authors observed with this stimulation the exact trias of the Betzold-Jarisch reflex, they correlate this subtype of neurons to be responsible for this reflex.

This subtype of afferent vagal neurons has been previously described to be present in the lung and have defined roles in controlling breathing.

Major points:

1. The authors observe the symptom trias of the BJR when stimulating the NP2YR positive neurons in the area postrema and conclude that they are responsible for the BJR. However, they did not stimulate the nerve endings in the heart itself and the question of how this translates to real physiological/pathophysiological activation remains.

a) How are such signals altered for example by baroreflex or other input into the AP?

b) Which kind of effect could evoke such a strong activation within the heart?

c) Are N2PYR receptor or neurons sensitive to veratum or other known BJR triggers?

d) What happens at the lower frequencies to blood pressure and the vasotone? BJR is thought to inhibit sympathetic tone. However the data now suggest rather an increase in vagal tone (Bradycardia is reduced but not systolic pump function, drop in blood pressure is most likely indirectly)

2. Overall, the presentation of pictures and videos is a bit difficult to judge. The nerve branches seem to be not homogenously dispersed throughout the ventricles with hotspots on (the apex) of right ventricle and much less on the anterior superior part, for example. The Septum cannot be seen at all and also the epi-to-endocard distribution despite using tissue clearing. In the video the atria are very dim but in the pictures one can see clearly some nerves. This is a bit puzzling since literature states that at least sensory c fiber types are equally distributed throughout the whole heart and all 4 chambers. Are the other one Piezo2 attributable? Furthermore, the statistical analysis is not clear. Is the percentage of cells analysed in regard to the total amount of all fibers in this chamber or of all positive fibers in the whole heart? If it is the latter, 15% of all in the atria is actually a lot taking the different sizes into account. Why did the authors use a t test to compare 3 groups?

3. Do you have any measures to determine cerebral blood flow to exclude indirect effects.

Ultrasound would be one option.

4. Page 13: volume sensing neurons. Additionally, this is the first attempt 1 to make an unbiased 2 activity chart of brainwide networks in combination with targeted afferent stimulation for 3 any peripheral organ.

That's an overstatement given that similar approaches have been published for the lung and gut.

Minor:

5. Stretch sensor:

In contrast to Piezo2 Neurons it is not so clear who might be the stretch sensor in such neurons.

Having the RNASeq data, are there any classical stretch sensing channels/proteins expressed?

6. The link between nausea before syncope and aversive behaviour is maybe likely but definitely not clear. Mice will avoid spaces also just because of having a syncope. Are there any other indicators for nausea?

1. Page 12: 9 turn projects to the hypothalamus (including PVN, amygdala, and reciprocal 10 connections back to the AP43–45.

End) is missing

Author Rebuttals to Initial Comments:

Response to Reviewers

We are delighted that all the reviewers are excited and have a high level of enthusiasm about our findings. Reviewer 1 called it “*beautiful and convincing*”, and “*a terrific tour de force that dramatically improves our understanding of vagal sensory afferents, cardiovascular function, and syncope. This would be of widespread interest to the readers of Nature*”. Reviewer 2 mentioned we “*elegantly conduct experiments to characterize the behavioral, physiological, and neural phenotype of heart innervating VSN stimulation*” and, “*have identified a lot of interesting small details of syncope with this model*”. Reviewer 3 said that “*The authors performed a tremendous amount of work, different state-of-the-art techniques and it is a very good example how tissue clearing, gene transfer, and optogenetic stimulation can be used to identify and determine organ cross-talk*”.

We thank the reviewers and the editor for their constructive suggestions on the specificity of the vNAS approach, detailed dissection of Bezold-Jarisch reflex (BJR), and mechanisms of syncope which have greatly improved the manuscript. We are happy to report that in this revised manuscript, we conducted almost all the experiments that the reviewers asked and included these results as 5 main figures (including 2 new ones) and doubled the number of extended data figures to 16.

Referees' comments:

Referee #1 (Remarks to the Author):

Summary and significance:

In this study Lovelace et al provide beautiful and convincing data showing that Neuropeptide Y2 (NPY2R) expressing vagal sensory neurons (VSNs) project to organs including the ventricles (but not atria) of the heart and also the lung, and that these NPY2R-VSNs form distinct populations within the nodose ganglion, supporting the notion of a one-to-one map for organ innervation by VSNs. While both subsets project to the NTS, the ventricle innervating NPY2R VSNs also strongly project to the area postrema (AP).

We are grateful that the reviewer found our work “*beautiful and convincing*” and for constructive feedback for improvement.

The authors then use an optogenetic approach to investigate the potential physiological role of these ventricle-projecting NPY2R-VSNs. Optogenetic stimulation of the AP in NPY2R-CHR2 mice (vNAS) induces a striking cluster of responses that include syncope (Fig 1), aversion/negative valence (Fig 2), Bezold-Jarisch reflex like responses (bradycardia, hypotension, reduced ventilation rate; Fig 3), and widespread inactivation of central neural activity (Figs 4).

Although quite impressive, the experiments described in Figure 5 (including the computational analysis) do not add substantially to the major findings of the paper.

We appreciate that the reviewer found our work impressive. We have now condensed the Neuropixels and computational analysis in *original Figures 4&5*, so that the most relevant findings are consolidated into a new single main *Figure 3*, and re-organized *new Extended Figures 6,7 and 8*.

New Figure 3 shows the first ever analyses of syncope that accounted for the large effect spontaneous movements have on neural activity in mice. The mouse engages in various spontaneous movements during vNAS. Additionally, the mouse ceases to move during syncope. These sorts of changes in movements can independently influence neural activity (Stringer et al, Science 2019; Musall et al, Nature Neuroscience 2019; Salkoff et al, Cerebral Cortex 2019). However, we found differences in neural activity when the mouse stopped moving during normal wakefulness versus when it stopped moving due to syncope – importantly implying that the observed reduction in brain activity *cannot* be explained by loss of movement alone, and that it is indeed a unique brain state.

Although they do provide additional evidence for neuronal inactivation during the syncope phase and indicate this inactivation is not indirectly related to facial and eye movements also caused by vNAS, they do not shed direct light on the downstream pathways that causally mediate the overt cardiac effects, other phenotypes, or the basis for neuronal inactivation.

The authors do indicate in their discussion that “Beyond clinical application, utilization of this 18 optogenetic “switch” could benefit the study of historically difficult questions regarding conscious and unconscious brain state transitions“. Although this is true, it is still completely unclear if the inactivation results from the recruitment of a hypothetical “syncope network” or simply reduced blood flow associated with the reduced cardiac output.

We appreciate the reviewer’s suggestion to shed light on downstream pathways. Our brain-wide Neuropixels data indicated that the periventricular zone (PVZ) was an ideal candidate to test since neurons in that region were activated with the shortest latency (~8ms) from the onset of vNAS. Furthermore, it was the only region that displayed increased spiking activity during syncope (*new Figure 3e,g and new Extended Figure 8c-d*). Hence, to probe the involvement of central network circuits in syncope we performed additional experiments using chemogenetics to bidirectionally manipulate the PVZ alongside vNAS. In *new Extended Figure 9*, we show that manipulating the PVZ did not alter physiological changes associated with vNAS- bradycardia and reduction in respiration-rate were the same as controls. By extension, this would imply that cardiac output would not have changed much. However, PVZ inactivation caused a significant increase in the duration of syncope and lower levels of gamma-band activity. PVZ activation had the opposite effect and caused a general state of arousal (increased locomotion and gamma activity). Thus, manipulating the PVZ had a push/pull effect on syncope without altering cardiovascular changes associated with vNAS, thereby implicating a circuit mechanism associated with syncope and not just a result of reduced cardiac output.

To directly check the consequences of reduced cardiac output, we developed an extremely challenging experimental preparation to record blood-flow in the cortex using Laser Doppler Flowmetry (LDF) during vNAS alongside ECG-EEG recordings (*new*

Figure 4). To our knowledge, this is the first time that such simultaneous data has been collected from single animals. We found that there was indeed a reduction in cerebral blood flow following vNAS, with a latency of at least 200ms from laser onset (*new Extended Figure 11a inset*), whereas brain areas like the PVZ activated ~8ms from vNAS onset (*Extended Figure 8c*). This highlights an order of magnitude difference in timescales. Therefore, a neural signal has already propagated (~8ms of laser onset) which is followed by much slower changes in cerebral blood flow (~200ms). Further, the syncope-related phenotypes (fainting) occurred 6-8s after reductions in blood flow. We interpret these new results to mean that reduction in cerebral blood flow is the likely proximal cause for syncope, and that vNAS additionally also activates a central network which then influences syncope. The PVZ is an important node of this central network as it can directly modulate syncope without altering physiological changes (*new Extended Figure 9c-d, j-k*).

If consolidated by additional experiments (see below), these data would provide the first evidence for a genetically- and anatomically defined sensory pathway involved in a ventricular-cardiac reflex arc. Although the efferent components of the reflex remain to be determined, this is a terrific tour de force that dramatically improves our understanding of vagal sensory afferents, cardiovascular function and syncope. This would be of widespread interest to the readers of Nature.

We thank the reviewer for calling our study a “terrific tour de force”

Critique:

1) Although the phenotypes induced by vNAS were very dramatic and interesting, it appears that “NPY2R-CHR2” mice were simply produced by crossing NPY2R-Cre mice to the Ai32 (cre-dependent CHR2) line. As such, AP neurons expressing NPY2R (if present) would also express CHR2, as would the axon terminals of any central NPY2R-expressing afferents to AP neurons, and therefore optogenetic stimulation of the AP area in NPY2R-CHR2 mice (“vNAS”) would not only activate the terminals of NPY2R-VSNs, but also local AP neurons that may be unrelated (not synaptically connected) to the NPY2R-VSN input. If this is the case, it cannot be concluded that the effects caused by vNAS are specifically mediated by ventricular NPY2R-VSNs because the AP is well known to be involved in the genesis of nausea and cardiac effects. Thus, additional data should be provided to prove the specificity of the vNSA approach.

We appreciate the suggestions on further characterization of the specificity of vNAS. We agree that it is necessary to rule out the effects of activating local NPY2R neurons in AP and other brain regions projecting to AP. The following approaches were applied to address these issues.

Approach 1) Direct Chr2 injection in nodose of NPY2R-Cre mice: Expressing Chr2 only in NPY2R VSNs and stimulating terminals in the AP would rule out the effects of local NPY2R neurons in the AP as well as the axon terminals of any central NPY2R-expressing afferents to AP neurons as pointed out by the reviewer. We tested at least 5 viral vectors: AAV. PHP. S-CAG-DIO-hChr2(H134R)-EYFP (PT-0474, BrainVTA), AAV. PHP.S-EF1a-DIO- hChr2(H134R)-EYFP (Gift from Dr. Li Ye), AAV2-EF1a-DIO-hChr2(H134R)-EYFP (UNC virus core), AAV5-EF1a-DIO-hChr2(H134R)-eYFP

(20298, Addgene) and AAV9-CAG-DIO- hChR2(H134R)-EYFP (Neurophotonics). We had 3-6 mice for each condition. However, none of them showed adequate expression in the nodose ganglia.

Approach 2) Anatomical tracing: We next took an anatomical approach to rule out the effects of axon terminals of any central NPY2R-expressing afferents to AP neurons. We performed direct retrograde viral tracing from the AP in NPY2R-Cre mice. In *new Extended Figure 3b-f*, we show that the AP received almost all input from the nodose ganglia, ruling out other central NPY2R axons. Furthermore, we found most fiber terminals in the heart as compared to the lung. This showed that the AP predominantly receives inputs from NPY2R cardiac VSNs and not lung VSNs or other central terminals.

Additionally, we also conducted retrograde tracing directly from multiple organ pairs- the heart ventricles/lung, heart ventricles/gut, and heart ventricles/trachea through dual color paired injections and quantified overlap in the nodose and fiber distribution in the AP. Our results show that there is a one-to-one map (i.e., distinct VSNs project to different organs) and the AP predominantly receives NPY2R input from the heart ventricles. The lung and gut NPY2R VSNs project mostly to the NTS. These data are in *new Figure 1a-j* and *new Extended Figures 2 and 3a*. This highlights the anatomical specificity of the vNAS approach in targeting NPY2R cardiac VSNs.

Approach 3) NPY2R VSN ablation: Next we used a loss of function strategy to show that the effects of vNAS were indeed via vagal afferents and not through central afferents or local neurons in the AP. To this end we applied vNAS before and after NPY2R VSN ablation. As shown in *new Extended Figure 13*, we prepared a cohort of fainting mice (NPY2R/ChR2) with vNAS. After ablating NPY2R VSNs by direct injection of AAV-flex-DTA (diphtheria toxin subunit A) into the same fainting NPY2R/ChR2 mice, the animals no longer displayed syncope with vNAS. Additionally, syncope related 50% EEG power drops were also abolished. In addition, the cardioinhibitory physiological changes associated with vNAS (bradycardia, hypotension, and reduced respiration) were also greatly blunted. Furthermore, changes in cardiac function as measured by ultrasonography with vNAS were also greatly suppressed following NPY2R nodose ablation. These data confirm that NPY2R VSNs are required for physiological and behavioral phenotypes induced by vNAS (*new Extended Figure 13*).

Taken together, these results clearly show phenotypes caused by vNAS are from activation of NPY2R VSNs (that innervate the heart) and not from local NPY2R cell bodies in the AP or central NPY2R axons.

Ideally this would include experiments in mice which express CHR2 exclusively in the ventricular-projecting VSNs (e.g. using viral delivery via the ventricle or other equivalent approach).

We thank the reviewer for this suggestion. We used multiple different strategies to address this:

Experiment 1) Direct expression of ChR2 in the nodose ganglia: We targeted heart ventricle specific VSNs using 4 separate viral strategies involving many viral vectors. 3-6 mice were used for testing each strategy.

We have used following approaches:

a. Injecting AAVrg-Cre into the heart-ventricles of Ai32 mice.

Virus: AAVrg-CAG-Cre-WPRE (Boston children hospital)/AAVrg-hsyn-Cre (105553, Addgene)

b. Injecting AAVrg-ChR2 into the heart ventricle of wildtype mice.

Virus: AAVrg-Syn-ChR2-GFP (#58880, Addgene)/ AAVrg-hsyn-ChR2-YFP (#26974, Addgene)

c. Injecting AAVrg-Cre into the heart and Cre dependent AAV-ChR2 into nodose of wild type mice.

Virus: AAVrg-hsyn-Cre (#105553, Addgene), AAVrg-hsyn-Cre-GFP (#105540, Addgene) and AAV9-ef1a-double floxed-hChR2(H134R)-EYFP (#20298, Addgene)

d. Injecting Cre dependent AAVrg-ChR2 into the heart of NPY2R Cre mice.

Virus: AAVrg-ef1a-double floxed-hChR2(H134R)-EYFP (#20298, Addgene)

However, none of these virus-based methods showed adequate expression in the nodose ganglia (*Response Figure 1*). Though puzzling at first, this is consistent with the empirical fact that virus mediated ChR2 expression in the peripheral ganglia is highly dependent on the mouse line being used and may need engineering of new serotypes (Bedbrook et al, Annual Review of Neuroscience, 2018).

Response Figure 1: Retrograde virus expression in the nodose ganglia. Scale bar, 100 μ m.

Hence, we utilized other strategies to prove our vNAS is heart-specific at the anatomical and functional level, excluding the possibility that vNAS is activating lung or gut fibers.

Experiment 2) Anatomical tracing: As mentioned earlier at the anatomical level, NPY2R VSN fiber terminals were primarily observed in the heart ventricles after retrograde tracing from the AP (*new Extended Figure 3b-f*). Additionally, retrograde tracing from the gut (stomach, small intestine, and large intestine) and lung in NPY2R-Cre mice showed terminals in the NTS and not in the AP (*new Figure 1a-j, new Extended Figure 2a-g, new Extended Figure 3a*). This indicated that vNAS predominantly targets NPY2R VSNs innervating the cardiac ventricles.

Experiment 3) Functional and anatomical isolation: We utilized a published strategy used by Dr. Ardem Patapoutian's lab (HHMI, Scripps Research) to show specificity of the baroreflex (Zeng et al, Science 2018). We directly stimulated different areas of the vagal bed in Piezo2/ChR2 (mediates the baroreflex) and NPY2R/ChR2 mice (*new Figure 5f-h and new Extended Figure 14g-j*) to directly test specificity. Since, our single cell RNA sequencing analysis shows that there is genetic segregation between NPY2R VSNs and Piezo2 VSNs (*new Extended Figure 1a-b*), we reasoned there could be functional segregation as well. In NPY2R/ChR2 mice, heart-rate and respiration-rate did not change when the optic fiber was placed above either the carotid sinus nerve (mediates the baroreflex), the superior laryngeal nerve (carries inputs from the respiratory system) or gastric branches (innervate gut) of the vagus nerve. Only while stimulating the vagus trunk (at the level of the cardiac branch), significant heart-rate and respiration-rate reductions were observed. In stark contrast, Piezo2/ChR2 mice showed bradycardia when stimulated at the carotid sinus and superior laryngeal nerve, but not at the vagal trunk (*new Figure 5f-h and new Extended Figure 14g-j*), consistent with published results (Zeng et al, Science 2018). This direct evidence clearly shows that there is genetic, anatomical, and functional segregation between the baroreflex and our vNAS strategy.

Experiment 4) NPY2R VSN ablation: To further highlight specificity and function we ablated NPY2R VSNs with AAV mediated diphtheria toxin subunit A (DTA) in NPY2R/ChR2 mice and measured physiological and behavioral changes induced by vNAS. NPY2R VSN ablation did not alter drinking behavior after water deprivation, thereby showing that gut mediated functions are unaltered (*new Extended Figure 14f*). More importantly, ablating NPY2R VSNs very specifically abolished the Bezold-Jarisch reflex while the baroreflex was intact (*new Figure 5a-c*). The Bezold-Jarisch reflex is a cardioinhibitory reflex that was speculated to be mediated by VSNs innervating the cardiac ventricles for more than a century, while the baroreflex is mediated by a different anatomical location of the cardiovascular system- the aortic arch and the carotid sinus. Taken together, our data confirms this hypothesis and at the same time highlights not just the organ specificity of NPY2R VSNs in mediating the BJR but also specificity within the cardiovascular system as well- cardiac ventricles vs the aortic arch (for the baroreflex).

Experiment 5) Comparative in-vivo/behavioral analysis: A recently published report from Dr. Yang Dan's lab (HHMI, UC-Berkeley) showed that most baroreflex sensitive

neurons project to the NTS and are also involved in sleep-wake regulation (Yao et al, Neuron 2022). They also showed that activation of these barosensitive neurons caused severe bradycardia (thereby reduced cardiac output) to the same degree as our vNAS. However, instead of syncope they reported promotion of non-REM sleep and an increase in delta EEG power. We reached out to Dr. Dan and reanalyzed their EEG data with our pipeline. We found no drops in broadband EEG power that was consistently observed with vNAS (*new Extended Figure 14k-l*), which indicated no syncope (with confirmation from Dr. Yang Dan as well). Furthermore, Dan and colleagues observed closing of the eyes with stimulation (private communication from Dr. Dan), whereas we observed pupil dilation and eyerolls with vNAS (*new Figure 3c-d and new Extended Figure 7a-b*). In sum, this further shows that despite dramatic bradycardia and thereby reduced cardiac output, stimulation of barosensitive neurons and vNAS have very different effects on brain states and behavior, thus further pointing to central circuit mechanisms at play.

Taken together, these data strongly demonstrate the specificity of our vNAS approach. NPY2R VSNs have organ-based specificities and unique fiber distribution in the brainstem. In addition, the physiological and behavioral phenotypes we observed from vNAS were mostly from the heart-ventricles but not lung, gut or baroreflex neurons.

Referee #2 (Remarks to the Author):

Lovelace and colleagues utilize genetics to specifically manipulate subtypes of vagus sensory neurons (VSNs) and study their function by observing animal's behavior as well as physiology and neural activity. They first found that a subset of Npy2r-expressing VSNs project to the AP and innervate the heart but not the lung. Furthermore, they found that optogenetic activation of these neurons induce syncope-like phenotypes in mice, suggesting that this could serve be used as model to study syncope in mice. Recent studies to genetically characterize the vagus sensory neurons (VSNs) has provided new insight to the coding logic of the autonomic sensory system. Classification of VSNs permits for manipulation and observation the physiological functions of these neurons. While many studies have investigated VSNs that innervate the lung or stomach, little was known about how VSNs regulate the cardiac interoceptive system. While the authors elegantly conduct experiments to characterize the behavioral, physiological, and neural phenotype of heart innervating VSN stimulation, the overall significance of the findings requires additional experiments and analysis to improve the manuscript. Below are 4 major critiques related to the significance of the findings (Major critique 1 and 2) or related to the experiment and analysis design (Major critique 3 and 4).

We thank the reviewer for commenting that we “elegantly conduct experiments” and providing constructive feedback for improvement.

Major critiques

1. While the manuscript provides a new model to study syncope in mice, the paper does not deliver much new insights in how syncope is regulated or how the vagus nerve regulates heart activity. Dysfunction of the heart, or preventing blood flow to the brain, is

very likely to cause syncope. Therefore, demonstrating that activation of cardiac VSNs, which likely disrupts heart activity, causes syncope is not surprising.

We thank the reviewer and agree that heart activity and blood-flow to the brain is an essential consideration when investigating syncope. Our brain-wide Neuropixels data indicated that the periventricular zone (PVZ) is functionally downstream of vNAS. It showed ~8ms latency to significant neuronal activation from laser onset and was also the only region that predominantly displayed increased spiking activity during syncope (*new Figure 3e,g and new Extended Figure 8c*). This indicated that the PVZ may be involved in modulating syncope. Hence, we used chemogenetics to bidirectionally manipulate the PVZ alongside vNAS. We found that inactivation of the PVZ did not alter the bradycardia or reduced respiration associated with vNAS but increased the duration of syncope. In contrast, activation of the PVZ caused arousal (increased locomotion and gamma activity) and quick recovery from syncope, again without altering the bradycardia and reduced respiration associated with vNAS (*new Extended Figure 9*). Since these cardiovascular changes are unaltered, it is likely that reduction in blood flow to the brain is also unchanged. Our interpretation is that reduced cerebral blood-flow is likely essential for inducing syncope, however vNAS also engages neurons in the PVZ, thereby indicating a central circuit mechanism as well.

Furthermore, a recent study from Dr. Yang Dan's lab at HHMI, UC-Berkeley (Yao et al, Neuron 2022) showed that activation of baroreflex sensitive neurons caused bradycardia and hypotension to the same degree as our vNAS. This would further imply reduced blood-flow to the brain. However, Dan and colleagues do not report syncope but promotion of non-REM sleep. We reanalysed EEG data from that report and found no drops in power (in the gamma range) but increases in delta (*new Extended Figure 14k-l*), indicating no syncope (with verbal confirmation from Dr. Yang Dan as well). Furthermore, Dan and colleagues observed closing of the eyes with stimulation (private communication from Dr. Dan), whereas we observed pupil dilation and eyerolls with vNAS (*new Figure 3c-d and new Extended Figure 7a-b*). This shows that alteration in blood flow to the brain can have varied effects on brain states: sleep-wake regulation in the case of the baroreflex, and syncope with our vNAS, thus further pointing to central circuit mechanisms at play.

On the other hand, the authors have identified a lot of interesting small details of syncope with this model.

One example is the latency to syncope after laser stimulation. What is causing this latency? This is more interesting since the effect of laser stimulation to the hearts function and delta/theta brain waves are acute (Figure 1O and 3).

We thank the reviewer for this suggestion and agree that it is important to look at details of syncope. To this end, we have conducted additional experiments combining vNAS with pretreatment of atropine (a parasympathetic output blocker). The underlying reasoning is that this would greatly blunt and delay the cardioinhibitory changes associated with vNAS. Indeed, this was the case. In *new Figure 4a-c*, we show that atropine significantly blunted the bradycardia and hypotension associated with vNAS.

Next, we developed an extremely challenging experimental preparation to directly measure cerebral blood flow alongside EEG-ECG with vNAS. To our knowledge, this is the first time that such simultaneous data has been collected from single animals. We found that syncope still occurs under atropine (*new Figure 4d-f*), but all recorded measures with respect to stimulus onset are delayed. Interestingly, our new data suggests that for syncope to occur, a threshold (in terms of both magnitude and duration) of cerebral hypoperfusion needs to be met. This could account for latency to syncope onset. We have added this to *new Figure 4g-n*, and *new Extended Figure 11* and the discussion.

We are also fascinated with the spectrum of latencies we observed from neuronal spiking to behavior observation. Instead of a uniform shutdown or activation throughout, there were distinct patterns of activity across the brain. vNAS first caused brain-wide spiking activity to significantly increase within 8-60ms across all regions (*new Extended Figure 8c-d*), while observable physiological changes (ex. heart-rate, *new Figure 2a-f* or reduced blood-flow, *new Figure 4h-m*) occurred much later ~200ms (*new Extended Figure 11a* inset). In contrast, syncope occurs substantially later ~6-8s. Atropine delayed the onset of all these latencies (*new Figure 4m* and *new Extended Figure 11e*) and reduced blood-flow drop (*new Figure 4j-l*), but syncope still occurred. Thus, it is clear that central brain circuits are engaged first, followed by physiological responses which reduces cerebral blood-flow below a certain threshold after which syncope occurs, thus accounting for the latency from laser onset.

Another example is the recovery from syncope. While some of the physiological dysfunction of the heart is stable throughout the laser stimulation, the animals remain in syncope for only around 10 seconds and before the offset of the stimulation. Why does the animal recover from syncope even when the heart remains dysfunctional?

We thank the reviewer for their comment. Indeed, syncope is a transient loss in consciousness and motor tone followed by quick recovery. We also see this with our vNAS model. We postulate that since prolonged stimulation of sensory systems often results in adaptation, i.e., a reduced response after repeated stimulation, this could be a likely explanation. We have thoroughly demonstrated an adaptation effect with vNAS on heart rates in *new Extended Figure 10d-f*, using different stimulation rates and drug conditions. The animal could recover from syncope before vNAS ends because the initial sensory signal causing downstream effects is diminished over time due to adaptation, as reflected in the heart-rate. Additionally, changes in breathing patterns over time differ compared to heart-rates. Breathing initially drops rapidly along with heart rate, but increases and consistently overshoots baseline levels, in stark contrast to heart-rate patterns (*new Figure 2b,f*). The delay in increased breathing rate could be a homeostatic response in order to increase blood-oxygen levels in response to lower heart-rate and blood-flow, thus allowing the animal to regain consciousness and motor tone even during continued vNAS. We have added this adaptation data to *new Extended Figure 10d-f* and to the discussion.

The authors also highlight PVZ as a specific brain area that increases neural activity during syncope. What is the function of PVZ? Does the PVZ regulate syncope?

We thank the reviewer for this important point. As suggested, we sought to elucidate the role of the PVZ in syncope. The PVZ is a hypothalamic region that is known to influence basic physiological and motivational processes. In our Neuropixels recordings, it was the first region to activate following laser onset and predominantly remained active throughout syncope (*new Figure 3e, new Extended Figure 8a,c*). We used chemogenetics to activate/inactivate the PVZ alongside vNAS. Our data shows that activation/inactivation of the PVZ can indeed modulate vNAS induced syncope- inactivation increased the duration of syncope and activation caused arousal and quicker recovery (*new Extended Figure 9*). Interestingly, manipulating the PVZ had no effect on vNAS induced physiological changes- bradycardia (and by extension reduced cerebral blood flow) or reduced respiration associated with vNAS (*new Extended Figure 9c-d, j-k*). This further argues for a circuit mechanism associated with syncope and has been expanded upon in the discussion as well.

Further analysis to address at least one of these questions will significantly increase the impact of the paper. Not only will it provide new insight in how syncope is regulated, but it will also demonstrate that the optogenetic stimulation of cardiac VSN can be a good model to study syncope.

We have endeavored to address all the above questions. We speculate that with vNAS a quick neural signal is transmitted across a central network of which PVZ is a node (~8ms), followed by slower physiological changes like reduction in heart-rate and cerebral hypoperfusion (~200ms), which eventually leads to syncope. Additionally, directly modulating central circuits, like PVZ could influence syncope without altering physiological changes (bradycardia and reduced respiration-rate).

2. The authors claim that they have identified a novel genetic marker for cardiac VSNs, but the fact that Npy2r neurons innervate the heart and optogenetic activation affects heart function was demonstrated previously (Chang et al. 2015). It has already been suggested that Npy2r-expressing VSNs can be subcategorized into multiple cell types (Chang et al. 2015, Kupari et al. 2019). With the current standards, to claim that you have genetically identified a subset of neurons with a specific function, it is necessary for the authors to examine whether there is a genetic marker that separates the heart innervating VSNs from the other Npy2r+ VSNs.

We thank the reviewer for this suggestion. However, recently published single cell RNA sequencing analysis of the nodose ganglia suggests that it is difficult to find explicit single markers for individual organs. There is a one-to-one map for vagal innervation of the organs but almost all of the genetic markers which have been reported target more than one organ. There may be predominant innervation by one genetic marker of one organ as compared to others, but explicit single organ genetic markers have not been reported (Zhao et al, Nature 2022). Hence, in our manuscript we achieved specificity by using Cre targeting and the fact that vagal afferents form an organ specific brainstem projection map which was confirmed by multiple groups (Han et al, Cell 2018; Ran et al, Nature 2022).

As correctly pointed out by the reviewer, NPY2R innervation of the lung has been reported (Chang et al, Cell 2015). However, the anatomical, physiological, and behavioral characteristics of any genetically defined VSN population innervating the heart has never been reported before. Hence, our manuscript detailing the characteristics of NPY2R VSNs in cardiac function is novel.

Additionally, we performed a comprehensive scRNA seq analysis of the nodose ganglia and looked for comparisons between NPY2R and other published genetic markers innervating various organs (*new Extended Figure 15a*). Furthermore, syncope has never been reported with stimulation of any other genetically defined vagal afferents. Notably, *Drd2* vagal sensory neurons which have been suggested to innervate mainly the cardiac muscle (Zhao et al, Nature 2022) has partial overlap with NPY2R. We have now added this new detailed scRNAseq analysis in *new Extended Figure 15* and the discussion.

3. The manuscript uses optogenetic activation of VSN fibers at AP since heart innervating VSNs project to this area while the lung VSNs do not. While this successfully evoke syncope-like phenotypes, which is a phenotype strongly associated to heart dysfunction, the authors lack control experiment to show that this stimulation is specific to the heart. The authors do not show that this stimulation method does not affect lung-related phenotypes.

The manuscript also does not demonstrate how AP-projecting Npy2r+ VSNs innervate other organs such as the intestine and the stomach, making it difficult to deny the possibility of syncope could be caused by stomach or intestine innervating VSNs.

The authors could use a AAVrg-FLEX-tdTomato virus to label VSNs projecting to AP, and observe if tdTomato fibers in the stomach and intestine are seen.

While these control experiments are critical to evaluate the specificity of this manipulation method, the authors can just use a retrograde virus to specifically label the heart innervating VSNs as they do in Figure 1E. They have already demonstrated that heart innervating VSNs specifically project to the heart. Therefor, if they can replicate the syncope phenotype by stimulating the retrogradely labelled heart innervating VSNs, that will guarantee the phenotype is caused by heart manipulation specifically.

We thank the reviewer for these control suggestions. To characterize the specificity of vNAS we took multiple approaches

Approach 1) Direct expression of ChR2 in the nodose ganglia: We used 4 strategies to virally deliver ChR2 to heart specific neurons in 3-6 mice for each condition.

a. Injecting AAVrg-Cre into the heart of Ai32 mice.

Virus: AAVrg -CAG-Cre-WPRE (Boston children hospital)/ AAVrg-hsyn-Cre (105553, Addgene)

b. Injecting AAVrg-ChR2 into the heart of wildtype mice.

Virus: AAVrg-Syn-ChR2-GFP (#58880, Addgene)/ AAVrg-hsyn-ChR2-YFP (#26974, Addgene)

c. Injecting AAVrg-Cre into the heart and Cre dependent AAV-ChR2 into nodose of wildtype mice.

Virus: AAVrg-hsyn-Cre (#105553, Addgene), AAVrg-hsyn-Cre-GFP (#105540, Addgene) and AAV9-ef1a-double floxed-hChR2(H134R)-EYFP (#20298, Addgene)

d. Injecting Cre dependent AAVrg-ChR2 into the heart of NPY2R Cre mice.

Virus: AAVrg-ef1a-double floxed-hChR2(H134R)-EYFP (#20298, Addgene)

As shown in *Reponse Figure 1*, no ChR2 expression in the nodose ganglia was observed. Though puzzling at first, this is consistent with the empirical fact that virus mediated ChR2 expression in the peripheral ganglia is highly dependent on the mouse line being used and may need engineering of new serotypes (Bedbrook et al, Annual Review of Neuroscience, 2018).

Hence, we used alternate approaches:

Approach 2) Anatomical tracing: We retrogradely traced from the lung, the heart ventricles and the gut (stomach, small and large intestine) and found that distinct subsets of NPY2R VSNs project to the heart, lung, and gut. The lung and gut NPY2R VSNs mainly project to the NTS whereas the AP receives predominant input from the heart VSNs. Hence, placing a fiber over the AP would mainly target heart VSNs (*new Figure 1a-j* and *new Extended Figure 1e-f, 2, 3a*). As the reviewer suggested, we also performed direct retrograde tracing from the AP in NPY2R-Cre mice. In this case as well, predominantly heart fibers were observed with very little fibers in the lung (*new Extended Figure 3b-f*). These results showed that stimulating the AP predominantly targets the heart VSNs and not lung or gut VSNs.

Approach 3) Functional and anatomical isolation: Next we employed a published strategy used by the Patapoutian Lab (HHMI, Scripps) to show functional specificity (Zheng et al, Science 2018). We directly photostimulated different areas of the vagal bed in NPY2R/ChR2 mice. We found no effect when photostimulating the superior-laryngeal nerve (which carries signals from the respiratory system), the carotid sinus nerve (has the baroreflex afferents) or the abdominal branch of the vagus nerve (which innervates the gut). Only stimulation of the vagal trunk at the level of the cardiac branch resulted in bradycardia and reduced breathing. This new data is in *new Figure 5f-h* and *new Extended Figure 14g-j*. Taken together, this further highlights organ specific distinction and confirms that the phenotypes associated with vNAS are mainly driven by the heart and not by the gut or lung.

Approach 4) NPY2R VSN ablation: To further highlight specificity and function we ablated NPY2R VSNs with AAV mediated diphtheria toxin subunit A (DTA) in NPY2R/ChR2 mice and measured physiological and behavioral changes induced by vNAS. As shown in *new Extended Figure 13*, DTA ablation blocked syncope and its related 50% EEG power drop. It also almost completely abolished reductions in heart-rate, respiration-rate and cardiac output. However, NPY2R VSN ablation did not alter drinking behavior after water deprivation, thereby showing that gut mediated functions are unaltered (*new Extended Figure 14f*). More importantly, ablating NPY2R VSNs very specifically abolished the Bezold-Jarisch reflex while the baroreflex was intact (*new*

Figure 5a-c). The Bezold-Jarisch reflex is a cardioinhibitory reflex that was speculated to be mediated by VSNs innervating the cardiac ventricles for more than a century, while the baroreflex is mediated by a different anatomical location of the cardiovascular system- the aortic arch and the carotid sinus. Taken together, our data from this approach and approach 3 above, confirms this hypothesis and at the same time highlights not just the organ specificity of NPY2R VSNs in mediating the BJR but also specificity within the cardiovascular system as well- cardiac ventricles vs the aortic arch (for the baroreflex).

Approach 5) Functional occlusion: To further show functional specificity, we performed long duration optogenetic vNAS stimulation (10 min) and found that there is adaptation in heart-rate. In this adapted state, we induced the Bezold Jarisch reflex and the baroreflex by using well established chemical injection paradigms. Specifically, the Bezold Jarisch response was impaired and the baroreflex was intact. The rationale is that with long term vNAS the BJR pathway has already adapted, so chemical injection acting on the same pathway should not elicit a strong response. In contrast, the baroreflex utilizing a separate pathway that has not been subjected to long term adaptation, had an unaffected response. This new data is in *new Figure 5d-e*. This further highlights the functional segregation between two cardiac reflexes and the specificity of vNAS with the BJR.

Approach 6) Genetic screening: Beyond our work, previous studies (Bai et al, Cell 2019; Han et al. Cell 2018) stimulated gut projecting VSNs in freely moving mice and did not find any syncope related behaviors or other cardiovascular changes. In fact, to date syncope has not been reported with stimulation of genetically defined VSNs, further showing specificity of vNAS. We did a comprehensive scRNAseq analysis of these different vagal genetic markers and their overlap NPY2R. This new data is in *new Extended Figure 15* and the discussion.

Taken together, all these different approaches from multiple scientifically strategic angles highlight the specificity of vNAS being predominantly driven by the heart. We suggest that vNAS predominantly activates heart ventricle VSNs, is necessary for the Bezold Jarisch reflex, and induces syncope.

4. The analysis of neuropixel data is very preliminary. The analysis conducted in Figure 5 can be done in Figure 4. The pre-laser period can be treated as baseline and the change in firing rate pre vs post laser or syncope onset could be quantified.

We thank the reviewer for this suggestion. We have combined the original Figure 4 and 5 into a single *new Figure 3*. We computed the baseline z-scored data in *new Extended Figure 7g-i*. The suppression is visibly apparent with 20Hz stimulation in *new Extended Data Figure 7i*. We now report the amount of suppression in the figure legend. The average suppression for syncope onset was -0.17 (z-scored units) across all recorded neurons. This difference was significantly less than zero ($p < 0.01$) for all brain areas except PVZ, BMA, PALm, PALd, and STRv, demonstrating the widespread suppression observed.

The effect size from the z-score baseline will depend on how active the neuron is before the laser stimulation – neurons with higher firing rates can exhibit larger suppression because they are starting from a higher average firing rate. Therefore, in *new Figure 3*, we chose a criterion for suppression that also captured low firing neurons by defining neurons as inactive during syncope if they did not fire after syncope onset for a time period that was unexpected (probability of less than 1%). This also allowed us to quantify the average time without firing during syncope across brain areas.

In *new Figure 3f-g* and *new Extended Figure 8*, we are ensuring that changes in firing rates are not due to the changes in the behavior of the mouse, which will not be controlled for by the z-score baselining. Several recent studies have convincingly demonstrated that spontaneous movements alone can explain a substantial proportion of neuronal spiking activity across the brain (Stringer et al, *Science* 2019; Musall et al, *Nature Neuroscience* 2019; Salkoff et al, *Cerebral Cortex* 2019). Since vNAS has obvious effects on locomotor behavior (as shown in *new Figure 1l-m*, *new Extended Figure 7a-f*, and *new Extended Figure 12a-e*), it is crucial for our study to show that wide-spread shutdown in brain activity due to vNAS is not solely a reflection of the loss of motor tone during syncope. To address this important concern, we have constructed a sophisticated model that can predict behavior-driven neural activity derived from video data and applied it to periods during laser stimulation. When we compared our movement based model's prediction against recorded observation, we report the difference between the movement model and our recorded observation as the “residual” value we show in *new Figure 3f-g* and *new Extended Figure 8e-f*.

In addition, the analysis in Figure 5 is not ideal to test how light stimulation impacts neural representation of syncope in the brain. Moreover, constructing a model based on neural activity during optogenetic stimulation to predict activity during optogenetic epochs, is basically examining the quality of the constructed model. To address the authors question, they should construct a model using behavior and activity during a non-light stim period.

We apologize for the confusion. As the reviewer pointed out, the right way to perform the analysis is indeed to exclude the optogenetic stimulation periods from training when fitting the model. We had already done this, but now we have adjusted the manuscript to make this explicitly clear in the text and the methods as well.

Furthermore, the authors should conduct more basic evaluation of the constructed model. The quality of the model is critical to interpret the analysis. The authors can evaluate the performance of the model by comparing it to a control model constructed using a shuffled dataset.

We thank the reviewer and fully agree that the quality of the model is important to the analyses. We note that all the analyses in *new Figure 3* are performed on test data because the laser stimulation is held-out from training. The short pre-laser period is also not in our training set, and therefore, all the residuals quantified in *new Figure 3g* are from test data and therefore cross-validated. Because all of our analyses were cross-

validated, and not done on the training set, there is no need for a shuffled control model, which is performed when there is no train/test split.

However, we agree more statistics on the quality of the model during the laser stimulation period and during a period without laser stimulation, as a control for the performance during the laser stimulation would be beneficial. For this quantification, the laser stimulation period was defined as 63s from -3s to 60s around the onset of the 20Hz - 30s long stimulation, and the control period was defined as a 63s period without laser stimulation. We now have explicitly mentioned this in the methods. We next binned the predictions and the neural activity in time bins of 1 second. We found that the model predictions were correlated with the true neural firing with an average $r=0.75$ in the control period across brain areas, whereas the average correlation was $r=0.44$ during the 20Hz laser stim (see *new Extended Figure 8b*). Thus, the model performed well in the absence of laser stimulation.

We note the correlations for PVZ: $r=0.91$ for the control, and $r=0.36$ for the stim. Thus, the large residuals found in the PVZ population were not due to a poor fitting of the model to the neural activity without laser stimulation, but instead due to a substantial effect of the stimulation on the PVZ population firing.

Referee #3 (Remarks to the Author):

This MS describes a genetically defined cardiac afferent nerve system that can induce bradycardia, hypotension and syncope when stimulated within the area postrema. The authors performed a tremendous amount of work, different state-of-the-art techniques and it is a very good example how tissue clearing, gene transfer and optogenetic stimulation can be used to identify and determine organ cross-talk. Since the authors observed with this stimulation the exact trias of the Betzold-Jarisch reflex, they correlate this subtype of neurons to be responsible for this reflex.

We thank the reviewer for their enthusiasm about our work, commenting that “*The authors performed a tremendous amount of work, different state-of-the-art techniques and it is a very good example how tissue clearing, gene transfer and optogenetic stimulation can be used to identify and determine organ cross-talk*” and for constructive feedback for improvement.

This subtype of afferent vagal neurons has been previously described to be present in the lung and have defined roles in controlling breathing.

Major points:

1. The authors observe the symptom trias of the BJR when stimulating the NP2YR positive neurons in the area postrema and conclude that they are responsible for the BJR. However, they did not stimulate the nerve endings in the heart itself and the question of how this translates to real physiological/pathophysiological activation remains.

We thank the reviewer for pointing out this critical question. We used multiple approaches to look at the specificity of our vNAS manipulation.

Approach 1) Direct expression of ChR2 in the nodose ganglia: We used retrograde AAVs to target heart specific VSNs. 3-6 mice were used for each strategy.

a. Injecting AAVrg-Cre into the heart of Ai32 mice.

Virus: AAVrg -CAG-Cre-WPRE (Boston children hospital)/ AAVrg-hsyn-Cre (105553, Addgene)

b. Injecting AAVrg-ChR2 into the heart of wildtype mice.

Virus: AAVrg-Syn-ChR2-GFP (#58880, Addgene)/ AAVrg-hsyn-ChR2-YFP (#26974, Addgene)

c. Injecting AAVrg-Cre into the heart and Cre dependent AAV-ChR2 into nodose of wildtype mice.

Virus: AAVrg-hsyn-Cre (#105553, Addgene), AAVrg-hsyn-Cre-GFP (#105540, Addgene) and AAV9-ef1a-double floxed-hChR2(H134R)-EYFP (#20298, Addgene)

d. Injecting Cre dependent AAVrg-ChR2 into the heart of NPY2R Cre mice.

Virus: AAVrg-ef1a-double floxed-hChR2(H134R)-EYFP (#20298, Addgene)

However, we were not able to detect expression of ChR2 expression in any condition (*Response Figure 1*). Though puzzling at first, this is a technical limitation and is consistent with the empirical fact that virus mediated ChR2 expression in the peripheral ganglia is highly dependent on the mouse line being used and may need engineering of new serotypes (Bedbrook et al, Annual Review of Neuroscience, 2018). Hence, we used alternate approaches.

Approach 2) Anatomical tracing: We used an anatomical approach to show the specificity of vNAS. As suggested by reviewer 2, we performed retrograde viral tracing from the AP in NPY2R-Cre mice. In *new Extended Figure 3b-f*, we show that the AP predominantly receives input from the nodose ganglia. Additionally, we found most fiber terminals in the heart as compared to the lung, further confirming that placing a fiber over the AP will predominantly stimulate heart specific VSNs. Additionally, we also performed dual color paired retrograde tracing from the heart ventricles/lung, heart ventricles/gut, and heart ventricles/trachea and quantified overlap in the nodose ganglia and fiber distribution patterns in the AP. Our results again show that there is a one-to-one map in organ specific innervation and the AP predominantly receives NPY2R vagal fibers from the heart ventricles. Lung and gut NPY2R VSNs project mostly to the NTS excluding the possibility that vNAS is stimulating lung or gut projecting fibers. These data are in *new Figure 1a-j* and *new Extended Figures 1e-f, 2, 3*

Approach 3) Functional and anatomical isolation: To show specificity at the functional level, we utilized a published strategy used by Dr. Ardem Patapoutian's (HHMI, Scripps) lab to highlight specificity of the baroreflex (Zeng et al, Science 2018) by directly stimulating vagal afferents. We directly stimulated different areas of the vagal bed in NPY2R/ChR2 mice. Heart-rate and respiration-rate did not change when the stimulating

optic fiber was placed above either the carotid sinus nerve (mediates the baroreflex), the superior laryngeal nerve (carries inputs from the respiratory system) or gastric branches (innervate gut). Only while stimulating the vagus trunk (at the level of the cardiac branch), significant heart-rate and respiration-rate reduction were observed. Additionally, when stimulating PIEZO2/ChR2 mice (targeting the baroreflex neurons) at the carotid sinus and superior laryngeal nerve bradycardia was observed but not at the vagal trunk (*new Figure 5f-h* and *new Extended Figure 14g-j*). Taken together, these data not only show that NPY2R VSNs have organ-based specificities but also have specificity within the cardiovascular system as well- cardiac ventricles vs the aortic arch (for the baroreflex) and that physiological phenotypes we observed in response to vNAS were primarily from the heart-ventricles and not the lung or gut.

a) How are such signals altered for example by baroreflex or other input into the AP?

We agree it is necessary to confirm that vNAS-induced phenotypes are not associated with baroreflex or other inputs into AP. As mentioned above, we designed a series of experiments to show that the AP receives NPY2R vagal inputs predominantly from the heart, and not the lung or gut.

Experiment 1) Genetic and anatomical isolation: Since the baroreflex is mediated by PIEZO2 vagal afferents, we used scRNA seq data of the nodose ganglia to show that PIEZO2 forms a distinct cluster apart from NPY2R, which we also confirmed through immunohistochemistry (*new Extended Figure 1a-d*). Additionally, PIEZO2 VSNs form a different fiber distribution pattern in the brainstem when compared to NPY2R VSNs. PIEZO2 VSNs fibers are localized in the NTS with no obvious innervation found in the AP (*Response Figure 2*), which is consistent with published literature (Nonomura et al, Nature 2016). This anatomical data indicated that our vNAS is most likely involved in a unique function that is separated from the baroreflex.

Response Figure 2: Distribution of PIEZO2 vagal fibers in the brainstem. $p < 0.01^{**}$ by paired two-tailed t-test. All error bars show mean \pm s.e.m. Scale bar, 100 μ m.

Experiment 2) NPY2R VSN ablation: We took a loss of function approach using diphtheria toxin A (DTA) delivered via AAV to the nodose to ablate NPY2R VSNs. The baroreflex can be chemically induced by retro-orbital injection of phenylephrine (PE) and sodium nitroprusside (SNP), while the Bezold Jarisch reflex (BJR) can be induced

by phenyl biguanide (PBG). These are well characterized chemicals for both of these reflexes. In the absence of NPY2R VSNs, the baroreflex could still be chemically evoked, however the BJR was nearly abolished (*new Figure 5a-c*) showing that NPY2R VSNs are necessary for the BJR but not the baroreflex.

Experiment 3) Functional occlusion: We found that heart-rate responses adapted over the course of long duration vNAS (*new Extended Data Figure 10d-e*). We leveraged this adaptation effect to show that vNAS shares common pathways with the PBG induced BJR and not the PE induced baroreflex. We performed timed injections of PE (to induce baroreflex) and PBG (to induce BJR) once heart-rate responses strongly adapted during long term vNAS. We found that the PBG induced response during the adaptation period was greatly blunted, while the phenylephrine response was unaffected. The rationale is that with long term vNAS the BJR pathway has already adapted, so chemical injection acting on the same pathway should not elicit a strong response. This further confirmed that distinct pathways mediate the BJR and the baroreflex, and that vNAS shares common signaling pathways with the BJR. This data is in *new Figure 5d-e*.

Experiment 4) Functional and anatomical isolation: To directly show functional segregation between the baroreflex and the BJR, we stimulated different areas of the vagal bed. We directly stimulated the carotid sinus and superior laryngeal nerve, known to carry baroreflex vagal afferents, in NPY2R/ChR2 and Piezo2/ChR2 mice. No changes in heart-rate and respiration-rate were observed in NPY2R/ChR2 mice but Piezo2/ChR2 (baroreflex) mice showed significant reductions in both as expected. When we stimulated the vagal trunk at the level of the cardiac branch, we observed strong reductions in heart-rate and respiration-rate in NPY2R/ChR2 but not in Piezo2/ChR2 mice. This is further evidence that NPY2R VSNs mediate the BJR and that genetic, anatomical, and functional segregation between the BJR and the baroreflex (mediated by PIEZO VSNs) exists. This new data is in *new Figure 5f-h* and *new Extended Figure 14g-j*.

Experiment 5) Comparative in-vivo/behavioral analysis: Furthermore, a recent study (Yao et al, Neuron 2022) from Dr. Yang Dan's lab (HHMI, UC-Berkeley) showed that activation of baroreflex sensitive neurons caused bradycardia and hypotension (to a similar degree as vNAS) and is involved in sleep-wake regulation. We reanalyzed EEG data from that report and found no drops in power in the gamma range (marker for syncope). This clearly shows that these genetically separated cardiac reflexes can have distinct impact on brain states and behavior. Sleep-wake regulation in the case of the baroreflex, and syncope with our vNAS associated BJR. This data is in *new Extended Figure 14k-l*. Furthermore, via private communication, Dr. Dan confirmed that stimulation of barosensitive neurons caused mice to close their eyes and sleep. In contrast, in the case of our vNAS, we observed pupil dilation, eyerolls, and syncope (*new Figure 3c-d* and *new Extended Figure 7a-b*).

Taken together, these data showed that NPY2R vagal afferents mediate the BJR, and is distinct from the baroreflex at genetic, anatomical, physiological and behavioral levels.

b) Which kind of effect could evoke such a strong activation within the heart?

It has been speculated that induction of the Bezold Jarisch reflex could cause such strong activation in the heart and associated dramatic bradycardia, hypotension and apnea. It has also been suggested that the BJR is triggered via activation of c-fibers in the heart. Additionally, the link between the Bezold Jarisch reflex and syncope has been postulated but never explicitly demonstrated. This manuscript shows all this.

NPY2R VSN ablation nearly blocked the Bezold Jarisch reflex (BJR) as induced by injection of phenyl biguanide. Interestingly, with NPY2R VSN ablation the resting heart-rate, respiration, and other cardiac parameters were not altered, showing that these neurons are not required for homeostatic maintenance of resting cardiac physiology. However, when activated, the BJR is triggered and there is strong cardioinhibition followed by syncope. This is consistent with speculations in prior literature and our new findings confirm this. We have added these data to *new Figure 5a-c and Extended Figure 14a-d*

c) Are N2PYR receptor or neurons sensitive to veratum or other known BJR triggers?

We thank the reviewer for this suggestion. As suggested, we did experiments with veratridine. However, in our hands, we were not able to get consistent heart rate changes in control mice with veratridine injection. This prevented us from drawing meaningful conclusions.

As an alternative, we used phenyl biguanide (PBG, a serotonin receptor agonist), a well-characterized chemical to induce the BJR. PBG induced a dose-dependent dip in heart-rate that was almost abolished by NPY2R VSNs ablation. We also observed that long-term vNAS decreased the sensitivity of the drug induced BJR. These new results confirmed that NPY2R VSNs are required for the BJR and that vNAS shares common neural pathways (*new Figure 5a-e, Extended Figure 14a-e*)

d) What happens at the lower frequencies to blood pressure and the vasotone? BJR is thought to inhibit sympathetic tone. However the data now suggest rather an increase in vagal tone (Bradycardia is reduced but not systolic pump function, drop in blood pressure is most likely indirectly)

At lower frequencies, there are no significant changes in blood pressure or respiration (*new Extended Figure 5g-h*). Additionally, our ultrasound imaging with lower frequency stimulation revealed that other cardiac parameters do not change much (*new Extended Figure 5d-f*).

To directly address the reviewer's question regarding inhibition of sympathetic tone vs an increase in vagal tone, we tested vNAS during atropine (a parasympathetic blocker) exposure. Pretreatment with atropine significantly reduced the bradycardia and hypotension associated with vNAS. Additionally, atropine pretreatment significantly delayed the latency to syncope onset (*new Figure 4a-f and new Extended Figure 10*). These experiments, as speculated by the reviewer, further showed that the effects of vNAS in mediating the BJR are because of increased vagal tone, and not a result of inhibited sympathetic tone.

2. Overall, the presentation of pictures and videos is a bit difficult to judge. The nerve branches seem to be not homogenously dispersed throughout the ventricles with hotspots on (the apex) of right ventricle and much less on the anterior superior part, for example.

The Septum cannot be seen at all and also the epi-to-endocard distribution despite using tissue clearing. In the video the atria are very dim but in the pictures one can see clearly some nerves.

We apologize for the confusion. We found no innervation of the septum. Images of the septum across multiple positions are shown in *Response Figure 3 and Extended Figure 1e* (see “cross section”). There are very few nerves in the atria; NPY2R VSNs predominantly innervated the ventricular wall.

We would also like to clarify that the images shown in *new Figure 1d* were collected by a confocal microscope and the video was acquired using lightsheet imaged samples, which has a lower resolution compared to confocal imaging. We have explicitly mentioned this in the methods.

Response Figure 3: Cross section of the heart in NPY2R-Cre mice with AAV-flex-GFP transduction in the nodose ganglia. No fibers are seen in the septum. Scale bar, 1000 μm .

This is a bit puzzling since literature states that at least sensory c fiber types are equally distributed throughout the whole heart and all 4 chambers. Are the other one Piezo2 attributable?

Previous literature has suggested sensory fibers are distributed across the whole heart uniformly. However, with newer papers using modern tracing and imaging methods (Zhao et al, Nature 2022; Rajendran et al, Nature Communications 2020; Shenton et al, Journal of Anatomy 2020), and including ours, it is becoming clear that there are sensory “hotspots” that are innervated by specific vagal afferents. The modern studies

indicate that there is a heterogenous distribution of sensory fibers, which may also be related to function. Furthermore, NPY2R VSNs are only a subset of VSNs innervating the heart so there may be further target preference in subregions of the heart ventricles.

Our single cell RNA seq results showed that most NPY2R neurons do not express PIEZO2, which was further confirmed by immunohistology (*new Extended Figure 1a-d*). Additionally, PIEZO2 VSNs are a-fibers and not c-fibers (Chang et al, Cell 2015; Kupari et al, Cell Reports 2019). It is also important to consider that NPY2R VSNs are only a subset of vagal afferents that innervate the heart. There should be other genetically distinct c-fiber VSNs that need to be characterized in future studies.

Furthermore, the statistical analysis is not clear. Is the percentage of cells analysed in regard to the total amount of all fibers in this chamber or of all positive fibers in the whole heart?

If it is the latter, 15% of all in the atria is actually a lot taking the different sizes into account. Why did the authors use a t test to compare 3 groups?

We apologize for the confusion. It was not the percentage of cells with regard to the chamber, but the percentage of all fibers in the whole heart. We now have added the total number of fibers in *new Figure 1d*. Also, in *new Extended Figure 1e*, we quantified the percentage area (taking into account the size) that fibers occupied in ventricle or atria. In the heart, NPY2R VSNs predominantly innervated the ventricular wall.

With regard to the statistical analysis, we reanalyzed the data using repeated measures ANOVA across regions, with the Geisser-Greenhouse correction, and Tukey's multiple comparisons post-hoc tests (*new Figure 1d*).

3. Do you have any measures to determine cerebral blood flow to exclude indirect effects. Ultrasound would be one option.

We thank the reviewer for the suggestion. We developed a challenging experimental preparation to record ECG-EEGs, pupil diameter, and cerebral blood-flow using laser doppler flowmetry (LDF) alongside vNAS (*new Figure 4g*). To our knowledge, this is the first time that such simultaneous data has been collected from single animals. We show that changes in cortical blood-flow (CBF) closely track the observed changes in heart-rate during vNAS (*new Figure 4h-i*). Next we used pretreatment with atropine (a parasympathetic blocker). We found that syncope still occurs under atropine (*new Figure 4d-f*), but all recorded measures with respect to stimulus onset are delayed (*new Figure 4i-n, new Extended Figure 11*). Interestingly, our new data suggests that for syncope to occur a threshold (both in terms of magnitude and duration) of cerebral hypoperfusion needs to be met. This could account for latency to syncope onset.

Additionally, changes in cerebral blood-flow occur only after ~200ms after the laser onset (*new Extended Figure 11a inset*). However, from our neuropixels data we found that all the brain regions are already activated by this time (latency ~8-60ms post laser onset, *new Extended Figure 8c-d*). This discrepancy in timescales argues that circuit mechanisms precede the changes in blood flow. From our neuropixels data, we found that the PVZ was the quickest brain region to activate and was also predominantly active during syncope (*new Figure 3e and new Extended Figure 8a,c*). Thus, we

reasoned that the PVZ could be playing a role in regulating syncope. We found that inactivation of the PVZ causes an increase in the duration of syncope, while activation causes arousal without changing the cardioinhibitory physiology (bradycardia and reduced respiration) associated with vNAS. This evidence shows that the PVZ may be a hub underlying the circuit mechanisms associated with syncope (*new Extended Figure 11*). In sum, it seems likely that vNAS sends out a neural signal across a central network much prior to changes in cerebral hypoperfusion. Eventually, cerebral hypoperfusion crosses a threshold which leads to syncope. Thus, reduced cerebral blood flow as well as central circuit mechanisms are important factors in the expression of syncope.

4. Page 13: *volume sensing neurons. Additionally, this is the first attempt 1 to make an unbiased 2 activity chart of brainwide networks in combination with targeted afferent stimulation for 3 any peripheral organ.*

That's an overstatement given that similar approaches have been published for the lung and gut.

We have removed the sentence from the manuscript.

Minor:

5. *Stretch sensor:*

In contrast to Piezo2 Neurons it is not so clear who might be the stretch sensor in such neurons. Having the RNASeq data, are there any classical stretch sensing channels/proteins expressed?

We thank the reviewer for this suggestion. We have now done a comprehensive scRNA seq analysis of the putative mechanosensors that are expressed in NPY2R VSNs. This data is shown in *new Extended Figure 15* and in the discussion.

6. *The link between nausea before syncope and aversive behaviour is maybe likely but definitely not clear. Mice will avoid spaces also just because of having a syncope. Are there any other indicators for nausea?*

We agree that our original link with nausea and the real time place preference assay was not clear. Since the nausea detail is not central to the scope of the newly revised manuscript, we have removed that data and the link with nausea from the text.

1. Page 12: *9 turn projects to the hypothalamus (including PVN, amygdala, and reciprocal 10 connections back to the AP43–45. End) is missing*

We have made corrections in the text.

Reviewer Reports on the First Revision:

Referees' comments:

Referee #1 (Remarks to the Author):

The authors made several attempts (with various AAV strains and constructs) to perform experiments in mice expressing CHR2 exclusively in the ventricular-projecting VSNs using organ based viral delivery, which I considered to be a critical experiment required to support the study's conclusion. However due to low expression, these experiments were unsuccessful. Instead, the authors introduced 4 additional experiments (tracing, behavior, etc) to demonstrate the specificity of the results obtained using optogenetic targeting of the AP in NPY2R-CHR2 mice. These additional experiments provide convincing evidence of the anatomical specificity of the vNAS approach. Of special note, the results obtained with NPY2R VSN ablation are quite convincing. The authors are to be commended for this effort and for solidifying the main conclusions of their study. The authors have also effectively addressed all other points raised in my review. All Editorial policies have been complied with, including statistical descriptions in the text and figure legends. Congratulations for this very exciting study.

Referee #2 (Remarks to the Author):

The revised manuscript has been dramatically improved with the addition of many new experiments analyses. The authors have fully addressed all of my previous concerns, and I think the paper is appropriate for publication in Nature. These are groundbreaking findings that will be interest to multiple fields across biological sciences. I congratulate the authors on their highly impressive work.

Referee #3 (Remarks to the Author):

The MS improved a lot and the only remaining hole in the story is the missing selective stimulation of NPY2R VSNs after retrograde tracing from the heart. However the indirect evidence is convincing that the cardiac afferent VSNs are playing a key role in the BJR. Now, the next big question is of course which physiological effect is activating them in such a strong manner/whether such activity - how it is induced by the authors: VNAS - can happen by cardiac activation. Here one experiment can be to perform voltage imaging in these neurons. In regard to the current data, I have a few points to raise:

1. The quantification of NPY2R positive VSN in ventricles versus atria is still not convincing since the atria are much thinner compared to ventricles and the calculation presented by the authors takes the overall amount of many layers (after tissue clearing) into account. Therefore it is still not answered whether these VSN have a larger number in the ventricles compared to atria just because of the different size and whether there is really a higher density of VSN. This is especially important since the authors claim later a ventricle specific effect but do not stimulate only the ventricular NPY2R VSNs.

More obvious is the epicardial site of npy2r VSNs but this can also be explained by tissue clearing

affecting fluorescence signals and/or ab staining. Thus this has to be analyzed in slices without clearing.

3. One of the most important and convincing experiments is the loss of BJR after NPY2R VSN ablation after phenyl biguanid injection and stimulation of serotonin receptors. However the drug was given systemically and how do the authors know that it was acting on the cardiac site of tissue and/or afferent nerve endings? Has this been investigated, are the receptors expressed in these nerve endings. Please provide a co-staining.

4. Please show individual data points and not only mean plus sd/sem (Extended Figure 1 b right, Figure 1d right; Extended Figure 1 e right, Figure 1g and i and many more)

5. Please explain all research results in the results parts and not in the discussion (Like the last extended figure.

Author Rebuttals to First Revision:

Response to reviewers

We are delighted that all the reviewers are excited and have a high level of enthusiasm about our findings. Reviewers 1 and 2 congratulated us for this “exciting”, “groundbreaking” and “impressive” study. Reviewer 3 said that “the MS improved a lot” and had some suggestions for improvement. For the revised manuscript, we have addressed the remaining concerns raised by reviewer 3.

Referees' comments:

Referee #1 (Remarks to the Author):

The authors made several attempts (with various AAV strains and constructs) to perform experiments in mice expressing CHR2 exclusively in the ventricular-projecting VSNs using organ based viral delivery, which I considered to be a critical experiment required to support the study's conclusion. However due to low expression, these experiments were unsuccessful. Instead, the authors introduced 4 additional experiments (tracing, behavior, etc) to demonstrate the specificity of the results obtained using optogenetic targeting of the AP in NPY2R-CHR2 mice. These additional experiments provide convincing evidence of the anatomical specificity of the vNAS approach. Of special note, the results obtained with NPY2R VSN ablation are quite convincing. The authors are to be commended for this effort and for solidifying the main conclusions of their study. The authors have also effectively addressed all other points raised in my review. All Editorial policies have been complied with, including statistical descriptions in the text and figure legends. Congratulations for this very exciting study.

We thank the reviewer for suggestions that highly improved the manuscript.

Referee #2 (Remarks to the Author):

The revised manuscript has been dramatically improved with the addition of many new experiments analyses. The authors have fully addressed all of my previous concerns, and I think the paper is appropriate for publication in Nature. These are groundbreaking findings that will be interest to multiple fields across biological sciences. I congratulate the authors on their highly impressive work.

We thank the reviewer for constructive criticism that has greatly strengthened the manuscript.

Referee #3 (Remarks to the Author):

The MS improved a lot and the only remaining hole in the story is the missing selective stimulation of NPY2R VSNs after retrograde tracing from the heart. However the indirect evidence is convincing that the cardiac afferent VSNs are playing a key role in the BJR.

We thank the reviewer for finding our experiments convincing about the role of cardiac afferents in the BJR.

Now, the next big question is of course which physiological effect is activating them in such a strong manner/whether such activity - how it is induced by the authors: VNAS - can happen by cardiac activation. Here one experiment can be to perform voltage imaging in these neurons.

We agree with the reviewer that this is indeed the next big question for a future manuscript.

In regard to the current data, I have a few points to raise:

1. The quantification of NPY2R positive VSN in ventricles versus atria is still not convincing since the atria are much thinner compared to ventricles and the calculation presented by the authors takes the overall amount of many layers (after tissue clearing) into account. Therefore it is still not answered whether these VSN have a larger number in the ventricles compared to atria just because of the different size and whether there is really a higher density of VSN. This is especially important since the authors claim later a ventricle specific effect but do not stimulate only the ventricular NPY2R VSNs.

More obvious is the epicardial site of npy2r VSNs but this can also be explained by tissue clearing affecting fluorescence signals and/or ab staining. Thus this has to be analyzed in slices without clearing.

We thank the reviewer for this suggestion. We now have done additional experiments.

- a) We have quantified the fiber innervation density in the atria and ventricles after injecting fluorophores into the nodose ganglia of NPY2R-Cre mice. Our data clearly indicates that NPY2R VSNs predominantly innervate the ventricles as compared to the atria, even after taking relative sizes into account. This new data is in Extended Figure 1e.
- b) As the reviewer suggested, we have also quantified fiber innervation density in the atria and ventricles in *slices without tissue clearing* after retro tracing from the atria and ventricles in NPY2R-Cre mice. Retro AAV-flex-GFP was injected into the atria and ventricles of the same NPY2R-Cre animal. Fibers were mostly observed in the ventricles. This is clear direct evidence that NPY2R VSNs predominantly innervate the ventricles with high fiber density. This new data is in Extended Figure 2g.

Fiber innervation density was calculated as recently published by the Marshall and Patapoutian labs (Villarino et al, *Neuron* 2023).

3. One of the most important and convincing experiments is the loss of BJR after NPY2R VSN ablation after phenyl biguanid injection and stimulation of serotonin receptors. However the drug was given systemically and how do the authors know that it was acting on the cardiac site of tissue and/or afferent nerve endings? Has this been investigated, are the receptors expressed in these nerve endings. Please provide a co-staining.

We now have performed additional experiments/analysis as the reviewer suggested.

- a) We have done scRNA seq analysis of the nodose ganglia to show overlap between serotonin receptor (5HTR3a) and NPY2R VSNs. There is some clear overlap as seen in Extended Figure 16 (bottom).
- b) We have tried doing costaining just as the reviewer suggested, in cardiac tissue to show overlap between 5HTR3a and NPY2R. As you know, antibodies in general do not work very well for receptors. On top of that, costaining in nerve terminals is technically challenging, but we tried multiple antibodies for this. However, even though some overlap is seen (Response Figure 1), the results remain inconclusive due to poor staining. Using and characterizing transgenic mice lines to look at this overlap will take a long time and is beyond the scope of this manuscript. Indeed, looking at the role of serotonin in the BJR is a very interesting future direction.

Response Figure 1: Antibody staining of NPY2R and 5HTR3A in cardiac tissue. Top row: mouse anti-5HTR3a (Santa Cruz, sc-390168); Bottom row: goat anti-5HTR3A (Novus, NB100-41382); rabbit anti-NPY2R (Neuromics, RA14112); Scale bars, 100 μ m.

Phenyl biguanide (PBG) is one of the most well characterized compounds for inducing the BJR via vagal afferents including from the heart. It has been used and published in a large number of papers over several decades (Thoren, *Reviews of Physiology, Biochemistry and Pharmacology* 1979, Coleridge and Coleridge, *Handbook of Physiology* 1979). Furthermore, even though we delivered the drug systemically, NPY2R VSNs were very specifically ablated by direct AAV-flex-DTA injections into the vagal ganglia (Figure 5a-c). This undoubtedly shows that the effects of PBG are mediated by vagal afferents. In addition, timed injections of PBG alongside vNAS very specifically occludes the BJR response (Figure 5d-e). Since vNAS mostly targets NPY2R cardiac vagal sensory neurons, this clearly shows that the effects of PBG are predominantly driven by vagal cardiac afferents. The baroreflex, another cardiovascular reflex modulated by the aortic arch is unaltered, further showing specificity that PBG mostly acts on NPY2R cardiac vagal afferents.

Reviewer Reports on the Second Revision:

Referees' comments:

Referee #3 (Remarks to the Author):

Congratulations to the authors! This MS will push the field forward and will intrigue new experiments and ideas.